# Proxy-Normalizing Activations to Match Batch Normalization while Removing Batch Dependence

Antoine Labatie    Dominic Masters    Zach Eaton-Rosen    Carlo Luschi
Graphcore Research, UK
antoine.labatie@centraliens.net
{dominicm,zacher,carlo}@graphcore.ai

## Abstract

We investigate the reasons for the performance degradation incurred with batch-independent normalization. We find that the prototypical techniques of layer normalization and instance normalization both induce the appearance of failure modes in the neural network's pre-activations: (i) layer normalization induces a collapse towards channel-wise constant functions; (ii) instance normalization induces a lack of variability in instance statistics, symptomatic of an alteration of the expressivity. To alleviate failure mode (i) without aggravating failure mode (ii), we introduce the technique "Proxy Normalization" that normalizes post-activations using a proxy distribution. When combined with layer normalization or group normalization, this batch-independent normalization emulates batch normalization's behavior and consistently matches or exceeds its performance.

## 1   Introduction

Normalization plays a critical role in deep learning as it allows successful scaling to large and deep models. In vision tasks, the most well-established normalization technique is Batch Normalization (BN) [1]. At every layer in the network, BN normalizes the intermediate activations to have zero mean and unit variance in each channel. While indisputably successful when training with large batch size, BN incurs a performance degradation in the regime of small batch size [2, 3, 4, 5, 6, 7]. This performance degradation is commonly attributed to an excessive or simply unwanted regularization stemming from the noise in the approximation of full-batch statistics by mini-batch statistics.

Many techniques have been proposed to avoid this issue, while at the same time retaining BN's benefits. Some techniques mimic BN's principle while decoupling the computational batch from the normalization batch [2, 8, 7]. Other techniques are "batch-independent" in that they operate independently of the batch in various modalities: through an explicit normalization either in activation space [9, 10, 11, 12, 13, 3, 5, 14, 15, 6] or in weight space [16, 17, 18, 19, 20]; through the use of an analytic proxy to track activation statistics [21, 22, 23]; through a change of activation function [24, 5, 15].

In this paper, we push the endeavor to replace BN with a batch-independent normalization a step further. Our main contributions are as follows: (i) we introduce a novel framework to finely characterize the various neural network properties affected by the choice of normalization; (ii) using this framework, we show that while BN's beneficial properties *are not* retained when solely using the prototypical batch-independent norms, they *are* retained when combining some of these norms with "Proxy Normalization", a novel technique that we hereby introduce; (iii) we demonstrate on an extensive set of experiments that, by reproducing BN's beneficial properties, our batch-independent normalization approach consistently matches or exceeds BN's performance.

As a starting point of our analysis, we need to gain a better understanding of those beneficial properties of BN that we aim at reproducing.

35th Conference on Neural Information Processing Systems (NeurIPS 2021).

## 2  Batch Normalization's beneficial properties

We consider throughout this paper a convolutional neural network with $d = 2$ spatial axes. This neural network receives an input $\mathbf{x} \in \mathbb{R}^{H \times W \times C_0}$ which, unless otherwise stated, is assumed sampled from a finite dataset $\mathcal{D}$. The neural network maps this input $\mathbf{x}$ to intermediate activations $\mathbf{x}^l \in \mathbb{R}^{H \times W \times C_l}$ of *height $H$*, *width $W$* and *number of channels $C_l$* at each layer $l$. The value of $\mathbf{x}^l$ at spatial position $\alpha \in \{1, \ldots, H\} \times \{1, \ldots, W\}$ and channel $c \in \{1, \ldots, C_l\}$ is denoted as $\mathbf{x}^l_{\alpha,c}$, with the dependency on $\mathbf{x}$ kept *implicit* to avoid overloading notations.

The inclusion of BN at layer $l$ leads in the full-batch setting to adding the following operations $\forall \alpha, c$:

$$\mathbf{y}^l_{\alpha,c} = \frac{\mathbf{x}^l_{\alpha,c} - \mu_c(\mathbf{x}^l)}{\sigma_c(\mathbf{x}^l)}, \qquad\qquad \tilde{\mathbf{y}}^l_{\alpha,c} = \boldsymbol{\gamma}^l_c \mathbf{y}^l_{\alpha,c} + \boldsymbol{\beta}^l_c, \qquad\qquad (1)$$

where $\mu_c(\mathbf{x}^l)$, $\sigma_c(\mathbf{x}^l)$ are the mean and standard deviation of $\mathbf{x}^l$ in channel $c$, and $\boldsymbol{\gamma}^l_c$, $\boldsymbol{\beta}^l_c$ are channel-wise scale and shift parameters restoring the degrees of freedom lost in the standardization. In the mini-batch setting, the full-batch statistics $\mu_c(\mathbf{x}^l)$, $\sigma_c(\mathbf{x}^l)$ are approximated by mini-batch statistics.

Table 1 summarizes the beneficial properties that result from including BN in the neural network. Below, we provide details on each of these properties, and we discuss whether each property is reproduced with batch-independent norms.

**Scale invariance.** When BN is present, the input-output mapping of the neural network is invariant to the scale of weights preceding any BN layer. With such scale invariance plus weight decay, the scale of weights during training reaches an equilibrium with an "effective" learning rate depending on both the learning rate and the weight decay strength [25, 26, 27, 8, 28, 29, 30]. Such mechanism of "auto rate-tuning" has been shown to provide optimization benefits [31, 26, 32].

*This property is easy to reproduce. It is already obtained with most existing batch-independent norms.*

**Control of activation scale in residual networks.** To be trainable, residual networks require the scale of activations to remain well-behaved at initialization [33, 34, 35, 20, 36, 37]. While this property naturally arises when BN is present on the residual path, when BN is not present it can also be enforced by a proper scaling decaying with the depth of the residual path. This "dilution" of the residual path with respect to the skip connection path reduces the effective depth of the network and enables to avoid coarse-grained failures modes [38, 39, 40, 41, 42, 35].

*This property is easy to reproduce. It is already obtained with most existing batch-independent norms.*

**Regularizing noise.** Due to the stochasticity of the approximation of $\mu_c(\mathbf{x}^l)$, $\sigma_c(\mathbf{x}^l)$ by mini-batch statistics, training a neural network with BN in the mini-batch setting can be seen as equivalent to performing Bayesian inference [43, 44] or to adding a regularizing term to the training of the same network with full-batch statistics [45]. As a result, BN induces a specific form of regularization.

*This regularization is not reproduced with batch-independent norms, but we leave it out of the scope of this paper.* To help minimize the bias in our analysis and "subtract away" this effect, we will perform all our experiments without and with extra degrees of regularization. This procedure can be seen as a coarse disentanglement of normalization's effects from regularization's effects.

**Avoidance of collapse.** Unnormalized networks with non-saturating nonlinearities are subject to a phenomenon of "collapse" whereby the distribution with respect to $\mathbf{x}$, $\alpha$ of the intermediate activation vectors $(\mathbf{x}^l_{\alpha,1}, \ldots, \mathbf{x}^l_{\alpha,C_l})^{\mathrm{T}}$ becomes close to zero- or one-dimensional in deep layers [39, 46, 41, 47, 48, 49, 37]. This means that deep in an unnormalized network: (i) layers tend to have their channels imbalanced; (ii) nonlinearities tend to become channel-wise linear with respect to $\mathbf{x}$, $\alpha$ and not add any effective capacity [39, 50, 51]. Consequently, unnormalized networks can neither effectively use their whole width (imbalanced channels) nor effectively use their whole depth (channel-wise linearity).

Conversely, when BN is used, the standardization at each layer prevents this collapse from happening. Even in deep layers, channels remain balanced and nonlinearities remain channel-wise nonlinear with respect to $\mathbf{x}$, $\alpha$. Consequently, networks with BN can effectively use their whole width and depth.

The collapse is, on the other hand, not always avoided with batch-independent norms [39, 18, 49]. Most notably, it is *not* avoided with Layer Normalization (LN) [9] or Group Normalization (GN) [3], as we show both theoretically and experimentally on commonly found networks in Section 4.

*To the extent possible, we aim at designing a batch-independent norm that avoids this collapse.*

Table 1: **BN's beneficial properties**. We show whether each property is (at least approximately) present (✓) or absent (✗) with various batch-independent norms: Layer Normalization (LN), Instance Normalization (IN), Layer Normalization + Proxy Normalization (LN+PN, cf Section 5). In this categorization, BN is considered in the mini-batch setting but still close to the full-batch setting, such that it approximately preserves expressivity [53].

| | Scale invariance | Control of activation scale | Regularizing noise | Avoidance of collapse | Preservation of expressivity |
|---|---|---|---|---|---|
| BN | ✓ | ✓ | ✓ | ✓ | ✓ |
| LN | ✓ | ✓ | ✗ | ✗ | ✓ |
| IN | ✓ | ✓ | ✗ | ✓ | ✗ |
| LN+PN | ✓ | ✓ | ✗ | ✓ | ✓ |

**Preservation of expressivity.** We can always express the identity with Eq. (1) by choosing $\boldsymbol{\beta}_c^l = \mu_c(\mathbf{x}^l)$ and $\boldsymbol{\gamma}_c^l = \sigma_c(\mathbf{x}^l)$. Conversely, for any choice of $\boldsymbol{\beta}_c^l, \boldsymbol{\gamma}_c^l$, we can always "re-absorb" Eq. (1) into a preceding convolution with bias. This means that BN in the full-batch setting *does not alter the expressivity* compared to an unnormalized network, i.e. it amounts to a plain reparameterization of the hypothesis space.

The expressivity is, on the other hand, not always preserved with batch-independent norms. In activation space, the dependence of batch-independent statistics on the input $\mathbf{x}$ turns the standardization into a channel-wise nonlinear operation that cannot be "re-absorbed" into a preceding convolution with bias [18]. This phenomenon is most pronounced when statistics get computed over few components. This means e.g. that Instance Normalization (IN) [10] induces a greater change of expressivity than GN, which itself induces a greater change of expressivity than LN.

In weight space, the expressivity can also be altered, namely by the removal of degrees of freedom. This is the case with Weight Standardization (WS) [18, 20] and Centered Weight Normalization [17] that remove degrees of freedom (one per unit) that *cannot* be restored in a succeeding affine transformation. This reduction of expressivity could explain the ineffectiveness of these techniques in EfficientNets [52], as previous works observed [20] and as we confirm in Section 6.

*To the extent possible, we aim at designing a batch-independent norm that preserves expressivity.*

## 3 Theoretical framework of analysis

We specified the different properties that we wish to retain in our design of batch-independent normalization: (i) scale invariance, (ii) control of activation scale; (iii) avoidance of collapse; (iv) preservation of expressivity. We now introduce a framework to quantify the presence or absence of the specific properties (iii) and (iv) with various choices of normalization.

**Propagation.** For simplicity, we assume in our theoretical setup that any layer $l$ up to depth $L$ consists of the following three steps: (i) convolution step with weights $\boldsymbol{\omega}^l \in \mathbb{R}^{K_l \times K_l \times C_{l-1} \times C_l}$; (ii) normalization step; (iii) activation step sub-decomposed into an affine transformation with scale and shift parameters $\boldsymbol{\gamma}^l, \boldsymbol{\beta}^l \in \mathbb{R}^{C_l}$ and an activation function $\phi$ which, unless otherwise stated, is assumed positive homogeneous and nonzero (e.g. $\phi = \mathrm{ReLU}$). If we denote $\mathbf{x}^l, \mathbf{y}^l, \mathbf{z}^l \in \mathbb{R}^{H \times W \times C_l}$ the intermediate activations situated just after (i), (ii), (iii) with the convention $\mathbf{z}^0 \equiv \mathbf{x}$, we may write the propagation through layer $l$ as

$$\mathbf{x}^l = \mathrm{Conv}(\mathbf{z}^{l-1}), \qquad \forall \alpha, c: \quad \mathrm{Conv}(\mathbf{z}^{l-1})_{\alpha,c} = (\boldsymbol{\omega}^l * \mathbf{z}^{l-1})_{\alpha,c}, \qquad (2)$$

$$\mathbf{y}^l = \mathrm{Norm}(\mathbf{x}^l), \qquad \forall \alpha, c: \quad \mathrm{Norm}(\mathbf{x}^l)_{\alpha,c} = \frac{\mathbf{x}_{\alpha,c}^l - \mu_{I_{\mathbf{x},c}}(\mathbf{x}^l)}{\sigma_{I_{\mathbf{x},c}}(\mathbf{x}^l)},^1 \qquad (3)$$

$$\mathbf{z}^l = \mathrm{Act}(\mathbf{y}^l), \qquad \forall \alpha, c: \quad \mathrm{Act}(\mathbf{y}^l)_{\alpha,c} = \phi(\tilde{\mathbf{y}}_{\alpha,c}^l) = \phi(\boldsymbol{\gamma}_c^l \mathbf{y}_{\alpha,c}^l + \boldsymbol{\beta}_c^l), \qquad (4)$$

where $\mu_{I_{\mathbf{x},c}}(\mathbf{x}^l), \sigma_{I_{\mathbf{x},c}}(\mathbf{x}^l)$ denote the mean and standard deviation of $\mathbf{x}^l$ conditionally on $I_{\mathbf{x},c} \equiv \{c\}$, $\{\mathbf{x}\}, \{\mathbf{x}, c\}, \{\mathbf{x}, c \bmod G\}$ for the respective cases Norm = BN, LN, IN, GN with $G$ groups.

---

[1]We omit the numerical stability constant and adopt the convention $\mathrm{Norm}(\mathbf{x}^l)_{\alpha,c} = 0, \forall \alpha$ if $\sigma_{I_{\mathbf{x},c}}(\mathbf{x}^l) = 0$.

**Moments.** Extending the previous notations, we use $\mu, \sigma, \mathcal{P}$ indexed with a (possibly empty) subset of variables to denote the operators of conditional mean, standard deviation and power. If we apply these operators to the intermediate activations $\mathbf{y}^l$, that implicitly depend on the input $\mathbf{x}$ and that explicitly depend on the spatial position $\alpha$ and the channel $c$, we get e.g.

$$\mu_c(\mathbf{y}^l) = \mathbb{E}_{\mathbf{x},\alpha}\big[\mathbf{y}^l_{\alpha,c}\big], \qquad \sigma_c(\mathbf{y}^l) = \sqrt{\mathrm{Var}_{\mathbf{x},\alpha}\big[\mathbf{y}^l_{\alpha,c}\big]}, \qquad \mathcal{P}_c(\mathbf{y}^l) = \underbrace{\mathbb{E}_{\mathbf{x},\alpha}\big[(\mathbf{y}^l_{\alpha,c})^2\big]}_{\mu_c(\mathbf{y}^l)^2 + \sigma_c(\mathbf{y}^l)^2},$$

$$\mu_{\mathbf{x},c}(\mathbf{y}^l) = \mathbb{E}_{\alpha}\big[\mathbf{y}^l_{\alpha,c}\big], \qquad \sigma_{\mathbf{x},c}(\mathbf{y}^l) = \sqrt{\mathrm{Var}_{\alpha}\big[\mathbf{y}^l_{\alpha,c}\big]}, \qquad \mathcal{P}_{\mathbf{x},c}(\mathbf{y}^l) = \underbrace{\mathbb{E}_{\alpha}\big[(\mathbf{y}^l_{\alpha,c})^2\big]}_{\mu_{\mathbf{x},c}(\mathbf{y}^l)^2 + \sigma_{\mathbf{x},c}(\mathbf{y}^l)^2},$$

where, by convention, $\mathbf{x}, \alpha, c$ are considered uniformly sampled among inputs of $\mathcal{D}$, spatial positions and channels, whenever they are considered as random.

**Power decomposition.** Using these notations, we may gain important insights by decomposing the power in channel $c$ of $\mathbf{y}^l$, just after the normalization step, as

$$\mathcal{P}_c(\mathbf{y}^l) = \underbrace{\mathbb{E}_{\mathbf{x}}\Big[\mu_{\mathbf{x},c}(\mathbf{y}^l)\Big]^2}_{\mathcal{P}_c^{(1)}(\mathbf{y}^l)} + \underbrace{\mathrm{Var}_{\mathbf{x}}\Big[\mu_{\mathbf{x},c}(\mathbf{y}^l)\Big]}_{\mathcal{P}_c^{(2)}(\mathbf{y}^l)} + \underbrace{\mathbb{E}_{\mathbf{x}}\Big[\sigma_{\mathbf{x},c}(\mathbf{y}^l)\Big]^2}_{\mathcal{P}_c^{(3)}(\mathbf{y}^l)} + \underbrace{\mathrm{Var}_{\mathbf{x}}\Big[\sigma_{\mathbf{x},c}(\mathbf{y}^l)\Big]}_{\mathcal{P}_c^{(4)}(\mathbf{y}^l)}. \qquad (5)$$

Since this four-terms *power decomposition* will be at the core of our analysis, we detail two useful views of it. The first view is that of a *hierarchy of scales*: $\mathcal{P}_c^{(1)}(\mathbf{y}^l)$ measures the power of $\mu_c(\mathbf{y}^l)$ at the dataset scale; $\mathcal{P}_c^{(2)}(\mathbf{y}^l)$ measures the power of $\mu_{\mathbf{x},c}(\mathbf{y}^l) - \mu_c(\mathbf{y}^l)$ at the instance scale; the sum of $\mathcal{P}_c^{(3)}(\mathbf{y}^l)$ and $\mathcal{P}_c^{(4)}(\mathbf{y}^l)$ measures the power of $\mathbf{y}^l_{\alpha,c} - \mu_{\mathbf{x},c}(\mathbf{y}^l)$ at the pixel scale. A particular situation where the power would be concentrated at the dataset scale with $\mathcal{P}_c^{(1)}(\mathbf{y}^l)$ equal to $\mathcal{P}_c(\mathbf{y}^l)$ would imply that $\mathbf{y}^l$ has its distribution fully "collapsed" in channel $c$, i.e. that $\mathbf{y}^l$ is constant in channel $c$.

The second view is that of a *two-level binary tree*: on one half of the tree, the sum of $\mathcal{P}_c^{(1)}(\mathbf{y}^l)$ and $\mathcal{P}_c^{(2)}(\mathbf{y}^l)$ measures the power coming from $\mu_{\mathbf{x},c}(\mathbf{y}^l)$, with the relative proportions of $\mathcal{P}_c^{(1)}(\mathbf{y}^l)$ and $\mathcal{P}_c^{(2)}(\mathbf{y}^l)$ functions of the inter-$\mathbf{x}$ similarity and inter-$\mathbf{x}$ variability of $\mu_{\mathbf{x},c}(\mathbf{y}^l)$; on the other half of the tree, the sum of $\mathcal{P}_c^{(3)}(\mathbf{y}^l)$ and $\mathcal{P}_c^{(4)}(\mathbf{y}^l)$ measures the power coming from $\sigma_{\mathbf{x},c}(\mathbf{y}^l)$, with the relative proportions of $\mathcal{P}_c^{(3)}(\mathbf{y}^l)$ and $\mathcal{P}_c^{(4)}(\mathbf{y}^l)$ functions of the inter-$\mathbf{x}$ similarity and inter-$\mathbf{x}$ variability of $\sigma_{\mathbf{x},c}(\mathbf{y}^l)$. A particular situation where $\mathcal{P}_c^{(2)}(\mathbf{y}^l)$, $\mathcal{P}_c^{(4)}(\mathbf{y}^l)$ would be equal to zero would imply that $\mu_{\mathbf{x},c}(\mathbf{y}^l)$, $\sigma_{\mathbf{x},c}(\mathbf{y}^l)$ have zero inter-$\mathbf{x}$ variability, i.e. that $\mu_{\mathbf{x},c}(\mathbf{y}^l)$, $\sigma_{\mathbf{x},c}(\mathbf{y}^l)$ are constant for all $\mathbf{x}$.

A version of Eq. (5) at the layer level instead of channel level will be easier to work with. Defining $\mathcal{P}^{(i)}(\mathbf{y}^l)$ as the averages of $\mathcal{P}_c^{(i)}(\mathbf{y}^l)$ over $c$ for $i \in \{1,2,3,4\}$, we obtain

$$\mathcal{P}(\mathbf{y}^l) = \mathcal{P}^{(1)}(\mathbf{y}^l) + \mathcal{P}^{(2)}(\mathbf{y}^l) + \mathcal{P}^{(3)}(\mathbf{y}^l) + \mathcal{P}^{(4)}(\mathbf{y}^l).$$

It should be noted that $\mathcal{P}(\mathbf{y}^l) = 1$ for any choice of $\mathrm{Norm} \in \{\mathrm{BN, LN, IN, GN}\}$ as long as the denominator of Eq. (3) is nonzero for all $\mathbf{x}, c$ [C.1]. Consequently, the terms $\mathcal{P}^{(i)}(\mathbf{y}^l)$ sum to one, meaning they can be conveniently seen as the proportion of each term $i \in \{1,2,3,4\}$ into $\mathcal{P}(\mathbf{y}^l)$.

**Revisiting BN's avoidance of collapse.** When BN is used, $\mathbf{y}^l$ is normalized not only layer-wise but also channel-wise with $\mathcal{P}_c^{(1)}(\mathbf{y}^l) = 0$ and $\mathcal{P}_c(\mathbf{y}^l) = 1$. As a first consequence, $\tilde{\mathbf{y}}^l$ (that is only one affine transformation away from $\mathbf{y}^l$) is *unlikely* to have its channel-wise distributions collapsed. This means that the nonlinearity $\phi$ acting on $\tilde{\mathbf{y}}^l$ is *likely* to be effectively nonlinear with respect to $\tilde{\mathbf{y}}^l$'s channel-wise distributions.[2] As a result, each layer adds capacity and the network effectively uses its whole *depth*. This is opposite to the situation where $\tilde{\mathbf{y}}^l$ has its channel-wise distributions collapsed with $\mathcal{P}_c(\tilde{\mathbf{y}}^l) - \mathcal{P}_c^{(1)}(\tilde{\mathbf{y}}^l) \ll \mathcal{P}_c(\tilde{\mathbf{y}}^l)$ for all $c$, which results in $\phi$ being close to linear with respect to $\tilde{\mathbf{y}}^l$'s channel-wise distributions. This is illustrated in Figure 1 and formalized in Appendix C.2.

---

[2]Note that: (i) the effective nonlinearity of $\phi$ with respect to $\tilde{\mathbf{y}}^l$'s channel-wise distributions could be quantified in the context of random nets of Definition 1 with "reasonable" choices of $\beta, \gamma$; (ii) BN only guarantees an intra-distribution nonlinearity and not an intra-mode nonlinearity in contexts such as adversarial training [54, 55] or conditional GANs [56], unless modes are decoupled in BN's computation [57, 58, 59, 60, 61].

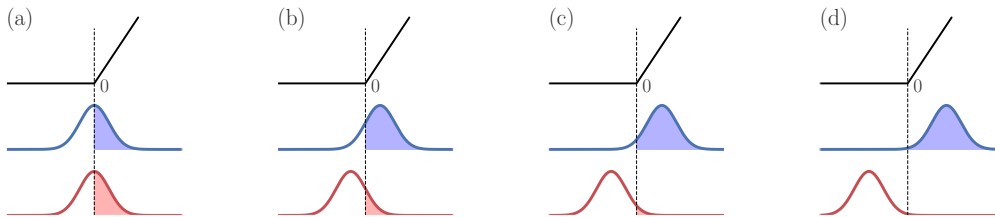

Figure 1: **Channel-wise collapse induces channel-wise linearity**. Each subplot shows $\phi = \text{ReLU}$ (black activation function) as well as two channel-wise distributions (blue and red distributions) positioned symmetrically around 0 with $\frac{\mathcal{P}_c(\tilde{\mathbf{y}}^l) - \mathcal{P}_c^{(1)}(\tilde{\mathbf{y}}^l)}{\mathcal{P}_c(\tilde{\mathbf{y}}^l)} = 1, \frac{1}{2}, \frac{1}{4}, \frac{1}{8}$ for (a), (b), (c), (d), respectively. When progressing from (a) to (d), the part of the distribution corresponding to active ReLU (shaded region) becomes either overly dominant (blue distribution) or negligible (red distribution). In either case, the channel-wise distribution ends up concentrated on only one side of piece-wise linearity.

As an additional consequence, $\mathbf{y}^l$ is guaranteed to have its channels well balanced with equal power $\mathcal{P}_c(\mathbf{y}^l)$ for all $c$. As a result, the network effectively uses its whole *width*. This is opposite to the situation where a single channel $c$ becomes overly dominant over the others with $\mathcal{P}_c(\mathbf{y}^l) \gg \mathcal{P}_{c'}(\mathbf{y}^l)$ for $c \neq c'$, which results in downstream layers only "seeing" this channel $c$ and the network behaving as if it had a width equal to one at layer $l$.

**Revisiting BN's preservation of expressivity.**    When BN is used, $\mathcal{P}_c^{(1)}(\mathbf{y}^l) = 0$ implies for all $c$ that the terms $\mathcal{P}_c^{(2)}(\mathbf{y}^l), \mathcal{P}_c^{(3)}(\mathbf{y}^l), \mathcal{P}_c^{(4)}(\mathbf{y}^l)$ sum to one. Apart from that, BN does not impose any particular constraints on the relative proportions of each term into the sum. This means that the relative proportions of $\mathcal{P}_c^{(i)}(\mathbf{y}^l)$ and $\mathcal{P}^{(i)}(\mathbf{y}^l)$ for $i \in \{2, 3, 4\}$ are free to evolve as naturally dictated by the task and the optimizer during learning.

This absence of constraints seems sensible. Indeed, imposing constraints on these relative proportions would *alter the expressivity*, which would not have any obvious justification in general and could even be detrimental in some cases, as we discuss in Section 4.

## 4    Failure modes with batch-independent normalization

With our theoretical framework in hand, we now turn to showing that the prototypical batch-independent norms are subject to failures modes opposite to BN's beneficial properties.

In the case of LN, the failure mode does not manifest in an absolute sense but rather as a "soft" inductive bias, i.e. as a preference or a favoring in the hypothesis space. This "soft" inductive bias is quantified by Theorem 1 in the context of networks with random model parameters.

**Definition 1** (random net). *We define a "random net" as a neural network having an input $\mathbf{x}$ sampled from the dataset $\mathcal{D}$ and implementing Eq. (2), (3), (4) in every layer up to depth L, with the components of $\boldsymbol{\omega}^l, \boldsymbol{\gamma}^l, \boldsymbol{\beta}^l$ at every layer $l \in \{1, \ldots, L\}$ sampled i.i.d. from the fixed distributions $\nu_{\boldsymbol{\omega}}, \nu_{\boldsymbol{\gamma}}, \nu_{\boldsymbol{\beta}}$ (up to a fan-in's square root scaling for $\boldsymbol{\omega}^l$).*

*In such networks, we assume that: (i) none of the inputs in the dataset $\mathcal{D}$ are identically zero; (ii) $\nu_{\boldsymbol{\omega}}$, $\nu_{\boldsymbol{\beta}}$, $\nu_{\boldsymbol{\gamma}}$ have well-defined moments, with strictly positive associated root mean squares $\omega, \gamma, \beta > 0$; (iii) $\nu_{\boldsymbol{\omega}}$, $\nu_{\boldsymbol{\beta}}$ are symmetric around zero.*

**Theorem 1** (layer-normalized networks collapse (informal)). *[D.3] Fix a layer $l \in \{1, \ldots, L\}$ and $\nu_{\boldsymbol{\omega}}, \nu_{\boldsymbol{\beta}}, \nu_{\boldsymbol{\gamma}}, \mathcal{D}$ in Definition 1. Further suppose $\text{Norm} = \text{LN}$ and suppose that the convolution of Eq. (2) uses periodic boundary conditions.*

*Then for random nets of Definition 1, it holds when widths are large enough that*

$$\mathcal{P}(\mathbf{y}^l) - \mathcal{P}^{(1)}(\mathbf{y}^l) \lesssim \rho^{l-1}, \qquad\qquad \mathcal{P}(\mathbf{y}^l) \simeq 1, \qquad\qquad (6)$$

*where $\rho \equiv \gamma^2/(\gamma^2 + \beta^2) < 1$, and $\lesssim$ and $\simeq$ denote inequality and equality up to arbitrarily small constants with probability arbitrarily close to 1 when $\min_{1 \leq k \leq l} C_k$ is large enough.*

**Discussion on LN's failure mode.** Theorem 1 implies that, with high probability, $\mathbf{y}^l$ is subject to *channel-wise collapse* in deep layers ($l \gg 1$) with $\mathcal{P}(\mathbf{y}^l) - \mathcal{P}^{(1)}(\mathbf{y}^l) \ll \mathcal{P}(\mathbf{y}^l)$. This means that $\tilde{\mathbf{y}}^l$ (that is only one affine transformation away from $\mathbf{y}^l$) is likely to have its channel-wise distributions collapsed with $\mathcal{P}_c(\tilde{\mathbf{y}}^l) - \mathcal{P}_c^{(1)}(\tilde{\mathbf{y}}^l) \ll \mathcal{P}_c(\tilde{\mathbf{y}}^l)$ for most $c$. The nonlinearity $\phi$ acting on $\tilde{\mathbf{y}}^l$ is then likely to be close to linear with respect to $\tilde{\mathbf{y}}^l$'s channel-wise distributions [C.2]. Being close to channel-wise linear in deep layers, layer-normalized networks are unable to effectively use their whole depth.

Since the inequality $\lesssim$ can be replaced by an equality $\simeq$ in the case $\phi = \text{identity}$ of Theorem 1 [D.4], the aggravation at each layer $l$ of the upper bound of Eq. (6) does not stem from the activation function itself but rather by the preceding affine transformation. The phenomenon of channel-wise collapse — also known under the terms of "domain collapse" [39] or "elimination singularity" [18] — is therefore not only induced by a "mean-shifting" activation function such as $\phi = \text{ReLU}$ [62, 20], but also by the injection of non-centeredness through the application of the channel-wise shift parameter $\boldsymbol{\beta}^l$ at each layer $l$. The fact that the general case of positive homogeneous $\phi$ is upper bounded by the case $\phi = \text{identity}$ in Eq. (6) still means that the choice $\phi = \text{ReLU}$ can only be an aggravating factor.

Crucially, in the context of random nets of large widths, LN's operation at each layer $l$ does not compensate this "mean shift". This comes from the fact that LN's mean and variance statistics can be approximated by zero and a constant value independent of $\mathbf{x}$, respectively. This means that LN's operation can be approximated by a layer-wise constant scaling independent of $\mathbf{x}$.[3]

The predominance of LN's failure mode in the hypothesis space — implied by its predominance in random nets — is expected to have at least two negative effects on the actual learning and final performance: (i) being expected along the training trajectory and being associated with reduced effective capacity, the failure mode is expected to cause degraded performance on the training loss; (ii) even if avoided to some extent along the training trajectory, the failure mode is still expected in the vicinity of this training trajectory, implying an ill-conditioning of the loss landscape [62, 64] and a prohibition of large learning rates that could have led otherwise to generalization benefits [65].

After detailing LN's failure mode, we now detail IN's failure mode.

**Theorem 2** (instance-normalized networks lack variability in instance statistics). [E] *Fix a layer $l \in \{1, \ldots, L\}$ and lift any assumptions on $\phi$. Further suppose* $\text{Norm} = \text{IN}$, *with Eq. (3) having nonzero denominator at layer $l$ for all inputs and channels.*

*Then it holds that*

- $\mathbf{y}^l$ *is normalized in each channel $c$ with*

$$\mathcal{P}_c^{(1)}(\mathbf{y}^l) = 0, \qquad\qquad \mathcal{P}_c(\mathbf{y}^l) = 1;$$

- $\mathbf{y}^l$ *lacks variability in instance statistics in each channel $c$ with*

$$\mathcal{P}_c^{(2)}(\mathbf{y}^l) = 0, \qquad \mathcal{P}_c^{(3)}(\mathbf{y}^l) = 1, \qquad \mathcal{P}_c^{(4)}(\mathbf{y}^l) = 0.$$

**Discussion on IN's failure mode.** We see in Theorem 2 that $\mathbf{y}^l$'s power decomposition with IN is constrained to be such that $\mathcal{P}^{(1)}(\mathbf{y}^l), \mathcal{P}^{(2)}(\mathbf{y}^l), \mathcal{P}^{(4)}(\mathbf{y}^l) = 0$ and $\mathcal{P}^{(3)}(\mathbf{y}^l) = 1$. While the constraints on $\mathcal{P}^{(1)}(\tilde{\mathbf{y}}^l), \mathcal{P}^{(3)}(\tilde{\mathbf{y}}^l)$ are removed by the affine transformation between $\mathbf{y}^l$ and $\tilde{\mathbf{y}}^l$, the constraints on $\mathcal{P}^{(2)}(\tilde{\mathbf{y}}^l), \mathcal{P}^{(4)}(\tilde{\mathbf{y}}^l)$, on the other hand, remain even after the affine transformation. These constraints on $\mathcal{P}^{(2)}(\tilde{\mathbf{y}}^l), \mathcal{P}^{(4)}(\tilde{\mathbf{y}}^l)$ translate into $2C_l$ fixed constraints in activation space that apply to each $\tilde{\mathbf{y}}^l \in \mathbb{R}^{H \times W \times C_l}$ associated with each choice of input $\mathbf{x}$ in the dataset $\mathcal{D}$ [53].

Such constraints on $\tilde{\mathbf{y}}^l$ are symptomatic of an alteration of expressivity. They notably entail that some network mappings that can be expressed without normalization cannot be expressed with IN. One such example is the identity mapping [C.3]. Another such example is a network mapping that would provide in channel $c$ through $\tilde{\mathbf{y}}_{\alpha,c}^l$, just before the nonlinearity, a detector of a given concept at position $\alpha$ in the input $\mathbf{x}$. With IN, the lack of variability in instance statistics implies that the mean $\mu_{\mathbf{x},c}(\tilde{\mathbf{y}}^l)$ and standard deviation $\sigma_{\mathbf{x},c}(\tilde{\mathbf{y}}^l)$ of the feature map in channel $c$ are necessarily constant for all $\mathbf{x}$, equal to $\boldsymbol{\beta}_c$ and $\boldsymbol{\gamma}_c$, respectively. This does not allow to express for some inputs $\mathbf{x}$ the presence of the concept at some position $\alpha$: $\mu_{\mathbf{x},c}(\tilde{\mathbf{y}}^l) > 0, \sigma_{\mathbf{x},c}(\tilde{\mathbf{y}}^l) > 0$; and for other inputs $\mathbf{x}$ the absence of the concept: $\mu_{\mathbf{x},c}(\tilde{\mathbf{y}}^l) = 0, \sigma_{\mathbf{x},c}(\tilde{\mathbf{y}}^l) = 0$.

---

[3]In this view, we expect layer-normalized networks to be also subject to a phenomenon of increasingly imbalanced channels with depth [46, 63, 48].

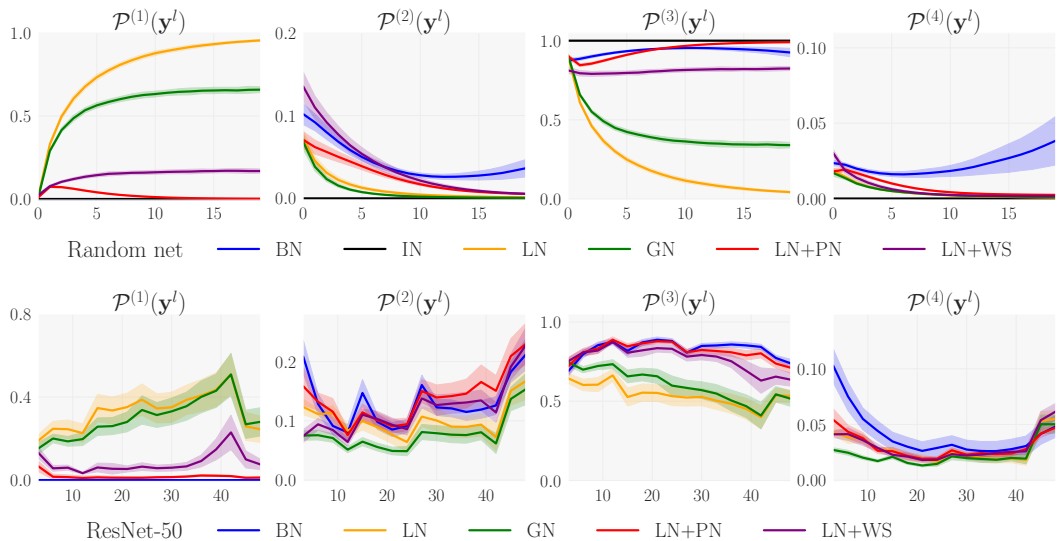

Figure 2: **"Power plots"**. The power decomposition of $\mathbf{y}^l$ is shown as a function of the depth $l$ for BN and different batch-independent norms: IN, LN, GN, LN+PN, LN+WS. Top row: Random net of Definition 1 with $\phi = \text{ReLU}$, widths $C_l = 1024$, kernel sizes $K_l = 3$, convolutions using periodic boundary conditions and components of $\boldsymbol{\beta}^l, \boldsymbol{\gamma}^l$ sampled i.i.d. from $\mathcal{N}(0, 0.2^2)$ and $\mathcal{N}(1, 0.2^2)$, respectively. Bottom row: ResNet-50 (v2) throughout 100 epochs of training on ImageNet (IN is not shown in this row due to numerical stability issues). Further experimental details are reported in Appendix A.2.

This latter example is not just anecdotal. Indeed, it is accepted that networks trained on high-level conceptual tasks have their initial layers related to low-level features and their deep layers related to high-level concepts [66, 67]. This view explains the success of IN on the low-level task of style transfer with fixed "style" input, IN being then incorporated inside a generator network that only acts on the low-level features of the "content" input [10, 68, 69, 70]. On high-level conceptual tasks, on the other hand, this view hints at a harmful tension between IN's constraints and the requirement of instance variability to express high-level concepts in deep layers. In short, with IN not only is the expressivity altered, but the alteration of expressivity results in the exclusion of useful network mappings.

**Failure modes with GN.** Group Normalization is a middle ground between the two extremes of LN ($G = 1$ group) and IN ($G = C_l$ groups at layer $l$). Networks with GN are consequently affected by both failure modes of Theorem 1 and Theorem 2, but to a lesser extent than networks with LN for the failure mode of Theorem 1 and IN for the failure mode of Theorem 2. On the one hand, since GN becomes equivalent to a constant scaling when group sizes become large, networks with GN are likely to be subject to channel-wise collapse. On the other hand, since GN can be seen as removing a fraction — with an inverse dependence on the group size — of $\mathcal{P}^{(2)}(\mathbf{y}^l), \mathcal{P}^{(4)}(\mathbf{y}^l)$ in between each layer, networks with GN are likely to lack variability in instance statistics.

The balance struck by GN between the two failure modes of LN and IN could still be beneficial, which would explain GN's superior performance in practice. It makes sense intuitively that being weakly subject to two failure modes is preferable over being strongly subject to one failure mode.

**Experimental validation.** The "power plots" of Figure 2 show the power decomposition of $\mathbf{y}^l$ as a function of the depth $l$ in both a random net of Definition 1 and ResNet-50 (v2) [71] trained on ImageNet.[4] Looking at the cases of BN and IN, LN, GN in these power plots, we confirm that: (i) unlike networks with BN and IN, networks with LN, and to a lesser extent GN, are subject to channel-wise collapse as depth increases (see Appendix A.2.1 for a precise verification of Theorem 1); (ii) networks with IN, and to a lesser extent GN, lack variability in instance statistics.

---

[4]To ensure that activation steps are directly preceded by normalization steps (cf Section 5), we always use v2 instantiations of ResNets and instantiations of ResNeXts having the same reordering of operations inside residual blocks as ResNets v2 (cf Appendix A.1).

# 5  Proxy Normalization

With the goal of remedying the failure modes of Section 4, we now introduce our novel technique *"Proxy Normalization" (PN)* that is at the core of our batch-independent normalization approach.

PN is incorporated into the neural network by replacing the activation step of Eq. (4) by the following "proxy-normalized activation step" (cf Figure 3 and the practical implementation of Appendix B):

$$\tilde{\mathbf{z}}^l = \text{PN-Act}(\mathbf{y}^l), \qquad \forall \alpha, c: \quad \text{PN-Act}(\mathbf{y}^l)_{\alpha,c} = \frac{\phi(\boldsymbol{\gamma}_c^l \mathbf{y}_{\alpha,c}^l + \boldsymbol{\beta}_c^l) - \mathbb{E}_{Y_c^l}[\phi(\boldsymbol{\gamma}_c^l Y_c^l + \boldsymbol{\beta}_c^l)]}{\sqrt{\text{Var}_{Y_c^l}[\phi(\boldsymbol{\gamma}_c^l Y_c^l + \boldsymbol{\beta}_c^l)] + \epsilon}}, \quad (7)$$

where $\epsilon \geq 0$ is a numerical stability constant and $Y_c^l$ is a Gaussian "proxy" variable of mean $\tilde{\boldsymbol{\beta}}_c^l$ and variance $(1 + \tilde{\gamma}_c^l)^2$ depending on the additional parameters $\tilde{\boldsymbol{\beta}}^l, \tilde{\boldsymbol{\gamma}}^l \in \mathbb{R}^{C_l}$ of PN. Unless otherwise stated, we let $\tilde{\boldsymbol{\beta}}^l, \tilde{\boldsymbol{\gamma}}^l$ be nonzero but still subject to weight decay and thus close to zero. We show in Appendix A.3.3 that it is also effective to let $\tilde{\boldsymbol{\beta}}^l, \tilde{\boldsymbol{\gamma}}^l$ be strictly zero and $Y_c^l \sim \mathcal{N}(0, 1)$.

If we assume (as hinted by Section 4) that only the affine transformation and the activation function $\phi$ (i.e. the activation step) play a role in the aggravation at each layer $l$ of channel-wise collapse and channel imbalance, then PN provides the following guarantee of channel-wise normalization.

**Theorem 3** (guarantee of channel-wise normalization in proxy-normalized networks (informal)). [F] *Fix a layer $l \in \{1, \ldots, L\}$ and lift any assumptions on $\phi$ and $\mathbf{x}$'s distribution. Further suppose that the neural network implements Eq. (2), (3), (7) at every layer up to depth L, with $\epsilon = 0$ and Eq. (3), (7) having nonzero denominators for all layers, inputs and channels.*

*Then both $\mathbf{y}^l$ and $\tilde{\mathbf{z}}^l$ at layer $l$ are channel-wise normalized if the following conditions hold:*

   *(i) $\tilde{\mathbf{z}}^{l-1}$ at layer $l - 1$ is channel-wise normalized;*

   *(ii) The convolution and normalization steps at layer $l$ do not cause any aggravation of channel-wise collapse and channel imbalance;*

   *(iii) $\mathbf{y}^l$ at layer $l$ is channel-wise Gaussian and PN's additional parameters $\tilde{\boldsymbol{\beta}}^l, \tilde{\boldsymbol{\gamma}}^l$ are zero.*

**Our batch-independent approach: LN+PN or GN+PN.** At this point, we crucially note that PN: (i) is batch-independent; (ii) does not cause any alteration of expressivity. *This leads us to adopt a batch-independent normalization approach that uses either LN or GN (with few groups) in the normalization step and that replaces the activation step by the proxy-normalized activation step (+PN).* With such a choice of normalization step, we guarantee three of the benefits detailed in Table 1: "scale invariance", "control of activation scale" and "preservation of expressivity". With the proxy-normalized activation step, we finally guarantee the fourth benefit of "avoidance of collapse" without compromising any of the benefits provided by the normalization step.

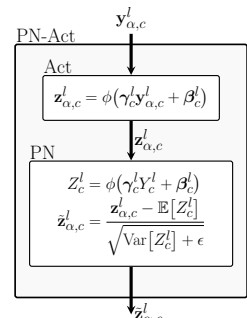

Figure 3: **Plugging PN**. PN is plugged into the activation step Act to yield the proxy-normalized activation step PN-Act.

**Experimental validation.** We confirm in Figure 2 *that BN's behavior is emulated in a fully batch-independent manner with our approach, LN+PN or GN+PN.* Indeed, the power plots of networks with LN+PN resemble the power plots of networks with BN. As desired, PN remedies LN's failure mode without incurring IN's failure mode.

**Approach strength 1: Normalizing beyond initialization.** Our batch-independent approach maintains channel-wise normalization throughout training (cf Figure 2). In contrast, many alternative approaches, either explicitly or implicitly, focus on initialization [21, 24, 17, 18, 49, 20, 72]. Centered Weight Normalization [17], WS [18, 20, 72] or PreLayerNorm [49] notably rely on the implicit assumption that different channels have the same channel-wise means after the activation function $\phi$. While valid in networks at initialization with $\boldsymbol{\beta}^l = 0$ and $\boldsymbol{\gamma}^l = 1$ (cf Appendix A.2.2), this assumption becomes less valid as the affine transformation starts deviating from the identity. We see in Figure 2 that networks with LN+WS are indeed less effective in maintaining channel-wise normalization, both in networks with random $\boldsymbol{\beta}^l, \boldsymbol{\gamma}^l$ (top row) and in networks considered throughout training (bottom row). Such a coarser channel-wise normalization might in turn lead to a less effective use of model capacity and a degradation of the conditioning of the loss landscape [62, 64].

**Approach strength 2: Wide applicability.** Our batch-independent approach matches BN with consistency across choices of model architectures, model sizes and activation functions (cf Section 6). Its only restriction, namely that activation steps should be immediately preceded by normalization steps for $\mathbf{y}^l$ and its associated proxy $Y^l$ to be at the same scale, has easy workarounds. Applicability restrictions might be more serious with alternative approaches: (i) alternative approaches involving a normalization in weight space [16, 17, 18, 19, 20] might be ill-suited to architectures with less "dense" kernels such as EfficientNets [20]; (ii) approaches involving the tracking of activation statistics [21, 22, 23] might be nontrivial to apply to residual networks [73, 74]; (iii) approaches involving a change of activation function [24, 5, 15] might precisely restrict the choice of activation function.

**Approach strength 3: Ease of implementation.** Our approach is straightforward to implement when starting from a batch-normalized network. It simply requires: (i) replacing all BN steps with LN or GN steps, and (ii) replacing all activation steps with proxy-normalized activation steps. The proxy-normalized activation steps themselves are easily implemented (cf Appendix B).

# 6 Results

We finally assess the practical performance of our batch-independent normalization approach. While we focus on ImageNet [75] in the main text of this paper, we report in Appendix A some additional results on CIFAR [76]. In Appendix A, we also provide all the details on our experimental setup.

**Choices of regularization and batch size.** As mentioned in Section 2, we perform all our experiments with different degrees of regularization to disentangle normalization's effects from regularization's effects. We detail all our choices of regularization in Appendices A.1 and A.3.4.

In terms of batch size, we set: (i) the "global batch size" in between weight updates to the same value independently of the choice of norm; (ii) the "normalization batch size" to a near-optimal value with BN. These choices enable us to be conservative when concluding on a potential advantage of our approach over BN at small batch size. Indeed, while the performance of batch-independent approaches would remain the same or slightly improve at small batch size [77, 65], the performance of BN would eventually degrade due to a regularization that eventually becomes excessive [2, 3, 4, 5, 6, 7].

**Effect of adding PN.** We start as a first experiment by analyzing the effect of adding PN on top of various norms in ResNet-50 (RN50). As visible in Table 2, PN is mostly beneficial when added on top of LN or GN with a small number of groups $G$. The consequence is that the optimal $G$ shifts to lower values in GN+PN compared to GN. This confirms the view that PN's benefit lies in addressing LN's failure mode.

It is also visible in Table 2 that PN does not provide noticeable benefits to BN. This confirms again the view that PN's benefit lies in addressing the problem — not present with BN — of channel-wise collapse. Importantly, since PN does not entail effects other than normalization that could artificially boost the performance, *GN+PN can be compared in a fair way to BN when assessing the effectiveness of normalization.*

Table 2: **Effect of adding PN**. ResNet-50 is trained on ImageNet with BN and LN, GN, GN+WS with $G$ groups, either without or with PN added on top (plain vs. PN). Results are formatted as X / Y with X, Y the validation accuracies (%) without and with extra regularization, respectively.

|  |  | RN50 | |
| --- | --- | --- | --- |
|  | $G$ | plain | +PN |
| BN |  | **76.3** / 75.8 | 76.2 / 76.0 |
| LN | 1 | 74.5 / 74.6 | 75.9 / 76.5 |
| GN | 8 | 75.4 / 75.4 | 76.3 / **76.7** |
| GN | 32 | 75.4 / 75.3 | 75.8 / 76.1 |
| GN+WS | 8 | 76.6 / 76.7 | 76.8 / **77.1** |

This is unlike WS which has been shown to improve BN's performance [18]. In our results of Table 2, the high performance of GN+WS without extra regularization and the fact that PN still provides benefits to GN+WS suggests that: (i) on top of its normalization benefits, WS induces a form of regularization; (ii) GN+WS is still not fully optimal in terms of normalization.

**GN+PN vs. BN.** Next, we turn to comparing the performance of our batch-independent approach, GN+PN, to that of BN across a broad range of models trained on ImageNet. As visible in Figure 4 and Tables 3, 4, GN+PN outperforms BN in ResNet-50 (RN50) and ResNet-101 (RN101) [71], matches BN in ResNeXt-50 (RNX50) and ResNeXt-101 (RNX101) [78], and matches BN in EfficientNet-B0 (EN-B0) and EfficientNet-B2 (EN-B2), both in the original variant with depthwise convolutions and

expansion ratio of 6 [52] and in an approximately parameter-preserving variant (cf Appendix A.1) with group convolutions of group size 16 and expansion ratio of 4 [79]. *In short, our batch-independent normalization approach, GN+PN, matches BN not only in behavior but also in performance.*

With regard to matching BN's performance with alternative norms, various positive results have been reported in ResNets and ResNeXts [8, 7, 12, 13, 15, 20, 36] but only a limited number in EfficientNets [15]. In EfficientNets, we are notably not aware of any other work showing that BN's performance can be matched with a batch-independent approach. As a confirmation, we assess the performance of various existing batch-independent approaches: GN [3], GN+WS [18], Evo-S0 [15], FRN+TLU [80, 5]. Unlike GN+PN, none of these approaches is found in Table 4 to match BN with consistency.

**Normalization and regularization.** Our results suggest that while an efficient normalization is not sufficient in itself to achieve good performance on ImageNet, it is still a necessary condition, together with regularization. In our results, it is always with extra regularization that GN+PN yields the most benefits. Importantly, the fact that GN+PN consistently leads to large improvements in training accuracy (cf Appendix A.3.2) suggests that *additional benefits would be obtained on larger datasets without the requirement of relying on regularization* [81, 72].

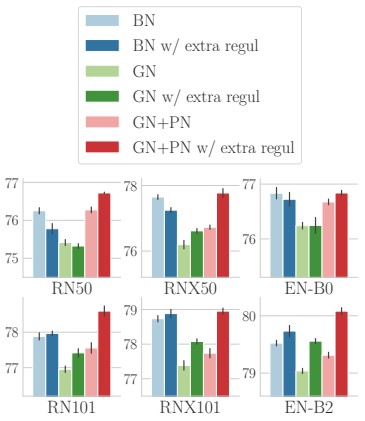

Figure 4: **BN vs. GN, GN+PN**. Validation accuracies (%) of ResNets, ResNeXts and Efficient-Nets trained on ImageNet with BN and GN, GN+PN, without and with extra regularization. Efficient-Nets are considered in the variant with group convolutions [79].

Table 3: **BN vs. GN, GN+PN**. ResNets and ResNeXts are trained on ImageNet with BN and GN, GN+PN. Results are formatted as in Table 2.

|  | RN50 | RN101 | RNX50 | RNX101 |
|---|---|---|---|---|
| BN | **76.3** / 75.8 | 77.9 / **78.0** | **77.6** / 77.2 | 78.7 / **78.9** |
| GN | 75.4 / 75.3 | 77.0 / 77.4 | 76.2 / 76.6 | 77.4 / 78.1 |
| GN+PN | 76.3 / **76.7** | 77.6 / **78.6** | 76.7 / **77.8** | 77.7 / **79.0** |

Table 4: **BN vs. batch-independent approaches**. Efficient-Nets are trained on ImageNet with BN and various batch-independent approaches. Results are formatted as in Table 2.

|  | depthwise convs [52] | | group convs [79] | |
|---|---|---|---|---|
|  | EN-B0 | EN-B2 | EN-B0 | EN-B2 |
| BN | 76.9 / **77.2** | 79.4 / **80.0** | **76.8** / 76.7 | 79.5 / **79.7** |
| GN | 76.2 / 76.2 | 78.9 / 79.4 | 76.2 / 76.2 | 79.0 / 79.6 |
| GN+PN | 76.8 / **77.0** | 79.3 / **80.0** | 76.7 / **76.8** | 79.3 / **80.1** |
| Evo-S0 | 75.8 / 75.8 | 78.5 / 78.7 | 76.2 / 76.5 | 78.9 / 79.6 |
| GN+WS | 74.2 / 74.1 | 77.8 / 77.8 | 76.2 / 76.3 | 79.2 / 79.4 |
| FRN+TLU | 75.7 / 75.7 | 78.4 / 78.8 | 74.9 / 75.1 | 78.2 / 78.6 |

# 7 Summary and broader impact

We have introduced a novel framework to finely characterize the various neural network properties affected by the choice of normalization. Using this framework, we have shown that while BN's beneficial properties are not retained when solely using the prototypical batch-independent norms, they are retained when combining some of these norms with the technique hereby introduced of Proxy Normalization. We have demonstrated on an extensive set of experiments that our batch-independent normalization approach consistently matches BN in both behavior and performance.

The main implications of this work could stem from the unlocked possibility to retain BN's normalization benefits while removing batch dependence. Firstly, our approach could be used to retain BN's normalization benefits while alleviating the burden of large activation memory stemming from BN's requirement of sufficiently large batch sizes. This is expected to be important in memory-intensive applications such as object detection or image segmentation, but also when using A.I. accelerators that leverage local memory to provide extra acceleration and energy savings in exchange for tighter memory constraints. Secondly, our approach could be used to retain BN's normalization benefits while avoiding BN's regularization when the latter is detrimental. As discussed in Section 6, this is expected to be important in the context — that will likely be prevalent in the future — of large datasets.

## Acknowledgments and Disclosure of Funding

We are thankful to Simon Knowles, Luke Hudlass-Galley, Luke Prince, Alexandros Koliousis, Anastasia Dietrich and the wider research team at Graphcore for the useful discussions and feedbacks. We are also thankful to the anonymous reviewers for their insightful comments that helped improve the paper.

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
