# A  Experimental details

## A.1  Experimental setup

**Architectures**.   As stated in Section 5, we use v2 instantiations of ResNets and instantiations of ResNeXts having the same reordering of operations inside residual blocks as ResNets v2. Practically, our instantiations of ResNeXts are obtained by starting from ResNets v2 and applying the same changes in bottleneck widths and number of groups in $3 \times 3$ convolutions as the changes yielding ResNeXts v1 from ResNets v1, using a cardinality 32 and a dimension 4 [78].

As also stated in Section 6, we consider two variants of EfficientNets: (i) the original variant with one depthwise convolution per MBConv block and with expansion ratio of 6 [52], and (ii) a variant with each depthwise convolution replaced by a group convolution of group size 16 and with expansion ratio of 4 [79]. Compared to the original variant, the variant with group convolutions has roughly the same number of parameters and slightly more floating point operations (FLOPs) (cf Table 5), but it is executed more efficiently on common A.I. accelerators. Interestingly, the fact that GN+WS does not perform well even in this variant (cf Tables 4, 8 and Figure 7) suggests that the problem related to the removal of degrees of freedom by WS goes beyond just depthwise convolutions [20].

Table 5: **Number of parameters and number of FLOPs in EfficientNets**. Quantities are reported for EfficientNets-B0 (EN-B0) and EfficientNets-B2 (EN-B2) in the variant with depthwise convolutions and expansion ratio of 6 (left) and in the variant with group convolutions of group size 16 and expansion ratio of 4 (right).

|                      | depthwise convs | | group convs | |
| --- | --- | --- | --- | --- |
|                      | EN-B0 | EN-B2 | EN-B0 | EN-B2 |
| Number of parameters | 5.3M  | 9.1M  | 5.9M  | 9.5M  |
| Number of FLOPs      | 0.4B  | 1.0B  | 0.6B  | 1.5B  |

**PN.**   We always set PN's numerical stability constant to $\epsilon = 0.03$, as we found smaller $\epsilon$ can lead to suboptimal performance. We use 200 samples uniformly sampled in probability in the proxy distribution (cf Appendix B).

In all networks, we disable the scaling part of PN in the proxy-normalized activation step just before the final mean pooling. This is to avoid an alteration of the effective learning rate. An alternative option would be to altogether remove PN before the final mean pooling.

In EfficientNets, we disable PN in squeeze-excite (SE) blocks given that no normalization step precedes each activation step in these blocks. When PN's additional parameters $\tilde{\beta}^l$, $\tilde{\gamma}^l$ are included, we replace the final affine transformation of each MBConv block by a single channel-wise scaling (i.e. we only keep the scaling parameter in the transformation). When PN's additional parameters $\tilde{\beta}^l$, $\tilde{\gamma}^l$ are omitted, on the other hand, we leave this final affine transformation as it is.

**WS.**   We set the numerical stability constant of WS to 0.

In all networks, we disable WS in fully-connected layers and in SE blocks. In ResNets and ResNeXts, we add an extra scale parameter after the final convolution of each residual block when using WS.

**Evo-S0.**   In EfficientNets, the final norm and affine transformation in each MBConv block are replaced by a single affine transformation when using Evo-S0.

**Initialization.**   We initialize the affine transformation's parameters as $\beta^l = 0$, $\gamma^l = 1$, and PN's additional parameters as $\tilde{\beta}^l = 0$, $\tilde{\gamma}^l = 0$ when these additional parameters are included. We initialize weights $\omega^l$ by sampling from truncated normal distributions with an inverse square root scaling with respect to fan-in (expect for some kernels in EfficientNets where the scaling is with respect to fan-out).

**ResNets and ResNeXts trained on ImageNet.**   We train for 100 epochs with SGD with a momentum of 0.9 and a batch size of 256. We start with a learning rate of 0.1 after a linear warmup over the

first 5 epochs [77], and we decrease this learning rate four times at the epochs 30, 60, 80 and 90, each time by a factor 10. We apply weight decay with a strength of $10^{-4}$ to all parameters including the additional parameters $\tilde{\boldsymbol{\beta}}^l$, $\tilde{\boldsymbol{\gamma}}^l$ of PN and the channel-wise scale and shift parameters $\boldsymbol{\beta}^l$, $\boldsymbol{\gamma}^l$ (this is sensible as $\phi = \mathrm{ReLU}$ is positive homogeneous).

We set the norm's numerical stability constant to $10^{-6}$ and, unless otherwise specified, we set the number of groups to $G = 8$ when using GN+PN and to $G = 32$ when using GN without PN. When using BN, we compute BN's statistics over 32 inputs $\mathbf{x}$ and we compute moving average statistics by exponentially weighted average with decay factor 0.97.

For the pre-processing, we follow [71]. When using extra regularization, we use label smoothing with factor 0.1 [82], dropout with rate 0.1 [83] and stochastic depth with rate 0.05 [84]. As the only exception, when changing the choice of the extra regularization in Appendix A.3.4, we use Mixup with strength 0.1 [85] in all networks, and in ResNet-101 and ResNeXt-101, we additionally use a variant of CutMix [86] that samples $U_0 \sim \mathrm{Uniform}(0, 1)$ and $U_1 \sim \mathrm{Uniform}(e^{-4}, 1)$ and sets the combination ratio as $\lambda = 1$ if $U_0 \leq 0.435$ and $\lambda = 1 + \frac{1}{4}\log(U_1)$ otherwise.

While we use float-16 to store and process intermediate activations (except in normalization steps and PN's statistics computation), model parameters are still stored and updated in float-32. Each time we provide a result, the mean and standard deviation are computed over 3 independent runs, at the final epoch of each run. As the only exception, the mean and $1\sigma$ intervals in the power plots of Figures 2, 6 are computed by "pooling together" either all 100 epochs in 5 independent runs (Figure 2) or the initialization state in 5 independent runs (Figure 6).

**EfficientNets with batch-independent norms trained on ImageNet.** Our experimental setup closely follows [52]. We train for 350 epochs with RMSProp with a batch size of 768. We start with a learning rate of $768 \times 2^{-14}$ (i.e. using a linear scaling [65]) after a linear warmup over the first 5 epochs [77], and we decay the learning rate exponentially by a factor 0.97 every 2.4 epochs. In RMSProp, we use a momentum of 0.9, a decay of $1.0 - (768 \times 2^{-14})$ and a numerical stability constant of $10^{-3}$. We apply weight decay with a strength of $10^{-5}$ on the convolutional weights and the additional parameters $\tilde{\boldsymbol{\beta}}^l$, $\tilde{\boldsymbol{\gamma}}^l$ of PN, but not on the other channel-wise parameters (this is sensible as $\phi = \mathrm{Swish}$ is not positive homogeneous).

We set the norm's numerical stability constant to $10^{-3}$ and we set the number of groups to $G = 4$ when using GN or Evo-S0.

For the baseline pre-processing, we follow [52]. In terms of regularization, we always use label smoothing with factor 0.1 [82], dropout with rate 0.2 [83] and stochastic depth with rate starting at 0.2 in the first MBConv block and decaying to zero linearly with the depth of the MBConv block [84]. When using extra regularization, we use Mixup with strength 0.1 [85] in all networks, and in EfficientNets-B2, we additionally use a variant of CutMix [86] that samples $U_0 \sim \mathrm{Uniform}(0, 1)$ and $U_1 \sim \mathrm{Uniform}(e^{-4}, 1)$ and sets the combination ratio as $\lambda = 1$ if $U_0 \leq 0.435$ and $\lambda = 1 + \frac{1}{4}\log(U_1)$ otherwise.

While we use float-16 to store and process intermediate activations (except in normalization steps and PN's statistics computation), model parameters are still stored and updated in float-32. Each time we provide a result, the mean and standard deviation are computed over 3 independent runs. For each run, performance is evaluated at the final epoch, with model parameters obtained by exponentially weighted average with decay factor 0.97 over checkpoints from previous epochs.

**EfficientNets with BN trained on ImageNet.** For these experiments, we run the public EfficientNet repository with the settings recommended in the repository.[5] When considering the variant with group convolutions, our only modifications consist in (i) replacing depthwise convolutions with group convolutions of group size 16, and (ii) changing the expansion ratio from 6 to 4.

In addition to BN's inherent regularization, these runs always incorporate label smoothing [82], dropout [83] and stochastic depth [84]. The runs with extra regularization additionally incorporate AutoAugment [87].

Each time we provide a result, the mean and standard deviation are computed over 3 independent runs.

---

[5] https://github.com/tensorflow/tpu/tree/master/models/official/efficientnet

**ResNets trained on CIFAR-10 and CIFAR-100 (cf Appendix A.4).** We train for 160 epochs with SGD with a momentum of 0.9 and a batch size of 128. We start with a learning rate of 0.1 after a linear warmup over the first 5 epochs [77], and we decrease this learning rate two times at the epochs 80 and 120, each time by a factor 10. We apply weight decay with a strength of $10^{-4}$ to all parameters including the additional parameters $\tilde{\beta}^l, \tilde{\gamma}^l$ of PN and the channel-wise scale and shift parameters $\beta^l, \gamma^l$ (this is sensible as $\phi = \mathrm{ReLU}$ is positive homogeneous).

We set the norm's numerical stability constant to $10^{-6}$ and we set the number of groups to $G = 4$ when using GN. When using BN, we compute BN's statistics over 128 inputs $\mathbf{x}$ and we compute moving average statistics by exponentially weighted average with decay factor 0.97.

For the pre-processing, we follow [71]. When using extra regularization, we use label smoothing with factor 0.1 [82], dropout with rate 0.25 [83] and stochastic depth with rate 0.1 [84].

We use float-16 to store and process intermediate activations (except in normalization steps and PN's statistics computation) and to store and update model parameters. Each time we provide a result, the mean and standard deviation are computed over 10 independent runs, at the final epoch of each run.

**Random nets.** We consider random nets following Definition 1. For the cases of BN, IN, LN, GN, random nets implement Eq. (2), (3), (4) at every layer $l$. For the case of LN+PN, we replace the activation step of Eq. (4) by the proxy-normalized activation step of Eq. (7). For the case of LN+WS, we add a step of kernel standardization before the convolution of Eq. (2). In all cases, convolutions use periodic boundary conditions to remain consistent with the assumptions of Theorem 1.

We set the activation function to $\phi = \mathrm{ReLU}$, widths to $C_l = 1024$, kernel sizes to $K_l = 3$.

We sample the components of the affine transformation's parameters $\beta^l, \gamma^l$ i.i.d. from $\nu_\beta = \mathcal{N}(0, 0.2^2)$ and $\nu_\gamma = \mathcal{N}(1, 0.2^2)$, respectively. This yields $\beta^2 = 0.2^2$, $\gamma^2 = 1^2 + 0.2^2$ and $\rho = \frac{1^2 + 0.2^2}{1^2 + 0.2^2 + 0.2^2} \approx 0.963$ in Definition 1. We sample the components of weight parameters $\omega^l$ i.i.d. from truncated normal distributions with $\frac{1}{\sqrt{C_l}}$ scaling. We set PN's additional parameters $\tilde{\beta}^l, \tilde{\gamma}^l$ to 0.

We set the norm's numerical stability constant to $10^{-6}$ and we set the number of groups to $G = 128$ when using GN to roughly preserve group sizes compared to ResNet-50. We use a batch size of 128 and we compute BN's statistics over all 128 inputs $\mathbf{x}$ in the mini-batch when using BN.

We use CIFAR-10 as the dataset $\mathcal{D}$ and we follow [71] for the pre-processing. To alleviate the memory burden, we add a downsampling by setting the stride to 2 in the first convolution of the network.

We use float-32 to store and process intermediate activations. Each time we provide a result, the mean and $1\sigma$ intervals are computed over 50 independent realizations.

## A.2 Additional details on power plots

### A.2.1 Power plots in random nets

**Additional experimental details.** We obtain the power plots in random nets using the experimental setup described in Appendix A.1 for random nets. We compute the terms $\mathcal{P}^{(1)}(\mathbf{y}^l)$, $\mathcal{P}^{(2)}(\mathbf{y}^l)$, $\mathcal{P}^{(3)}(\mathbf{y}^l)$, $\mathcal{P}^{(4)}(\mathbf{y}^l)$ (as well as the additional terms from Figure 5) for each layer $l$ using the 128 randomly sampled inputs $\mathbf{x}$ in the mini-batch as a proxy for the full dataset $\mathcal{D}$.

While we set the total depth to $L = 200$, we show only the first 20 layers in Figure 2 to facilitate a side-by-side comparison with ResNet-50. Indeed, while the "effective" depth is smaller than the "computational" depth in ResNet-50 (cf Appendix A.2.2), the two notions of depth coincide in random nets.

**Verification of Theorem 1.** The case of random nets with LN enables us to precisely verify Theorem 1. We provide this verification in the left and center subplots of Figure 5.

In the left subplot of Figure 5, we show $\mathcal{P}(\mathbf{y}^l) - \mathcal{P}^{(1)}(\mathbf{y}^l)$ (mean and $1\sigma$ interval) and the upper bound $\rho^{l-1} = \left( \frac{1^2 + 0.2^2}{1^2 + 0.2^2 + 0.2^2} \right)^{l-1}$ from Theorem 1 for depths $l$ up to $L = 200$. We confirm that $\mathcal{P}(\mathbf{y}^l) - \mathcal{P}^{(1)}(\mathbf{y}^l)$ is upper bounded with high probability by $\rho^{l-1}$ as predicted by Theorem 1. The rate of decay of $\mathcal{P}(\mathbf{y}^l) - \mathcal{P}^{(1)}(\mathbf{y}^l)$ is initially above the prediction of Theorem 1 due to the aggravating

effect of $\phi = \text{ReLU}$. In very deep layers ($l \gg 1$), this rate of decay ends up very slightly below the prediction of Theorem 1 due to the facts that: (i) $\phi$ becomes effectively close to channel-wise linear; (ii) the channel-wise collapse is slightly mitigated by LN in the case of a finite width $C_l = 1024$.

In the center subplot of Figure 5, we show $\mathcal{P}(\mathbf{y}^l)$ (mean and $1\sigma$ interval) for depths $l$ up to $L = 200$. We confirm that $\mathcal{P}(\mathbf{y}^l)$ is with high probability very close to one.

**Quantification of channel-wise linearity.** To confirm the connection between channel-wise collapse and channel-wise linearity, we finally report the evolution with depth of an additional measure of channel-wise linearity. In the right subplot of Figure 5, we show the measure $\mathcal{P}(\phi(\tilde{\mathbf{y}}^l) - \tilde{\mathbf{z}}^l)/\mathcal{P}(\phi(\tilde{\mathbf{y}}^l))$ (mean and $1\sigma$ interval) for depths $l$ up to $L = 200$, with $\tilde{\mathbf{z}}^l$ the channel-wise linear best-fit of $\phi(\tilde{\mathbf{y}}^l)$ using $\tilde{\mathbf{y}}^l$, that is defined in Eq. (8), (9). We confirm that deep in random nets, layers are effectively: (i) very close to channel-wise linear with LN; (ii) close to channel-wise linear with GN.

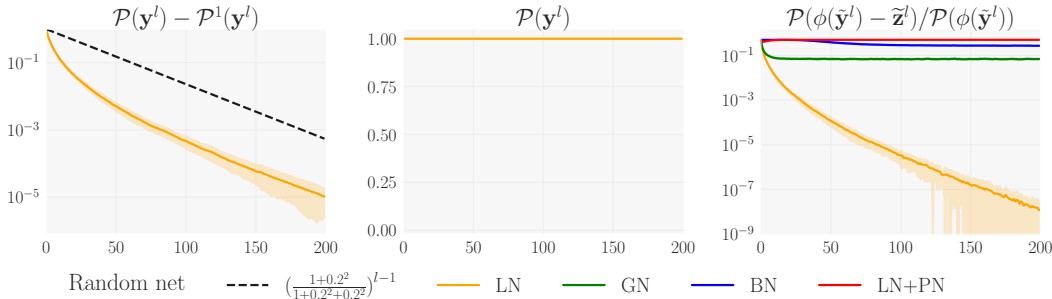

Figure 5: **Verification of Theorem 1 and quantification of channel-wise linearity**. Results are reported in random nets of Definition 1 for depths $l$ up to $L = 200$. Left: $\mathcal{P}(\mathbf{y}^l) - \mathcal{P}^{(1)}(\mathbf{y}^l)$ (mean and $1\sigma$ interval) and upper bound $\rho^{l-1}$ from Theorem 1. Center: $\mathcal{P}(\mathbf{y}^l)$ (mean and $1\sigma$ interval). Right: Additional measure of channel-wise linearity $\mathcal{P}(\phi(\tilde{\mathbf{y}}^l) - \tilde{\mathbf{z}}^l)/\mathcal{P}(\phi(\tilde{\mathbf{y}}^l))$ (mean and $1\sigma$ interval), with $\tilde{\mathbf{z}}^l$ the channel-wise linear best-fit of $\phi(\tilde{\mathbf{y}}^l)$ using $\tilde{\mathbf{y}}^l$, that is defined in Eq. (8), (9)

### A.2.2 Power plots in ResNet-50

**Additional experimental details.** We obtain the power plots in ResNet-50 using the experimental setup described in Appendix A.1 for ResNets on ImageNet. At each epoch, we compute the power terms $\mathcal{P}_c^{(1)}(\mathbf{y}^l)$, $\mathcal{P}_c^{(2)}(\mathbf{y}^l)$, $\mathcal{P}_c^{(3)}(\mathbf{y}^l)$, $\mathcal{P}_c^{(4)}(\mathbf{y}^l)$ for each layer $l$ and each channel $c$ using the last 256 randomly sampled inputs $\mathbf{x}$ as a proxy for the full dataset $\mathcal{D}$.

When looking at each norm separately in ResNets, we noticed artefacts that we attributed to the discrepancy between the "computational" depth $l$ and the "effective" depth (that oscillates with $l$). Indeed, the effective depth, defined in terms of the statistical properties of intermediate activations, grows linearly inside each residual block but gets reduced each time a residual path is summed with a skip connection path (since the latter originates from earlier layers). This phenomenon is tightly connected to the property discussed in Section 2 on the control of activation scale in residual networks.

To avoid such an artefact in Figures 2, 6, we report only a single measurement of $\mathcal{P}^{(1)}(\mathbf{y}^l)$, $\mathcal{P}^{(2)}(\mathbf{y}^l)$, $\mathcal{P}^{(3)}(\mathbf{y}^l)$, $\mathcal{P}^{(4)}(\mathbf{y}^l)$ per residual block by "pooling together" all the channels from the three norms inside each residual block. For the same reason, we do not report $\mathcal{P}^{(1)}(\mathbf{y}^l)$, $\mathcal{P}^{(2)}(\mathbf{y}^l)$, $\mathcal{P}^{(3)}(\mathbf{y}^l)$, $\mathcal{P}^{(4)}(\mathbf{y}^l)$ for the final norm just before the final mean pooling.

The presence of this artefact confirms the fact that the effective depth evolves more slowly than the computational depth $l$ in residual networks. This explains why $\mathbf{y}^l$ with LN or GN is not immoderately collapsed even at large $l$ in Figures 2, 6.

**Numerical stability issues with IN.** As stated in the caption of Figure 2, we did not succeed at training ResNet-50 v2 with IN. We found that using float-16 to store and process intermediate activations caused divergence in these networks. When replacing float-16 by float-32, even though divergence was avoided, ResNets-50 v2 still did not reach satisfactory performance with IN. We attributed this to a plain incompatibility of IN with v2 instantiations of ResNets, which could stem

from the presence of a final block of normalization and activation just before the final mean pooling. Intuitively, if we denote this final block as $L$ and if we subtract away the activation function by supposing $\phi = \mathrm{identity}$, then $\mu_{\mathbf{x},c}(\mathbf{z}^L)$ in channel $c$ is constant for all $\mathbf{x}$, equal to $\boldsymbol{\beta}_c^L$ (cf Section 4). Thus, if we subtract away the activation function, with IN all inputs $\mathbf{x}$ end up mapped to the same channel-wise constants after the final mean pooling, i.e. they become indistinguishable.

**Power plots at initialization.** Figure 6 reports the same power plots as Figure 2, except with $\mathcal{P}^{(1)}(\mathbf{y}^l), \mathcal{P}^{(2)}(\mathbf{y}^l), \mathcal{P}^{(3)}(\mathbf{y}^l), \mathcal{P}^{(4)}(\mathbf{y}^l)$ computed at initialization (mean and $1\sigma$ intervals).

When comparing Figure 6 to Figure 2, it is clearly visible that the channel-wise collapse with LN+WS gets aggravated during training compared to initialization. This confirms the importance of compensating during training the mean shift associated with the affine transformation.

It is also visible that the difference between GN and LN gets narrower during training compared to initialization. This means that despite a similar behavior of $\mathcal{P}^{(1)}(\mathbf{y}^l)$ along the training trajectories with GN and LN, differences could still exist in the vicinity of these trajectories, implying a better conditioning of the loss landscape with GN. A similar argument would make us expect a better conditioning of the loss landscape when enforcing $\mathbf{y}^l$ to be channel-wise normalized via an operation directly embedded in the network mapping [62, 64] as opposed to via an external penalty [88, 49, 89], despite the two approaches potentially leading to the same reduction of $\mathcal{P}^{(1)}(\mathbf{y}^l)$.

We believe that the notions of "channel-wise collapse" and "conditioning of the loss landscape" [62, 64] enable to quantify more accurately the underlying phenomenons at play than the notion of "internal covariate shift" [1, 90], despite the former and latter notions being connected.

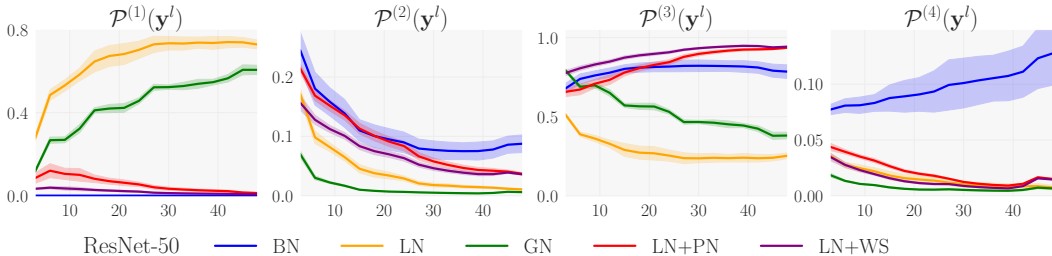

Figure 6: **Power plots at initialization**. $\mathcal{P}^{(1)}(\mathbf{y}^l), \mathcal{P}^{(2)}(\mathbf{y}^l), \mathcal{P}^{(3)}(\mathbf{y}^l), \mathcal{P}^{(4)}(\mathbf{y}^l)$ are shown as a function of the depth $l$ in ResNet-50 at initialization with BN and different batch-independent norms: LN, GN, LN+PN, LN+WS.

## A.3 More detailed results on ImageNet

### A.3.1 $1\sigma$ intervals

In Tables 6, 7, 8, we complement the results of Tables 2, 3, 4 with $1\sigma$ intervals. In Figure 7, we provide a visualization of the results of Tables 4, 8.

Table 6: **Effect of adding PN**. ResNet-50 is trained on ImageNet with BN and LN, GN, GN+WS with $G$ groups, either without or with PN added on top (plain vs. PN). Results are formatted as X / Y with X, Y the validation accuracies (mean and $1\sigma$ interval in %) without and with extra regularization, respectively.

|  |  | RN50 | |
| --- | --- | --- | --- |
|  | $G$ | plain | +PN |
| BN |  | **76.3**±0.1 / 75.8±0.2 | 76.2±0.1 / 76.0±0.1 |
| LN | 1 | 74.5±0.0 / 74.6±0.1 | 75.9±0.1 / 76.5±0.0 |
| GN | 8 | 75.4±0.1 / 75.4±0.1 | 76.3±0.1 / **76.7**±0.0 |
| GN | 32 | 75.4±0.1 / 75.3±0.1 | 75.8±0.2 / 76.1±0.1 |
| GN+WS | 8 | 76.6±0.0 / 76.7±0.1 | 76.8±0.1 / **77.1**±0.1 |

Table 7: **BN vs. GN, GN+PN**. ResNets and ResNeXts are trained on ImageNet with BN and GN, GN+PN. Results are formatted as in Table 6.

|       | RN50 | RN101 | RNX50 | RNX101 |
|-------|------|-------|-------|--------|
| BN    | **76.3**$_{\pm 0.1}$ / 75.8$_{\pm 0.2}$ | 77.9$_{\pm 0.1}$ / **78.0**$_{\pm 0.1}$ | **77.6**$_{\pm 0.1}$ / 77.2$_{\pm 0.1}$ | 78.7$_{\pm 0.1}$ / **78.9**$_{\pm 0.1}$ |
| GN    | 75.4$_{\pm 0.1}$ / 75.3$_{\pm 0.1}$ | 77.0$_{\pm 0.1}$ / 77.4$_{\pm 0.1}$ | 76.2$_{\pm 0.2}$ / 76.6$_{\pm 0.1}$ | 77.4$_{\pm 0.2}$ / 78.1$_{\pm 0.1}$ |
| GN+PN | 76.3$_{\pm 0.1}$ / **76.7**$_{\pm 0.0}$ | 77.6$_{\pm 0.2}$ / **78.6**$_{\pm 0.2}$ | 76.7$_{\pm 0.1}$ / **77.8**$_{\pm 0.2}$ | 77.7$_{\pm 0.2}$ / **79.0**$_{\pm 0.1}$ |

Table 8: **BN vs. batch-independent approaches**. EfficientNets are trained on ImageNet with BN and various batch-independent approaches: GN, GN+PN, Evo-S0, GN+WS, FRN+TLU. Results are formatted as in Table 6.

|        | depthwise convs | | group convs | |
|--------|------|------|------|------|
|        | EN-B0 | EN-B2 | EN-B0 | EN-B2 |
| BN     | 76.9$_{\pm 0.1}$ / **77.2**$_{\pm 0.1}$ | 79.4$_{\pm 0.0}$ / **80.0**$_{\pm 0.0}$ | **76.8**$_{\pm 0.1}$ / 76.7$_{\pm 0.2}$ | 79.5$_{\pm 0.1}$ / **79.7**$_{\pm 0.1}$ |
| GN     | 76.2$_{\pm 0.1}$ / 76.2$_{\pm 0.1}$ | 78.9$_{\pm 0.1}$ / 79.4$_{\pm 0.1}$ | 76.2$_{\pm 0.1}$ / 76.2$_{\pm 0.2}$ | 79.0$_{\pm 0.1}$ / 79.6$_{\pm 0.1}$ |
| GN+PN  | 76.8$_{\pm 0.0}$ / **77.0**$_{\pm 0.1}$ | 79.3$_{\pm 0.1}$ / **80.0**$_{\pm 0.1}$ | 76.7$_{\pm 0.1}$ / **76.8**$_{\pm 0.1}$ | 79.3$_{\pm 0.1}$ / **80.1**$_{\pm 0.1}$ |
| Evo-S0 | 75.8$_{\pm 0.1}$ / 75.8$_{\pm 0.2}$ | 78.5$_{\pm 0.1}$ / 78.7$_{\pm 0.1}$ | 76.2$_{\pm 0.0}$ / 76.5$_{\pm 0.1}$ | 78.9$_{\pm 0.0}$ / 79.6$_{\pm 0.0}$ |
| GN+WS  | 74.2$_{\pm 0.1}$ / 74.1$_{\pm 0.1}$ | 77.8$_{\pm 0.0}$ / 77.8$_{\pm 0.1}$ | 76.2$_{\pm 0.1}$ / 76.3$_{\pm 0.1}$ | 79.2$_{\pm 0.1}$ / 79.4$_{\pm 0.1}$ |
| FRN+TLU| 75.7$_{\pm 0.1}$ / 75.7$_{\pm 0.2}$ | 78.4$_{\pm 0.1}$ / 78.9$_{\pm 0.1}$ | 74.9$_{\pm 0.2}$ / 75.1$_{\pm 0.1}$ | 78.2$_{\pm 0.1}$ / 78.6$_{\pm 0.1}$ |

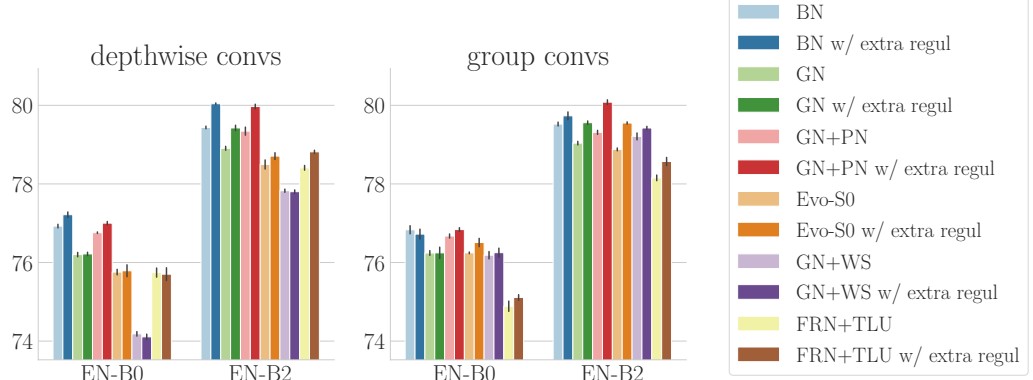

Figure 7: **BN vs. batch-independent approaches**. Validation accuracies (%) of EfficientNets trained on ImageNet with BN and various batch-independent approaches, without and with extra regularization.

### A.3.2  Training accuracies

In Tables 9, 10, 11, we complement the results of Tables 2, 3, 4 with training accuracies.

We stress that these training accuracies are highly dependent on the strength of applied regularization. This leads us to: (i) always separate the training accuracies obtained without and with extra regularization; (ii) report only the training accuracies obtained with batch-independent approaches, given that training accuracies obtained with BN would not be comparable due to BN's inherent regularization.

As visible in Tables 9, 10, 11, GN+PN outperforms alternative batch-independent approaches in terms of training accuracy on ImageNet. This applies both to training without extra regularization and to training with extra regularization. This suggests that, on larger datasets, GN+PN would outperform these alternative batch-independent approaches in terms of both training and validation accuracies [81, 72].

In Table 11, the fact that with extra regularization EfficientNets-B2 reach lower training accuracies than EfficientNets-B0 is explained by the different level of applied regularization (we add CutMix when training EfficientNets-B2).

Table 9: **Training accuracies in ResNet-50**. Networks are trained on ImageNet with LN, GN, GN+WS with $G$ groups, either without or with PN added on top (plain vs. +PN). Results are formatted as X with X the training accuracy at the final epoch (mean and $1\sigma$ interval in %). We report separately the results without extra regularization (top) and with extra regularization (bottom).

| | | | RN50 | |
| | | $G$ | plain | +PN |
|---|---|---|---|---|
| without extra regul | LN | 1 | $75.7_{\pm0.1}$ | $79.9_{\pm0.1}$ |
| | GN | 8 | $77.2_{\pm0.1}$ | $\mathbf{80.3}_{\pm0.1}$ |
| | GN | 32 | $77.0_{\pm0.0}$ | $79.2_{\pm0.2}$ |
| | GN+WS | 8 | $80.1_{\pm0.0}$ | $\mathbf{80.4}_{\pm0.0}$ |
| with extra regul | LN | 1 | $71.8_{\pm0.1}$ | $75.8_{\pm0.0}$ |
| | GN | 8 | $73.3_{\pm0.1}$ | $\mathbf{76.2}_{\pm0.1}$ |
| | GN | 32 | $73.1_{\pm0.1}$ | $75.1_{\pm0.1}$ |
| | GN+WS | 8 | $75.8_{\pm0.0}$ | $\mathbf{76.3}_{\pm0.0}$ |

Table 10: **Training accuracies in ResNets and ResNeXts**. Networks are trained on ImageNet with GN, GN+PN. Results are formatted as in Table 9.

| | | RN50 | RN101 | RNX50 | RNX101 |
|---|---|---|---|---|---|
| without extra regul | GN | $77.0_{\pm0.0}$ | $79.9_{\pm0.1}$ | $79.6_{\pm0.1}$ | $81.6_{\pm0.0}$ |
| | GN+PN | $\mathbf{80.3}_{\pm0.1}$ | $\mathbf{83.5}_{\pm0.0}$ | $\mathbf{84.1}_{\pm0.1}$ | $\mathbf{86.2}_{\pm0.0}$ |
| with extra regul | GN | $73.1_{\pm0.1}$ | $76.5_{\pm0.0}$ | $76.0_{\pm0.1}$ | $78.6_{\pm0.1}$ |
| | GN+PN | $\mathbf{76.2}_{\pm0.1}$ | $\mathbf{79.7}_{\pm0.1}$ | $\mathbf{79.8}_{\pm0.0}$ | $\mathbf{82.7}_{\pm0.0}$ |

Table 11: **Training accuracies in EfficientNets**. Networks are trained on ImageNet with various batch-independent approaches: GN, GN+PN, Evo-S0, GN+WS, FRN+TLU. Results are formatted as in Table 9.

| | | depthwise convs | | group convs | |
| | | EN-B0 | EN-B2 | EN-B0 | EN-B2 |
|---|---|---|---|---|---|
| without extra regul | GN | $75.4_{\pm0.0}$ | $80.9_{\pm0.1}$ | $74.7_{\pm0.0}$ | $80.1_{\pm0.1}$ |
| | GN+PN | $\mathbf{77.3}_{\pm0.0}$ | $\mathbf{82.7}_{\pm0.0}$ | $\mathbf{75.8}_{\pm0.0}$ | $\mathbf{81.4}_{\pm0.1}$ |
| | Evo-S0 | $74.6_{\pm0.2}$ | $79.8_{\pm0.2}$ | $75.1_{\pm0.0}$ | $80.4_{\pm0.1}$ |
| | GN+WS | $71.4_{\pm0.0}$ | $77.6_{\pm0.0}$ | $74.5_{\pm0.0}$ | $80.2_{\pm0.1}$ |
| | FRN+TLU | $75.0_{\pm0.1}$ | $80.4_{\pm0.0}$ | $72.9_{\pm0.1}$ | $78.5_{\pm0.1}$ |
| with extra regul | GN | $71.2_{\pm0.1}$ | $66.2_{\pm0.1}$ | $70.5_{\pm0.1}$ | $65.6_{\pm0.1}$ |
| | GN+PN | $\mathbf{72.8}_{\pm0.0}$ | $\mathbf{67.8}_{\pm0.1}$ | $\mathbf{71.5}_{\pm0.1}$ | $\mathbf{66.7}_{\pm0.0}$ |
| | Evo-S0 | $70.2_{\pm0.2}$ | $64.4_{\pm0.3}$ | $70.8_{\pm0.1}$ | $65.6_{\pm0.1}$ |
| | GN+WS | $67.3_{\pm0.1}$ | $63.4_{\pm0.1}$ | $70.4_{\pm0.1}$ | $65.4_{\pm0.0}$ |
| | FRN+TLU | $70.4_{\pm0.3}$ | $65.1_{\pm0.2}$ | $68.8_{\pm0.2}$ | $64.0_{\pm0.1}$ |

### A.3.3    Effect of omitting PN's additional parameters

In Tables 12, 13 and Figure 8, we report results with PN's additional parameters $\tilde{\beta}^l$, $\tilde{\gamma}^l$ set to 0. In that case, $\tilde{\beta}^l$, $\tilde{\gamma}^l$ can be equivalently omitted and the proxy variable $Y^l$ can be simply considered as a standard Gaussian variable in each channel $c$, i.e. $Y_c^l \sim \mathcal{N}(0, 1)$ (cf our implementation of Appendix B).

As visible in Tables 12, 13 and Figure 8, the omission of PN's additional parameters $\tilde{\beta}^l$, $\tilde{\gamma}^l$ is indeed harmful. However, the drop of performance that results from omitting $\tilde{\beta}^l$, $\tilde{\gamma}^l$ in GN+PN is very small (in average less than $0.1\%$ in validation accuracy).

Given that the omission of PN's additional parameters $\tilde{\beta}^l$, $\tilde{\gamma}^l$ leads to slight benefits in terms of computational requirements and simplicity of implementation, this variant of PN with $\tilde{\beta}^l$, $\tilde{\gamma}^l$ omitted might sometimes be a better trade-off.

Table 12: **Effect of omitting PN's additional parameters in ResNets and ResNeXts**. Networks are trained on ImageNet with BN and GN, GN+PN with $\tilde{\beta}^l$, $\tilde{\gamma}^l$ included, and GN+PN with $\tilde{\beta}^l$, $\tilde{\gamma}^l$ omitted. Results are formatted as in Table 6.

| | RN50 | RN101 | RNX50 | RNX101 |
|---|---|---|---|---|
| BN | **76.3**±0.1 / 75.8±0.2 | 77.9±0.1 / **78.0**±0.1 | **77.6**±0.1 / 77.2±0.1 | 78.7±0.1 / **78.9**±0.1 |
| GN | 75.4±0.1 / 75.3±0.1 | 77.0±0.1 / 77.4±0.1 | 76.2±0.2 / 76.6±0.1 | 77.4±0.2 / 78.1±0.1 |
| GN+PN with $\tilde{\beta}^l$, $\tilde{\gamma}^l$ included | 76.3±0.1 / **76.7**±0.0 | 77.6±0.2 / **78.6**±0.2 | 76.7±0.1 / **77.8**±0.2 | 77.7±0.2 / 79.0±0.1 |
| GN+PN with $\tilde{\beta}^l$, $\tilde{\gamma}^l$ omitted | 76.3±0.0 / 76.7±0.1 | 77.5±0.0 / 78.5±0.1 | 76.5±0.1 / 77.6±0.1 | 77.5±0.1 / **79.0**±0.1 |

Table 13: **Effect of omitting PN's additional parameters in EfficientNets**. Networks are trained on ImageNet with BN and GN, GN+PN with $\tilde{\beta}^l$, $\tilde{\gamma}^l$ included, and GN+PN with $\tilde{\beta}^l$, $\tilde{\gamma}^l$ omitted. Results are formatted as in Table 6.

| | depthwise convs | | group convs | |
|---|---|---|---|---|
| | EN-B0 | EN-B2 | EN-B0 | EN-B2 |
| BN | 76.9±0.1 / **77.2**±0.1 | 79.4±0.0 / **80.0**±0.0 | **76.8**±0.1 / 76.7±0.2 | 79.5±0.1 / **79.7**±0.1 |
| GN | 76.2±0.1 / 76.2±0.1 | 78.9±0.1 / 79.4±0.1 | 76.2±0.1 / 76.2±0.2 | 79.0±0.1 / 79.6±0.1 |
| GN+PN with $\tilde{\beta}^l$, $\tilde{\gamma}^l$ included | 76.8±0.0 / **77.0**±0.1 | 79.3±0.1 / **80.0**±0.1 | 76.7±0.1 / **76.8**±0.1 | 79.3±0.1 / **80.1**±0.1 |
| GN+PN with $\tilde{\beta}^l$, $\tilde{\gamma}^l$ omitted | 76.6±0.2 / 77.0±0.1 | 79.2±0.0 / 79.9±0.1 | 76.7±0.1 / 76.7±0.1 | 79.3±0.1 / 80.0±0.2 |

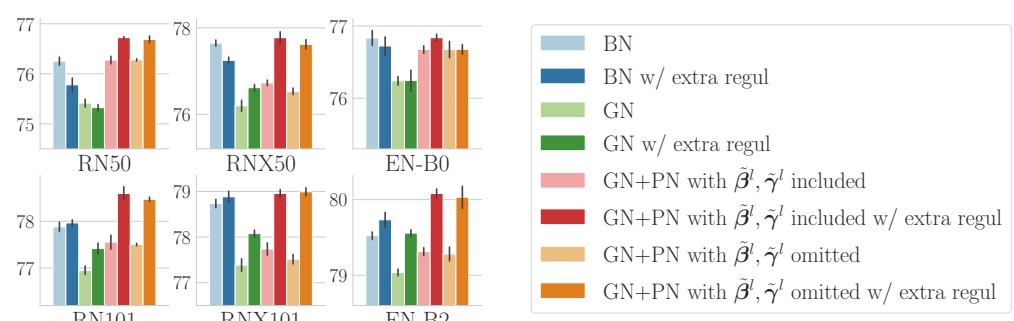

Figure 8: **Effect of omitting PN's additional parameters**. Validation accuracies (%) of ResNets, ResNeXts and EfficientNets trained on ImageNet with BN and GN, GN+PN with $\tilde{\beta}^l$, $\tilde{\gamma}^l$ included, and GN+PN with $\tilde{\beta}^l$, $\tilde{\gamma}^l$ omitted, without and with extra regularization. EfficientNets are considered in the variant with group convolutions [79].

### A.3.4 Effect of changing the choice of the extra regularization

In Table 14 and Figure 9, we report results in ResNets and ResNeXts with a change in the choice of the extra regularization. When using extra regularization, instead of using label smoothing [82], dropout [83] and stochastic depth [84], we use Mixup [85] in all networks, and in ResNet-101 and ResNeXt-101, we additionally use CutMix [86] (cf Appendix A.1).

We reach similar conclusions with the results of Table 14 and Figure 9 as with the results of Table 7: (i) BN is matched or outperformed by GN+PN, except for a small gap of performance in ResNeXt-50 (this gap of performance might be due to the imperfect "abstraction away" of regularization); (ii) good performance remains tied to the combination of both an efficient normalization and an efficient regularization.

Table 14: **Effect of changing the choice of the extra regularization**. ResNets and ResNeXts are trained on ImageNet with BN and GN, GN+PN. Results are formatted as X / Z with X, Z the validation accuracies (mean and $1\sigma$ interval in %) without extra regularization and with an extra regularization other than the one used in Table 7, respectively.

|  | RN50 | RN101 | RNX50 | RNX101 |
|---|---|---|---|---|
| BN | $76.3_{\pm0.1}$ / $\mathbf{76.3}_{\pm0.0}$ | $77.9_{\pm0.1}$ / $\mathbf{78.1}_{\pm0.1}$ | $77.6_{\pm0.1}$ / $\mathbf{78.0}_{\pm0.0}$ | $78.7_{\pm0.1}$ / $\mathbf{79.5}_{\pm0.0}$ |
| GN | $75.4_{\pm0.1}$ / $75.9_{\pm0.1}$ | $77.0_{\pm0.1}$ / $77.7_{\pm0.1}$ | $76.2_{\pm0.2}$ / $76.7_{\pm0.1}$ | $77.4_{\pm0.2}$ / $78.3_{\pm0.1}$ |
| GN+PN | $76.3_{\pm0.1}$ / $\mathbf{77.0}_{\pm0.0}$ | $77.6_{\pm0.2}$ / $\mathbf{78.9}_{\pm0.1}$ | $76.7_{\pm0.1}$ / $\mathbf{77.6}_{\pm0.1}$ | $77.7_{\pm0.2}$ / $\mathbf{79.6}_{\pm0.1}$ |

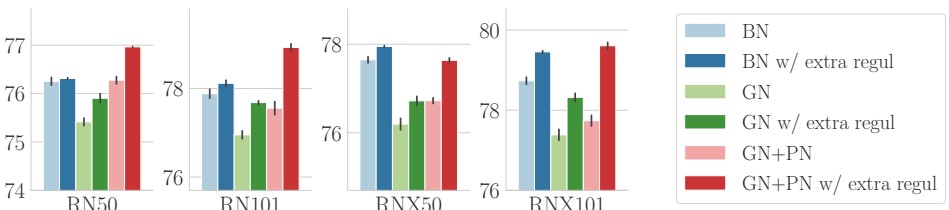

Figure 9: **Effect of changing the choice of the extra regularization**. Validation accuracies (%) of ResNets and ResNeXts trained on ImageNet with BN and GN, GN+PN, without extra regularization and with an extra regularization other than the one used in Table 7.

### A.4 Results on CIFAR-10 and CIFAR-100

In this section, we report results on CIFAR with different sizes of ResNets: ResNet-20 (RN20), ResNet-32 (RN32), ResNet-44 (RN44), ResNet-56 (RN56), ResNet-110 (RN110).

We report results on CIFAR-10 in Table 15, and results on CIFAR-100 in Table 16. We further provide a visualization of these results in Figure 10.

While slightly underperforming BN on CIFAR-10, GN+PN tends to slightly outperform BN on CIFAR-100. As a possible reason, BN's regularization could be more beneficial on the "easy" task of CIFAR-10 than on the "harder" task of CIFAR-100. To the extent that BN's regularization can be seen as a reduction of the network's effective capacity, such a reduction of the network's effective capacity could be more harmful for tasks that require more capacity, i.e. for harder tasks.

Table 15: **BN vs. GN, GN+PN on CIFAR-10**. ResNets are trained with BN and GN, GN+PN. Results are formatted as X / Y with X, Y the validation accuracies (mean and $1\sigma$ interval in %) without and with extra regularization, respectively.

|  | RN20 | RN32 | RN44 | RN56 | RN110 |
|---|---|---|---|---|---|
| BN | $91.6_{\pm0.3}$ / $\mathbf{91.8}_{\pm0.2}$ | $92.4_{\pm0.1}$ / $\mathbf{92.7}_{\pm0.2}$ | $92.7_{\pm0.2}$ / $\mathbf{93.1}_{\pm0.2}$ | $93.0_{\pm0.1}$ / $\mathbf{93.4}_{\pm0.2}$ | $93.5_{\pm0.1}$ / $\mathbf{93.7}_{\pm0.2}$ |
| GN | $90.8_{\pm0.2}$ / $90.7_{\pm0.1}$ | $91.5_{\pm0.2}$ / $91.5_{\pm0.1}$ | $91.8_{\pm0.2}$ / $92.0_{\pm0.1}$ | $92.2_{\pm0.2}$ / $92.2_{\pm0.2}$ | $92.6_{\pm0.2}$ / $92.9_{\pm0.3}$ |
| GN+PN | $91.4_{\pm0.2}$ / $\mathbf{91.6}_{\pm0.3}$ | $92.3_{\pm0.2}$ / $\mathbf{92.5}_{\pm0.2}$ | $92.8_{\pm0.2}$ / $\mathbf{92.9}_{\pm0.2}$ | $92.9_{\pm0.2}$ / $\mathbf{93.2}_{\pm0.2}$ | $93.2_{\pm0.1}$ / $\mathbf{93.6}_{\pm0.1}$ |

Table 16: **BN vs. GN, GN+PN on CIFAR-100**. ResNets are trained with BN and GN, GN+PN. Results are formatted as in Table 15.

|  | RN20 | RN32 | RN44 | RN56 | RN110 |
|---|---|---|---|---|---|
| BN | $\mathbf{66.8}_{\pm0.3}$ / $65.1_{\pm0.2}$ | $68.2_{\pm0.3}$ / $\mathbf{68.7}_{\pm0.2}$ | $69.2_{\pm0.4}$ / $\mathbf{70.5}_{\pm0.2}$ | $70.1_{\pm0.2}$ / $\mathbf{71.4}_{\pm0.3}$ | $71.7_{\pm0.3}$ / $\mathbf{73.3}_{\pm0.3}$ |
| GN | $65.0_{\pm0.3}$ / $61.7_{\pm0.3}$ | $66.5_{\pm0.4}$ / $65.3_{\pm0.4}$ | $67.3_{\pm0.6}$ / $67.0_{\pm0.3}$ | $67.8_{\pm0.4}$ / $68.1_{\pm0.5}$ | $69.5_{\pm0.3}$ / $70.2_{\pm0.4}$ |
| GN+PN | $66.3_{\pm0.4}$ / $\mathbf{66.7}_{\pm0.2}$ | $67.8_{\pm0.4}$ / $\mathbf{69.5}_{\pm0.2}$ | $68.9_{\pm0.3}$ / $\mathbf{70.8}_{\pm0.4}$ | $69.8_{\pm0.3}$ / $\mathbf{71.7}_{\pm0.4}$ | $71.4_{\pm0.2}$ / $\mathbf{73.1}_{\pm0.4}$ |

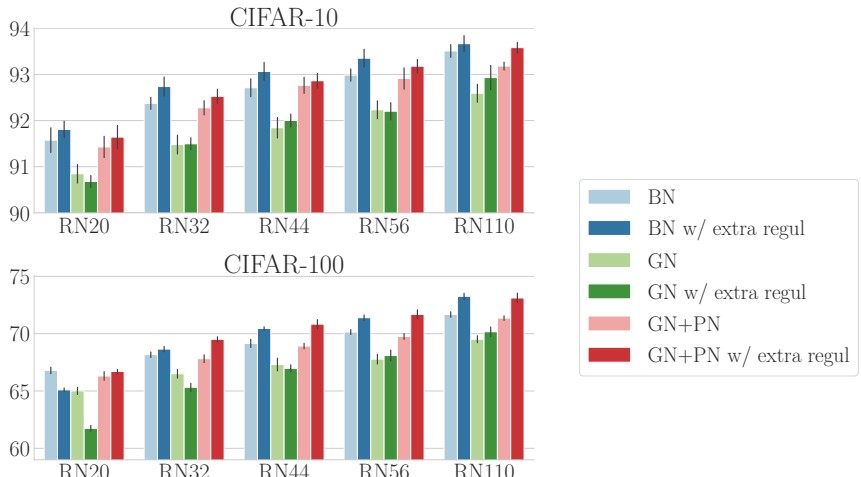

Figure 10: **BN vs. GN, GN+PN**. Validation accuracies (%) of ResNets trained on CIFAR-10 (top) and CIFAR-100 (bottom) with BN and GN, GN+PN, without and with extra regularization.

## B  Implementation of Proxy Norm

In this section, we provide a Tensorflow 1 implementation of the proxy-normalized activation step when PN's additional parameters $\tilde{\beta}^l$, $\tilde{\gamma}^l$ are set to zero, i.e. omitted (cf Section 5).

```python
import tensorflow as tf
import numpy as np
from scipy.special import erfinv
def uniformly_sampled_gaussian(num_rand):
    rand = 2 * (np.arange(num_rand) + 0.5) / float(num_rand) - 1
    return np.sqrt(2) * erfinv(rand)

def proxy_norm_act(y,
                   activation_fn=tf.nn.relu,
                   proxy_epsilon=0.03,
                   num_samples=256):
    """
    TensorFlow 1 implementation of the proxy normalized activation step.

    Following the same convention as in the main text of this paper,
    the affine transform is applied in this step rather than in the normalization step.

    :param y: 4D activation tensor after the normalization step
    :param activation_fn: activation function
    :param proxy_epsilon: PN's numerical stability constant (should not be too low)
    :param num_samples: number of samples in the proxy distribution
    :return tilde_z: 4D activation tensor after the proxy-normalized activation step
    """
    def create_channelwise_variable(name, init):
        num_channels = int(y.get_shape()[-1])
        return tf.get_variable(name,
                               dtype=y.dtype,
                               shape=[1, 1, 1, num_channels],
                               initializer=tf.constant_initializer(init))
    # shift and scale parameters after the norm
    beta = create_channelwise_variable('beta', 0.0)
    gamma = create_channelwise_variable('gamma', 1.0)
```

```
# activation step
z = gamma * y + beta  # affine transform
z = activation_fn(z)   # activation function

# proxy normalization
proxy_y = tf.constant(uniformly_sampled_gaussian(num_samples), y.dtype)
proxy_y = tf.reshape(proxy_y, [num_samples, 1, 1, 1])
proxy_z = gamma * proxy_y + beta  # affine transform on proxy distribution
proxy_z = activation_fn(proxy_z)   # activation function on proxy distribution
# compute proxy statistics in float32
proxy_mean, proxy_var = tf.nn.moments(
    tf.cast(proxy_z, tf.float32), axes=[0], keepdims=True)
proxy_mean = tf.cast(proxy_mean, y.dtype)
inv_proxy_std = tf.cast(tf.rsqrt(proxy_var + proxy_epsilon), y.dtype)
# normalize z according to proxy statistics
tilde_z = (z - proxy_mean) * inv_proxy_std
return tilde_z
```

# C  Proofs of results other than Theorems 1, 2, 3

## C.1  Layer-wise power equals one

**Proposition 1.** *If $\sigma_{I_{\mathbf{x},c}}(\mathbf{x}^l) \neq 0$ for all $\mathbf{x} \in \mathcal{D}$ and $c \in \{1, \dots, C_l\}$, then it holds that $\mathcal{P}(\mathbf{y}^l) = 1$ for any choice of* $\mathrm{Norm} \in \{\mathrm{BN}, \mathrm{LN}, \mathrm{IN}, \mathrm{GN}\}$.

**Proof.** The proof proceeds by distinguishing each case in $\mathrm{Norm} \in \{\mathrm{BN}, \mathrm{LN}, \mathrm{IN}, \mathrm{GN}\}$.

**Case of BN.**  If we fix a channel $c$, the assumption $\sigma_c(\mathbf{x}^l) \neq 0$ implies that

$$\mathcal{P}_c(\mathbf{y}^l) = \frac{\mathbb{E}_{\mathbf{x},\alpha}\left[\left(\mathbf{x}^l_{\alpha,c} - \mu_c(\mathbf{x}^l)\right)^2\right]}{\sigma_c(\mathbf{x}^l)^2} = \frac{\sigma_c(\mathbf{x}^l)^2}{\sigma_c(\mathbf{x}^l)^2} = 1.$$

We immediately get $\mathcal{P}(\mathbf{y}^l) = \mathbb{E}_c\left[\mathcal{P}_c(\mathbf{y}^l)\right] = 1$.

**Case of GN.**  Let us fix $\mathbf{x} \in \mathcal{D}$ and let us denote $\mathcal{G}_g$ for $g \in \{1, \dots, G\}$ the $G$ groups of channels and $I_{\mathbf{x}}^{(g)} = \{\mathbf{x}, c \in \mathcal{G}_g\}$ for $g \in \{1, \dots, G\}$ the $G$ conditional sets of standardization.

The assumption $\sigma_{I_{\mathbf{x}}^{(g)}}(\mathbf{x}^l) \neq 0$ implies for any $g$ that

$$\mathcal{P}_{I_{\mathbf{x}}^{(g)}}(\mathbf{y}^l) = \frac{\mathbb{E}_{\alpha,c|c\in\mathcal{G}_g}\left[\left(\mathbf{x}^l_{\alpha,c} - \mu_{I_{\mathbf{x}}^{(g)}}(\mathbf{x}^l)\right)^2\right]}{\sigma_{I_{\mathbf{x}}^{(g)}}(\mathbf{x}^l)^2} = \frac{\sigma_{I_{\mathbf{x}}^{(g)}}(\mathbf{x}^l)^2}{\sigma_{I_{\mathbf{x}}^{(g)}}(\mathbf{x}^l)^2} = 1.$$

This implies

$$\mathcal{P}_{\mathbf{x}}(\mathbf{y}^l) = \frac{1}{C_l}\sum_c \mathcal{P}_{\mathbf{x},c}(\mathbf{y}^l) = \frac{1}{C_l}\sum_g \sum_{c\in\mathcal{G}_g} \mathcal{P}_{\mathbf{x},c}(\mathbf{y}^l)$$

$$= \frac{1}{C_l}\sum_g |\mathcal{G}_g|\mathcal{P}_{I_{\mathbf{x}}^{(g)}}(\mathbf{y}^l) = \frac{1}{C_l}\sum_g |\mathcal{G}_g| = 1,$$

where we used $\mathcal{P}_{I_{\mathbf{x}}^{(g)}}(\mathbf{y}^l) = \frac{1}{|\mathcal{G}_g|}\sum_{c\in\mathcal{G}_g} \mathcal{P}_{\mathbf{x},c}(\mathbf{y}^l)$.

We immediately get $\mathcal{P}(\mathbf{y}^l) = \mathbb{E}_{\mathbf{x}}\left[\mathcal{P}_{\mathbf{x}}(\mathbf{y}^l)\right] = 1$.

**Cases of LN and IN.**  The cases of LN and IN immediately follow from the cases of GN with $G = 1$ group and $G = C_l$ groups. □

## C.2 Channel-wise collapse implies channel-wise linearity

Some additional notations are required in this section. We denote $\Theta^l \equiv (\boldsymbol{\omega}^1, \boldsymbol{\beta}^1, \boldsymbol{\gamma}^1, \ldots, \boldsymbol{\omega}^l, \boldsymbol{\beta}^l, \boldsymbol{\gamma}^l)$ the aggregated model parameters up to layer $l$.

We further define the linearized post-activations $\widetilde{\mathbf{z}}^l$ as

$$\forall \alpha, c: \quad \widetilde{\mathbf{z}}^l_{\alpha,c} = \widetilde{\lambda}_c \tilde{\mathbf{y}}^l_{\alpha,c}, \tag{8}$$

$$\forall c: \quad \widetilde{\lambda}_c = \arg\min_{\lambda_c} \mathbb{E}_{\mathbf{x},\alpha}\left[\left(\mathbf{z}^l_{\alpha,c} - \lambda_c \tilde{\mathbf{y}}^l_{\alpha,c}\right)^2\right] = \arg\min_{\lambda_c} \mathbb{E}_{\mathbf{x},\alpha}\left[\left(\phi(\tilde{\mathbf{y}}^l_{\alpha,c}) - \lambda_c \tilde{\mathbf{y}}^l_{\alpha,c}\right)^2\right]. \tag{9}$$

The linearized post-activations $\widetilde{\mathbf{z}}^l$ are the channel-wise linear best-fit of $\mathbf{z}^l = \phi(\tilde{\mathbf{y}}^l)$ using $\tilde{\mathbf{y}}^l$.

We start by proving that the inequality $\mathcal{P}_c(\tilde{\mathbf{y}}^l) - \mathcal{P}_c^{(1)}(\tilde{\mathbf{y}}^l) \leq \tilde{\eta}\mathcal{P}_c^{(1)}(\tilde{\mathbf{y}}^l)$ for sufficiently small $\tilde{\eta}$ implies channel-wise linearity (Proposition 2). We then prove that the inequality $\mathcal{P}_c(\tilde{\mathbf{y}}^l) - \mathcal{P}_c^{(1)}(\tilde{\mathbf{y}}^l) \leq \widetilde{\eta}\mathcal{P}_c(\tilde{\mathbf{y}}^l)$ for sufficiently small $\widetilde{\eta}$ implies channel-wise-linearity (Proposition 3).

**Proposition 2.** *If we fix some $d \in \mathbb{N}^*$, there exists $\tilde{\eta} > 0$ such that for any choice of $(\phi, H, W, \mathcal{D}, \Theta^l)$, it holds that*

$$\left(HW|\mathcal{D}| = d\right) \wedge \left(\mathcal{P}_c(\tilde{\mathbf{y}}^l) - \mathcal{P}_c^{(1)}(\tilde{\mathbf{y}}^l) \leq \tilde{\eta}\mathcal{P}_c^{(1)}(\tilde{\mathbf{y}}^l)\right) \implies \mathbf{z}^l_{\alpha,c} = \widetilde{\mathbf{z}}^l_{\alpha,c} \,\forall \mathbf{x}, \alpha,$$

*where $\widetilde{\mathbf{z}}^l$ are the linearized post-activations defined in Eq. (8), (9) and $\wedge$ is the logical "and".*

**Proof.** Any positive homogeneous $\phi$ satisfies $\phi(r) = r\phi(1)$ and $\phi(-r) = r\phi(-1)$ for any $r \geq 0$. This means that any positive homogeneous $\phi$ is: (i) fully determined by its values at $+1$ and $-1$; (ii) linear on the intervals $(-\infty, 0]$ and $[0, +\infty)$.

A sufficient condition for the linearity with respect to $\mathbf{x}, \alpha$ in channel $c$ is therefore a constant sign of $\tilde{\mathbf{y}}^l_{\alpha,c}$ for all $\mathbf{x}, \alpha$. Let us see that this constant sign is implied by a sufficiently severe channel-wise collapse.

We start by proving the result with the two distinct conditionalities: (i) $\sigma_c(\tilde{\mathbf{y}}^l) = 0$ and (ii) $\sigma_c(\tilde{\mathbf{y}}^l) > 0$.

**Conditionality $\sigma_c(\tilde{\mathbf{y}}^l) = 0$.** If $\sigma_c(\tilde{\mathbf{y}}^l) = 0$, then $\tilde{\mathbf{y}}^l_{\alpha,c} = \mu_c(\tilde{\mathbf{y}}^l), \forall \mathbf{x}, \alpha$.

Let us then define $\lambda_c$ such that $\lambda_c = 0$ if $\mu_c(\tilde{\mathbf{y}}^l) = 0$, and $\lambda_c = \frac{\phi(\mu_c(\tilde{\mathbf{y}}^l))}{\mu_c(\tilde{\mathbf{y}}^l)}$ otherwise.

For any choice of positive homogeneous $\phi$, it holds that $\phi(0) = 0$. Combined with the definition of $\lambda_c$, this implies $\phi(\mu_c(\tilde{\mathbf{y}}^l)) = \lambda_c \mu_c(\tilde{\mathbf{y}}^l)$ and thus $\forall \mathbf{x}, \alpha$:

$$\mathbf{z}^l_{\alpha,c} = \phi(\tilde{\mathbf{y}}^l_{\alpha,c}) = \phi(\mu_c(\tilde{\mathbf{y}}^l)) = \lambda_c \mu_c(\tilde{\mathbf{y}}^l) = \lambda_c \tilde{\mathbf{y}}^l_{\alpha,c}.$$

Given the definition of the linearized post-activations $\widetilde{\mathbf{z}}^l$, this means

$$\mathbb{E}_{\mathbf{x},\alpha}\left[\left(\mathbf{z}^l_{\alpha,c} - \widetilde{\mathbf{z}}^l_{\alpha,c}\right)^2\right] \leq \mathbb{E}_{\mathbf{x},\alpha}\left[\left(\mathbf{z}^l_{\alpha,c} - \lambda_c \tilde{\mathbf{y}}^l_{\alpha,c}\right)^2\right] = 0.$$

This immediately implies $\forall \mathbf{x}, \alpha$: $\mathbf{z}^l_{\alpha,c} = \widetilde{\mathbf{z}}^l_{\alpha,c}$. Thus, for any $(\phi, H, W, \mathcal{D}, \Theta^l)$ such that $\sigma_c(\tilde{\mathbf{y}}^l) = 0$, it holds that $\forall \mathbf{x}, \alpha$: $\mathbf{z}^l_{\alpha,c} = \widetilde{\mathbf{z}}^l_{\alpha,c}$.

More concisely, it holds for any choice of $(\phi, H, W, \mathcal{D}, \Theta^l)$ that

$$\sigma_c(\tilde{\mathbf{y}}^l) = 0 \implies \mathbf{z}^l_{\alpha,c} = \widetilde{\mathbf{z}}^l_{\alpha,c} \,\forall \mathbf{x}, \alpha. \tag{10}$$

**Conditionality $\sigma_c(\tilde{\mathbf{y}}^l) > 0$.** We start by fixing $(\phi, H, W, \mathcal{D}, \Theta^l)$. For any given $k > 0$, Chebyshev's inequality implies

$$\mathbb{P}_{\mathbf{x},\alpha}\left[|\tilde{\mathbf{y}}^l_{\alpha,c} - \mu_c(\tilde{\mathbf{y}}^l)| \geq k\sigma_c(\tilde{\mathbf{y}}^l)\right] \leq \frac{1}{k^2},$$

$$\mathbb{P}_{\mathbf{x},\alpha}\left[\left(\tilde{\mathbf{y}}^l_{\alpha,c} - \mu_c(\tilde{\mathbf{y}}^l)\right)^2 \geq k^2\sigma_c(\tilde{\mathbf{y}}^l)^2\right] \leq \frac{1}{k^2}.$$

Thus, if $\mathcal{P}_c(\tilde{\mathbf{y}}^l) - \mathcal{P}_c^{(1)}(\tilde{\mathbf{y}}^l) \leq \eta \mathcal{P}_c^{(1)}(\tilde{\mathbf{y}}^l)$ for some $\eta > 0$, it holds for any given $k > 0$ that

$$1 - \frac{1}{k^2} \leq \mathbb{P}_{\mathbf{x},\alpha}\left[\left(\tilde{\mathbf{y}}_{\alpha,c}^l - \mu_c(\tilde{\mathbf{y}}^l)\right)^2 < k^2 \sigma_c(\tilde{\mathbf{y}}^l)^2\right] = \mathbb{P}_{\mathbf{x},\alpha}\left[\left(\tilde{\mathbf{y}}_{\alpha,c}^l - \mu_c(\tilde{\mathbf{y}}^l)\right)^2 < k^2\left(\mathcal{P}_c(\tilde{\mathbf{y}}^l) - \mathcal{P}_c^{(1)}(\tilde{\mathbf{y}}^l)\right)\right]$$

$$\leq \mathbb{P}_{\mathbf{x},\alpha}\left[\left(\tilde{\mathbf{y}}_{\alpha,c}^l - \mu_c(\tilde{\mathbf{y}}^l)\right)^2 < k^2 \eta \mathcal{P}_c^{(1)}(\tilde{\mathbf{y}}^l)\right] = \mathbb{P}_{\mathbf{x},\alpha}\left[\left(\tilde{\mathbf{y}}_{\alpha,c}^l - \mu_c(\tilde{\mathbf{y}}^l)\right)^2 < k^2 \eta \mu_c(\tilde{\mathbf{y}}^l)^2\right]$$

$$\leq \mathbb{P}_{\mathbf{x},\alpha}\left[|\tilde{\mathbf{y}}_{\alpha,c}^l - \mu_c(\tilde{\mathbf{y}}^l)| < k\sqrt{\eta}|\mu_c(\tilde{\mathbf{y}}^l)|\right].$$

Choosing $k = \frac{1}{\sqrt{\eta}}$, we get

$$\mathbb{P}_{\mathbf{x},\alpha}\left[|\tilde{\mathbf{y}}_{\alpha,c}^l - \mu_c(\tilde{\mathbf{y}}^l)| \geq |\mu_c(\tilde{\mathbf{y}}^l)|\right] \leq \eta.$$

Now if we suppose that $\eta$ is such that $\frac{1}{HW|\mathcal{D}|} > \eta > 0$, we get

$$\mathbb{P}_{\mathbf{x},\alpha}\left[|\tilde{\mathbf{y}}_{\alpha,c}^l - \mu_c(\tilde{\mathbf{y}}^l)| \geq |\mu_c(\tilde{\mathbf{y}}^l)|\right] < \frac{1}{HW|\mathcal{D}|}. \tag{11}$$

Eq. (11) could not hold if there existed $\mathbf{x} \in \mathcal{D}$ and $\alpha \in \{1,\ldots,H\} \times \{1,\ldots,W\}$ such that $|\tilde{\mathbf{y}}_{\alpha,c}^l - \mu_c(\tilde{\mathbf{y}}^l)| \geq |\mu_c(\tilde{\mathbf{y}}^l)|$. Consequently, $|\tilde{\mathbf{y}}_{\alpha,c}^l - \mu_c(\tilde{\mathbf{y}}^l)| < |\mu_c(\tilde{\mathbf{y}}^l)|$ for all $\mathbf{x}, \alpha$, implying that there exists a tensor $\mathbf{r}^l \in \mathbb{R}^{H \times W \times C_l}$ that implicitly depends on $\mathbf{x}$ such that $\forall \mathbf{x}, \alpha$:

$$\tilde{\mathbf{y}}_{\alpha,c}^l = \mathbf{r}_{\alpha,c}^l \mu_c(\tilde{\mathbf{y}}^l), \qquad \mathbf{r}_{\alpha,c}^l \geq 0. \tag{12}$$

Now if we combine $\sigma_c(\tilde{\mathbf{y}}^l) > 0$ with $\mathcal{P}_c(\tilde{\mathbf{y}}^l) - \mathcal{P}_c^{(1)}(\tilde{\mathbf{y}}^l) \leq \eta \mathcal{P}_c^{(1)}(\tilde{\mathbf{y}}^l)$, we deduce that $\mathcal{P}_c^{(1)}(\tilde{\mathbf{y}}^l) > 0$ and thus that $\mu_c(\tilde{\mathbf{y}}^l) \neq 0$. Combining this with Eq. (12), we get $\forall \mathbf{x}, \alpha$:

$$\mathbf{z}_{\alpha,c}^l = \phi(\tilde{\mathbf{y}}_{\alpha,c}^l) = \mathbf{r}_{\alpha,c}^l \phi(\mu_c(\tilde{\mathbf{y}}^l)) = \frac{\phi(\mu_c(\tilde{\mathbf{y}}^l))}{\mu_c(\tilde{\mathbf{y}}^l)} \mathbf{r}_{\alpha,c}^l \mu_c(\tilde{\mathbf{y}}^l) = \lambda_c \tilde{\mathbf{y}}_{\alpha,c}^l,$$

where we defined $\lambda_c \equiv \frac{\phi(\mu_c(\tilde{\mathbf{y}}^l))}{\mu_c(\tilde{\mathbf{y}}^l)}$.

Given the definition of the linearized post-activations $\tilde{\mathbf{z}}^l$, this means

$$\mathbb{E}_{\mathbf{x},\alpha}\left[\left(\mathbf{z}_{\alpha,c}^l - \tilde{\mathbf{z}}_{\alpha,c}^l\right)^2\right] \leq \mathbb{E}_{\mathbf{x},\alpha}\left[\left(\mathbf{z}_{\alpha,c}^l - \lambda_c \tilde{\mathbf{y}}_{\alpha,c}^l\right)^2\right] = 0.$$

This immediately implies $\forall \mathbf{x}, \alpha$: $\mathbf{z}_{\alpha,c}^l = \tilde{\mathbf{z}}_{\alpha,c}^l$.

Thus, if we fix some $d \in \mathbb{N}^*$ and if we define $\tilde{\eta} = \frac{1}{2d}$, it holds for any choice of $(\phi, H, W, \mathcal{D}, \Theta^l)$ such that (i) $HW|\mathcal{D}| = d$, (ii) $\mathcal{P}_c(\tilde{\mathbf{y}}^l) - \mathcal{P}_c^{(1)}(\tilde{\mathbf{y}}^l) \leq \tilde{\eta} \mathcal{P}_c^{(1)}(\tilde{\mathbf{y}}^l)$, (iii) $\sigma_c(\tilde{\mathbf{y}}^l) > 0$, that $\forall \mathbf{x}, \alpha$: $\mathbf{z}_{\alpha,c}^l = \tilde{\mathbf{z}}_{\alpha,c}^l$.

More concisely, it holds for any choice of $(\phi, H, W, \mathcal{D}, \Theta^l)$ that

$$\left(HW|\mathcal{D}| = d\right) \wedge \left(\mathcal{P}_c(\tilde{\mathbf{y}}^l) - \mathcal{P}_c^{(1)}(\tilde{\mathbf{y}}^l) \leq \tilde{\eta} \mathcal{P}_c^{(1)}(\tilde{\mathbf{y}}^l)\right) \wedge \left(\sigma_c(\tilde{\mathbf{y}}^l) > 0\right) \implies \mathbf{z}_{\alpha,c}^l = \tilde{\mathbf{z}}_{\alpha,c}^l \, \forall \mathbf{x}, \alpha. \tag{13}$$

**General case.** To wrap up, if we fix some $d \in \mathbb{N}^*$ and if we reuse the definition $\tilde{\eta} = \frac{1}{2d}$, it holds for any choice of $(\phi, H, W, \mathcal{D}, \Theta^l)$ that

$$\left(HW|\mathcal{D}| = d\right) \wedge \left(\mathcal{P}_c(\tilde{\mathbf{y}}^l) - \mathcal{P}_c^{(1)}(\tilde{\mathbf{y}}^l) \leq \tilde{\eta} \mathcal{P}_c^{(1)}(\tilde{\mathbf{y}}^l)\right)$$

$$\implies \left(\left(HW|\mathcal{D}| = d\right) \wedge \left(\mathcal{P}_c(\tilde{\mathbf{y}}^l) - \mathcal{P}_c^{(1)}(\tilde{\mathbf{y}}^l) \leq \tilde{\eta} \mathcal{P}_c^{(1)}(\tilde{\mathbf{y}}^l)\right) \wedge \left(\sigma_c(\tilde{\mathbf{y}}^l) = 0\right)\right)$$

$$\vee \left(\left(HW|\mathcal{D}| = d\right) \wedge \left(\mathcal{P}_c(\tilde{\mathbf{y}}^l) - \mathcal{P}_c^{(1)}(\tilde{\mathbf{y}}^l) \leq \tilde{\eta} \mathcal{P}_c^{(1)}(\tilde{\mathbf{y}}^l)\right) \wedge \left(\sigma_c(\tilde{\mathbf{y}}^l) > 0\right)\right)$$

$$\implies \left(\mathbf{z}_{\alpha,c}^l = \tilde{\mathbf{z}}_{\alpha,c}^l \, \forall \mathbf{x}, \alpha\right) \vee \left(\mathbf{z}_{\alpha,c}^l = \tilde{\mathbf{z}}_{\alpha,c}^l \, \forall \mathbf{x}, \alpha\right) \tag{14}$$

$$\implies \left(\mathbf{z}_{\alpha,c}^l = \tilde{\mathbf{z}}_{\alpha,c}^l \, \forall \mathbf{x}, \alpha\right),$$

where Eq. (14) is obtained using Eq. (10) and Eq. (13) and $\wedge$, $\vee$ are the logical "and" and "or". $\qquad\square$

**Proposition 3.** *If we fix some $d \in \mathbb{N}^*$, there exists $\widetilde{\eta} > 0$ such that for any choice of $(\phi, H, W, \mathcal{D}, \Theta^l)$, it holds that*

$$\left(HW|\mathcal{D}| = d\right) \wedge \left(\mathcal{P}_c(\tilde{\mathbf{y}}^l) - \mathcal{P}_c^{(1)}(\tilde{\mathbf{y}}^l) \leq \widetilde{\eta}\mathcal{P}_c(\tilde{\mathbf{y}}^l)\right) \implies \mathbf{z}_{\alpha,c}^l = \widetilde{\mathbf{z}}_{\alpha,c}^l \; \forall \mathbf{x}, \alpha,$$

*where $\widetilde{\mathbf{z}}^l$ are the linearized post-activations defined in Eq. (8), (9) and $\wedge$ is the logical "and".*

**Proof.** We start by noting that for any $1 > \eta > 0$:

$$\mathcal{P}_c(\tilde{\mathbf{y}}^l) - \mathcal{P}_c^{(1)}(\tilde{\mathbf{y}}^l) \leq \eta\mathcal{P}_c(\tilde{\mathbf{y}}^l) \iff \mathcal{P}_c(\tilde{\mathbf{y}}^l) - \mathcal{P}_c^{(1)}(\tilde{\mathbf{y}}^l) \leq \eta\left(\mathcal{P}_c(\tilde{\mathbf{y}}^l) - \mathcal{P}_c^{(1)}(\tilde{\mathbf{y}}^l) + \mathcal{P}_c^{(1)}(\tilde{\mathbf{y}}^l)\right)$$

$$\iff \mathcal{P}_c(\tilde{\mathbf{y}}^l) - \mathcal{P}_c^{(1)}(\tilde{\mathbf{y}}^l) \leq \frac{\eta}{1-\eta}\mathcal{P}_c^{(1)}(\tilde{\mathbf{y}}^l).$$

Thus, if we fix some $d \in \mathbb{N}^*$ and if we define $\tilde{\eta} = \frac{1}{2d}$ and $\widetilde{\eta} = \frac{\tilde{\eta}}{1+\tilde{\eta}}$, it holds for any choice of $(\phi, H, W, \mathcal{D}, \Theta^l)$ that

$$\left(HW|\mathcal{D}| = d\right) \wedge \left(\mathcal{P}_c(\tilde{\mathbf{y}}^l) - \mathcal{P}_c^{(1)}(\tilde{\mathbf{y}}^l) \leq \widetilde{\eta}\mathcal{P}_c(\tilde{\mathbf{y}}^l)\right)$$

$$\iff \left(HW|\mathcal{D}| = d\right) \wedge \left(\mathcal{P}_c(\tilde{\mathbf{y}}^l) - \mathcal{P}_c^{(1)}(\tilde{\mathbf{y}}^l) \leq \frac{\widetilde{\eta}}{1-\widetilde{\eta}}\mathcal{P}_c^{(1)}(\tilde{\mathbf{y}}^l)\right)$$

$$\iff \left(HW|\mathcal{D}| = d\right) \wedge \left(\mathcal{P}_c(\tilde{\mathbf{y}}^l) - \mathcal{P}_c^{(1)}(\tilde{\mathbf{y}}^l) \leq \tilde{\eta}\mathcal{P}_c^{(1)}(\tilde{\mathbf{y}}^l)\right)$$

$$\implies \left(\mathbf{z}_{\alpha,c}^l = \widetilde{\mathbf{z}}_{\alpha,c}^l \; \forall \mathbf{x}, \alpha\right), \tag{15}$$

where Eq. (15) is obtained using Proposition 2 and $\wedge$ is the logical "and". $\qquad\square$

### C.3 Alteration of expressivity with IN

In this section, we first prove that, for any dataset $\mathcal{D}$, networks without normalization can express mappings arbitrarily close to the identity (Proposition 4). We then prove that, in general, networks with IN cannot express mappings arbitrarily close to the identity (Proposition 5).

**Proposition 4.** *Lift any assumptions on $\phi$ and suppose instead that $\phi$ is non-polynomial. Further suppose that each layer $l$ up to depth $L$ implements the following two steps $\forall \alpha, c$:*

$$\mathbf{y}_{\alpha,c}^l = (\boldsymbol{\omega}^l * \mathbf{z}^{l-1})_{\alpha,c} + \mathbf{b}_c^l, \tag{16}$$

$$\mathbf{z}_{\alpha,c}^l = \phi\left(\mathbf{y}_{\alpha,c}^l\right), \tag{17}$$

*where $\mathbf{z}^0 \equiv \mathbf{x}$, and $\boldsymbol{\omega}^l \in \mathbb{R}^{K_l \times K_l \times C_{l-1} \times C_l}$ and $\mathbf{b}^l \in \mathbb{R}^{C_l}$ are the weights and biases at layer $l$.*

*Now fix a layer $l \in \{1, \ldots, L\}$, the spatial extents $H$, $W$, the widths $C_0$, $C_l$ assumed equal at layer $0$ and layer $l$, and the dataset $\mathcal{D}$. Further denote $\Phi_l$ the network mapping from $\mathbf{x}$ to $\mathbf{y}^l$ such that $\mathbf{y}^l = \Phi_l(\mathbf{x})$.*

*Then for any $\epsilon > 0$, there exists a choice of intermediate widths $(C_k)_{1 \leq k < l}$ and model parameters $(\boldsymbol{\omega}^1, \mathbf{b}^1, \ldots, \boldsymbol{\omega}^l, \mathbf{b}^l)$ such that*

$$\max_{\mathbf{x} \in \mathcal{D}} ||\Phi_l(\mathbf{x}) - \mathbf{x}|| \leq \epsilon.$$

**Proof.** The proof proceeds in multiple steps of increasing generality.

**Case of unit spatial extent of activations and kernels.** When $H$, $W$ are equal to 1 and $K_k$ is equal to one at every layer $k$, the propagation of Eq. (16), (17) becomes strictly equivalent to the propagation in a fully-connected network.

If $l$ is the first layer in the network ($l = 1$), we may obtain the strict equality $\Phi_l = $ identity by choosing the reshaped matricial version $\boldsymbol{W}^l \in \mathbb{R}^{C_0 \times C_l}$ of $\boldsymbol{\omega}^l$ as the identity and $\mathbf{b}^l$ as zero.

Otherwise ($l \geq 2$), we may apply the universal approximation theorem [91]. Given the assumption of non-polynomial activation function $\phi$, this means that for any $\epsilon > 0$, there exists a choice of intermediate widths $(C_k)_{1 \leq k < l}$ and model parameters $(\boldsymbol{\omega}^1, \mathbf{b}^1, \ldots, \boldsymbol{\omega}^l, \mathbf{b}^l)$ such that

$$\max_{\mathbf{x} \in \mathcal{D}} ||\Phi_l(\mathbf{x}) - \mathbf{x}|| \leq \epsilon. \tag{18}$$

**Case of unit spatial extent of kernels.** When $K_k$ is equal to one at every layer $k$, the propagation of Eq. (16), (17) "occurs" strictly independently for each spatial position $\alpha$.

Let us then consider a neural network that takes an input $\bar{\mathbf{x}} \in \mathbb{R}^{1 \times 1 \times C_0}$ and provides $\bar{\mathbf{y}}^l$, $\bar{\mathbf{z}}^l$ at every layer $l$ by implementing the same steps as Eq. (16), (17). Let us denote $\Psi_l$ the network mapping from $\bar{\mathbf{x}}$ to $\bar{\mathbf{y}}^l$ such that $\bar{\mathbf{y}}^l = \Psi_l(\bar{\mathbf{x}})$. And let us further denote $\overline{\mathcal{D}} = \{\bar{\mathbf{x}}^{(\mathbf{x},\alpha)}\}_{\mathbf{x} \in \mathcal{D}, \alpha \in \{1,\ldots,H\} \times \{1,\ldots,W\}}$, where $\bar{\mathbf{x}}^{(\mathbf{x},\alpha)} \in \mathbb{R}^{1 \times 1 \times C_0}$ denotes the reshaped version of $\mathbf{x}_{\alpha,:} \in \mathbb{R}^{C_0}$ for any $\mathbf{x}$, $\alpha$.

If we fix any $\epsilon > 0$ and if we apply Eq. (18) from the previous case with $\frac{1}{\sqrt{HW}}\epsilon$, we get that there exists a choice of intermediate widths $(C_k)_{1 \leq k < l}$ and model parameters $(\boldsymbol{\omega}^1, \mathbf{b}^1, \ldots, \boldsymbol{\omega}^l, \mathbf{b}^l)$ such that

$$\max_{\bar{\mathbf{x}} \in \overline{\mathcal{D}}} ||\Psi_l(\bar{\mathbf{x}}) - \bar{\mathbf{x}}|| = \max_{\mathbf{x} \in \mathcal{D}, \alpha \in \{1,\ldots,H\} \times \{1,\ldots,W\}} ||\Psi_l(\bar{\mathbf{x}}^{(\mathbf{x},\alpha)}) - \bar{\mathbf{x}}^{(\mathbf{x},\alpha)}|| \leq \frac{1}{\sqrt{HW}}\epsilon. \tag{19}$$

Let us then fix $(C_k)_{1 \leq k < l}$ and $(\boldsymbol{\omega}^1, \mathbf{b}^1, \ldots, \boldsymbol{\omega}^l, \mathbf{b}^l)$ such that Eq. (19) holds.

Due to the independence of spatial positions, the mapping $\Phi_l$ is such that $\Phi_l(\mathbf{x})_{\alpha,:}$ is a reshaped version of $\Psi_l(\bar{\mathbf{x}}^{(\mathbf{x},\alpha)})$ for any $\mathbf{x}$, $\alpha$. This means that $\forall \mathbf{x} \in \mathcal{D}$ and $\forall \alpha \in \{1, \ldots, H\} \times \{1, \ldots, W\}$:

$$||\Phi_l(\mathbf{x})_{\alpha,:} - \mathbf{x}_{\alpha,:}|| = ||\Psi_l(\bar{\mathbf{x}}^{(\mathbf{x},\alpha)}) - \bar{\mathbf{x}}^{(\mathbf{x},\alpha)}|| \leq \frac{1}{\sqrt{HW}}\epsilon,$$

$$||\Phi_l(\mathbf{x}) - \mathbf{x}||^2 = \sum_{\alpha \in \{1,\ldots,H\} \times \{1,\ldots,W\}} ||\Phi_l(\mathbf{x})_{\alpha,:} - \mathbf{x}_{\alpha,:}||^2 \leq \sum_{\alpha \in \{1,\ldots,H\} \times \{1,\ldots,W\}} \frac{1}{HW}\epsilon^2 \leq \epsilon^2.$$

This immediately implies

$$\max_{\mathbf{x} \in \mathcal{D}} ||\Phi_l(\mathbf{x}) - \mathbf{x}|| \leq \epsilon.$$

**General case.** Let us consider the neural network that takes $\mathbf{x} \in \mathcal{D}$ as input and provides $\bar{\mathbf{y}}^l$, $\bar{\mathbf{z}}^l$ at every layer $l$ by implementing Eq. (16), (17) with weights $\bar{\boldsymbol{\omega}}^l \in \mathbb{R}^{1 \times 1 \times C_{l-1} \times C_l}$, biases $\bar{\mathbf{b}}^l \in \mathbb{R}^{C_l}$ and activation function $\phi$. Let us then denote $\Psi_l$ the network mapping from $\mathbf{x}$ to $\bar{\mathbf{y}}^l$ such that $\bar{\mathbf{y}}^l = \Psi_l(\mathbf{x})$.

If we fix any $\epsilon > 0$, we get from the previous case that there exists a choice of intermediate widths $(C_k)_{1 \leq k < l}$ and model parameters $(\bar{\boldsymbol{\omega}}^1, \bar{\mathbf{b}}^1, \ldots, \bar{\boldsymbol{\omega}}^l, \bar{\mathbf{b}}^l)$ such that

$$\max_{\mathbf{x} \in \mathcal{D}} ||\Psi_l(\mathbf{x}) - \mathbf{x}|| \leq \epsilon. \tag{20}$$

Let us then fix $(C_k)_{1 \leq k < l}$ and $(\bar{\boldsymbol{\omega}}^1, \bar{\mathbf{b}}^1, \ldots, \bar{\boldsymbol{\omega}}^l, \bar{\mathbf{b}}^l)$ such that Eq. (20) holds. Let us further define the weights and biases $\boldsymbol{\omega}^k$, $\mathbf{b}^k$ at each layer $k$ such that $\forall h, w, c, c'$:

$$\boldsymbol{\omega}^k_{h,w,c,c'} \equiv \begin{cases} \bar{\boldsymbol{\omega}}^k_{1,1,c,c'} & \begin{array}{l} \text{if the multi-index } (h,w,c,c') \text{ in the weights} \\ \text{associates spatial positions } \alpha \text{ in the convolution input } \mathbf{z}^{l-1}_{\alpha,c} \\ \text{to the same spatial positions } \alpha \text{ in the convolution output } \mathbf{y}^l_{\alpha,c'}, \end{array} \\ \\ 0 & \text{otherwise,} \end{cases}$$

$$\mathbf{b}^k_{c'} \equiv \bar{\mathbf{b}}^k_{c'}.$$

Then it holds that $\Phi_l = \Psi_l$, which in turn implies

$$\max_{\mathbf{x} \in \mathcal{D}} ||\Phi_l(\mathbf{x}) - \mathbf{x}|| \leq \epsilon. \qquad \square$$

**Proposition 5.** *Suppose that the neural network implements Eq. (2), (3), (4) in every layer up to depth $L$ and suppose $\mathrm{Norm} = \mathrm{IN}$.*

*Further fix a layer $l \in \{1, \ldots, L\}$, the spatial extents $H, W$, the widths $C_0$, $C_l$ assumed equal at layer 0 and layer $l$, and any dataset $\mathcal{D}$ such that there exists at least one channel in which the inputs of $\mathcal{D}$ do not all share the same statistics of instance mean, i.e.*

$$\exists c, \exists \mathbf{x}', \mathbf{x}'' \in \mathcal{D} : \qquad \mathbb{E}_\alpha[\mathbf{x}'_{\alpha,c}] \neq \mathbb{E}_\alpha[\mathbf{x}''_{\alpha,c}].$$

*Then there exists $\epsilon > 0$ such that for any choice of intermediate widths $(C_k)_{1 \leq k < l}$ and model parameters $(\boldsymbol{\omega}^1, \boldsymbol{\beta}^1, \boldsymbol{\gamma}^1, \ldots, \boldsymbol{\omega}^l, \boldsymbol{\beta}^l, \boldsymbol{\gamma}^l)$, it holds that*

$$\max_{\mathbf{x} \in \mathcal{D}} ||\Phi_l(\mathbf{x}) - \mathbf{x}|| > \epsilon,$$

*where $\Phi_l$ denotes the network mapping from $\mathbf{x}$ to $\tilde{\mathbf{y}}^l$ such that $\tilde{\mathbf{y}}^l = \Phi_l(\mathbf{x})$, $\forall \mathbf{x}$.*

**Proof.** Let us proceed by contradiction and suppose that for any $\epsilon > 0$, there exists a choice of intermediate widths $(C_k)_{1 \leq k < l}$ and model parameters $(\boldsymbol{\omega}^1, \boldsymbol{\beta}^1, \boldsymbol{\gamma}^1, \ldots, \boldsymbol{\omega}^l, \boldsymbol{\beta}^l, \boldsymbol{\gamma}^l)$ such that

$$\max_{\mathbf{x} \in \mathcal{D}} ||\Phi_l(\mathbf{x}) - \mathbf{x}|| \leq \epsilon. \tag{21}$$

Given the assumption on $\mathcal{D}$, there exists some channel $c$ and some inputs $\mathbf{x}', \mathbf{x}'' \in \mathcal{D}$ such that

$$\mathbb{E}_\alpha[\mathbf{x}'_{\alpha,c}] \neq \mathbb{E}_\alpha[\mathbf{x}''_{\alpha,c}]. \tag{22}$$

Let us then fix $c$ and $\mathbf{x}', \mathbf{x}'' \in \mathcal{D}$ satisfying Eq. (22), and let us define $\eta \equiv \left| \mathbb{E}_\alpha[\mathbf{x}'_{\alpha,c}] - \mathbb{E}_\alpha[\mathbf{x}''_{\alpha,c}] \right| > 0$.

Applying Eq. (21) with $\epsilon = \frac{\sqrt{HW}}{4}\eta$, we get that there exists a choice of intermediate widths $(C_k)_{1 \leq k < l}$ and model parameters $(\boldsymbol{\omega}^1, \boldsymbol{\beta}^1, \boldsymbol{\gamma}^1, \ldots, \boldsymbol{\omega}^l, \boldsymbol{\beta}^l, \boldsymbol{\gamma}^l)$ such that

$$\max_{\mathbf{x} \in \mathcal{D}} ||\Phi_l(\mathbf{x}) - \mathbf{x}|| \leq \frac{\sqrt{HW}}{4}\eta. \tag{23}$$

Let us then fix $(C_k)_{1 \leq k < l}$ and $(\boldsymbol{\omega}^1, \boldsymbol{\beta}^1, \boldsymbol{\gamma}^1, \ldots, \boldsymbol{\omega}^l, \boldsymbol{\beta}^l, \boldsymbol{\gamma}^l)$ such that Eq. (23) holds. This gives

$$\sum_\alpha \left( \Phi_l(\mathbf{x}')_{\alpha,c} - \mathbf{x}'_{\alpha,c} \right)^2 \leq ||\Phi_l(\mathbf{x}') - \mathbf{x}'||^2 \leq \left( \frac{\sqrt{HW}}{4}\eta \right)^2,$$

$$\sum_\alpha \left( \Phi_l(\mathbf{x}'')_{\alpha,c} - \mathbf{x}''_{\alpha,c} \right)^2 \leq ||\Phi_l(\mathbf{x}'') - \mathbf{x}''||^2 \leq \left( \frac{\sqrt{HW}}{4}\eta \right)^2,$$

$$\mathbb{E}_\alpha\left[ \left( \Phi_l(\mathbf{x}')_{\alpha,c} - \mathbf{x}'_{\alpha,c} \right)^2 \right] + \mathbb{E}_\alpha\left[ \left( \Phi_l(\mathbf{x}'')_{\alpha,c} - \mathbf{x}''_{\alpha,c} \right)^2 \right] \leq \frac{2}{HW} \left( \frac{\sqrt{HW}}{4}\eta \right)^2. \tag{24}$$

At the same time, for any input $\mathbf{x}$, it holds that

$$\mathbb{E}_\alpha\left[ \left( \Phi_l(\mathbf{x})_{\alpha,c} - \mathbf{x}_{\alpha,c} \right)^2 \right] = \mathbb{E}_\alpha\left[ \Phi_l(\mathbf{x})_{\alpha,c} - \mathbf{x}_{\alpha,c} \right]^2 + \mathrm{Var}_\alpha\left[ \Phi_l(\mathbf{x})_{\alpha,c} - \mathbf{x}_{\alpha,c} \right]$$

$$\geq \mathbb{E}_\alpha\left[ \Phi_l(\mathbf{x})_{\alpha,c} - \mathbf{x}_{\alpha,c} \right]^2. \tag{25}$$

Using $\forall a, b: (a - b)^2 \leq 2a^2 + 2b^2$, combined with Eq. (25) and Eq. (24), we get

$$\left( \mathbb{E}_\alpha\left[ \Phi_l(\mathbf{x}')_{\alpha,c} - \mathbf{x}'_{\alpha,c} \right] - \mathbb{E}_\alpha\left[ \Phi_l(\mathbf{x}'')_{\alpha,c} - \mathbf{x}''_{\alpha,c} \right] \right)^2$$

$$\leq 2\mathbb{E}_\alpha\left[ \Phi_l(\mathbf{x}')_{\alpha,c} - \mathbf{x}'_{\alpha,c} \right]^2 + 2\mathbb{E}_\alpha\left[ \Phi_l(\mathbf{x}'')_{\alpha,c} - \mathbf{x}''_{\alpha,c} \right]^2$$

$$\leq 2\mathbb{E}_\alpha\left[ \left( \Phi_l(\mathbf{x}')_{\alpha,c} - \mathbf{x}'_{\alpha,c} \right)^2 \right] + 2\mathbb{E}_\alpha\left[ \left( \Phi_l(\mathbf{x}'')_{\alpha,c} - \mathbf{x}''_{\alpha,c} \right)^2 \right]$$

$$\leq \frac{4}{HW} \left( \frac{\sqrt{HW}}{4}\eta \right)^2. \tag{26}$$

Next, we note that with IN all inputs $\mathbf{x}$ are associated to the same instance means in each channel of $\tilde{\mathbf{y}}^l = \Phi_l(\mathbf{x})$. This means in particular that

$$\mathbb{E}_\alpha\left[ \Phi_l(\mathbf{x}')_{\alpha,c} \right] = \mathbb{E}_\alpha\left[ \Phi_l(\mathbf{x}'')_{\alpha,c} \right]. \tag{27}$$

Combining Eq. (26) with Eq. (27), we get

$$\left( \mathbb{E}_\alpha[\mathbf{x}'_{\alpha,c}] - \mathbb{E}_\alpha[\mathbf{x}''_{\alpha,c}] \right)^2 \leq \frac{4}{HW} \left( \frac{\sqrt{HW}}{4}\eta \right)^2,$$

$$\left| \mathbb{E}_\alpha[\mathbf{x}'_{\alpha,c}] - \mathbb{E}_\alpha[\mathbf{x}''_{\alpha,c}] \right| \leq \frac{2}{\sqrt{HW}} \frac{\sqrt{HW}}{4}\eta = \frac{\eta}{2}.$$

Since we earlier defined $\eta$ as $\eta \equiv \left| \mathbb{E}_\alpha[\mathbf{x}'_{\alpha,c}] - \mathbb{E}_\alpha[\mathbf{x}''_{\alpha,c}] \right| > 0$, we reach a contradiction. $\qquad\square$

# D   Proof of Theorem 1

## D.1   Additional notations

Some additional notations are required in this section.

**Model parameters.**   We introduce the notations $\theta^l \equiv (\boldsymbol{\omega}^l, \boldsymbol{\beta}^l, \boldsymbol{\gamma}^l)$ for the model parameters at layer $l$ and $\Theta^l \equiv (\boldsymbol{\omega}^1, \boldsymbol{\beta}^1, \boldsymbol{\gamma}^1, \ldots, \boldsymbol{\omega}^l, \boldsymbol{\beta}^l, \boldsymbol{\gamma}^l)$ for the aggregated model parameters up to layer $l$.

**Activation tensors.**   For each layer $l$, we define the tensors $\hat{\mathbf{z}}^{l-1}, \hat{\mathbf{x}}^l, \hat{\mathbf{y}}^l, \check{\mathbf{y}}^l, \check{\mathbf{z}}^l$ such that $\forall \alpha, c$:

$$\hat{\mathbf{z}}^{l-1}_{\alpha,c} = \sqrt{\frac{\mathcal{P}(\mathbf{z}^{l-1})}{\mathcal{P}_{\mathbf{x}}(\mathbf{z}^{l-1})}} \mathbf{z}^{l-1}_{\alpha,c}, \tag{28}$$

$$\hat{\mathbf{x}}^l_{\alpha,c} = \sqrt{\frac{\mathcal{P}(\mathbf{z}^{l-1})}{\mathcal{P}_{\mathbf{x}}(\mathbf{z}^{l-1})}} \mathbf{x}^l_{\alpha,c} = (\boldsymbol{\omega}^l * \hat{\mathbf{z}}^{l-1})_{\alpha,c}, \tag{29}$$

$$\hat{\mathbf{y}}^l_{\alpha,c} = \frac{1}{\omega \sqrt{\mathcal{P}(\mathbf{z}^{l-1})}} \hat{\mathbf{x}}^l_{\alpha,c}, \tag{30}$$

$$\check{\mathbf{y}}^l_{\alpha,c} = \gamma^l_c \hat{\mathbf{y}}^l_{\alpha,c} + \beta^l_c, \tag{31}$$

$$\check{\mathbf{z}}^l_{\alpha,c} = \phi(\check{\mathbf{y}}^l_{\alpha,c}), \tag{32}$$

with the convention that, if $\mathcal{P}_{\mathbf{x}}(\mathbf{z}^{l-1}) = 0$, then $\forall \alpha, c$: $\hat{\mathbf{z}}^{l-1}_{\alpha,c} = 0$, $\hat{\mathbf{x}}^l_{\alpha,c} = 0$ and if $\mathcal{P}(\mathbf{z}^{l-1}) = 0$, then $\forall \alpha, c$: $\hat{\mathbf{y}}^l_{\alpha,c} = 0$.

**Moments.**   We introduce the notation $\varrho(\mathbf{y}^l)$ for the ratio of the traces of the covariance matrix and Gram matrix of the activation vectors $(\mathbf{y}^l_{\alpha,1}, \ldots, \mathbf{y}^l_{\alpha,C_l})^{\mathrm{T}}$ with respect to the randomness from $(\mathbf{x}, \alpha)$, i.e.

$$\varrho(\mathbf{y}^l) \equiv \frac{\mathcal{P}(\mathbf{y}^l) - \mathcal{P}^{(1)}(\mathbf{y}^l)}{\mathcal{P}(\mathbf{y}^l)} \leq 1,$$

with the convention that, if $\mathcal{P}(\mathbf{y}^l) = 0$, then $\varrho(\mathbf{y}^l) = 0$.

We extend the definition of the terms $\mathcal{P}^{(1)}_c(\mathbf{y}^l), \mathcal{P}^{(2)}_c(\mathbf{y}^l), \mathcal{P}^{(3)}_c(\mathbf{y}^l), \mathcal{P}^{(4)}_c(\mathbf{y}^l)$ and $\mathcal{P}^{(1)}(\mathbf{y}^l), \mathcal{P}^{(2)}(\mathbf{y}^l), \mathcal{P}^{(3)}(\mathbf{y}^l), \mathcal{P}^{(4)}(\mathbf{y}^l), \varrho(\mathbf{y}^l)$ to all the other activation tensors of layer $l$.

## D.2   Required Lemmas

**Lemma 1.** *Fix a layer $l \geq 1$, $\nu_{\boldsymbol{\omega}}$, $\nu_{\boldsymbol{\beta}}$, $\nu_{\boldsymbol{\gamma}}$, $\mathcal{D}$ in Definition 1 and model parameters $\Theta^{l-1}$ up to layer $l-1$ such that $\mathcal{P}_{\mathbf{x}}(\mathbf{z}^{l-1}) > 0$, $\forall \mathbf{x}$. Further suppose $\mathrm{Norm} = \mathrm{LN}$ and suppose that the convolution of Eq. (2) uses periodic boundary conditions.*

*Then for any $\eta > 0$ and any $\delta > 0$, there exists $N'(\eta, \delta) \in \mathbb{N}^*$ independent of $\Theta^{l-1}$, $l$ such that if $C_l \geq N'(\eta, \delta)$, it holds for random nets of Definition 1 that*

$$\mathbb{P}_{\theta_l}\left[ \left| \varrho(\hat{\mathbf{y}}^l) - \varrho(\hat{\mathbf{z}}^{l-1}) \right| \leq \eta \right] \geq 1 - \delta,$$

$$\mathbb{P}_{\theta_l}\left[ \left| \varrho(\check{\mathbf{z}}^l) - \rho \chi(\hat{\mathbf{z}}^{l-1}) \varrho(\hat{\mathbf{z}}^{l-1}) \right| \leq \eta \right] \geq 1 - \delta,$$

*where $\hat{\mathbf{z}}^{l-1}, \hat{\mathbf{y}}^l, \check{\mathbf{z}}^l$ are defined in Eq. (28), (30), (32), and where $\rho = \frac{\gamma^2}{\gamma^2 + \beta^2} < 1$ and $\chi(\hat{\mathbf{z}}^{l-1}) \in \mathbb{R}^+$ is dependent on $\Theta^{l-1}$ but independent of $\theta^l$ such that $\chi(\hat{\mathbf{z}}^{l-1}) \leq 1$ in general and $\chi(\hat{\mathbf{z}}^{l-1}) = 1$ if $\phi = \mathrm{identity}$.*

**Proof.** First noting that $\hat{\mathbf{x}}^l = \boldsymbol{\omega}^l * \hat{\mathbf{z}}^{l-1}$, we define $\hat{\mathbf{r}}^{l-1} \in \mathbb{R}^{H \times W \times K_l^2 C_{l-1}}$ the "receptive field" tensor containing at each spatial position $\alpha \in \{1, \ldots, H\} \times \{1, \ldots, W\}$ the $K_l^2 C_{l-1}$ elements of $\hat{\mathbf{z}}^{l-1}$ belonging to the receptive field of $(\hat{\mathbf{x}}^l_{\alpha,1}, \ldots, \hat{\mathbf{x}}^l_{\alpha,C_l})^{\mathrm{T}}$.

For a fixed fan-in element $c'$ originating from channel $c$, the assumption of periodic boundary conditions implies that $\hat{\mathbf{r}}^{l-1}_{\alpha,c'}$ has the same distribution as $\hat{\mathbf{z}}^{l-1}_{\alpha,c}$ with respect to $(\mathbf{x},\alpha)$, implying

$$\mathcal{P}_{c'}(\hat{\mathbf{r}}^{l-1}) = \mathcal{P}_c(\hat{\mathbf{z}}^{l-1}), \qquad\qquad \mathcal{P}^{(1)}_{c'}(\hat{\mathbf{r}}^{l-1}) = \mathcal{P}^{(1)}_c(\hat{\mathbf{z}}^{l-1}).$$

Since the number of fan-in elements $c'$ originating from channel $c$ is equal to $K_l^2$ for any choice of $c$, it follows that

$$\mathcal{P}(\hat{\mathbf{r}}^{l-1}) = \mathcal{P}(\hat{\mathbf{z}}^{l-1}), \qquad\qquad \mathcal{P}^{(1)}(\hat{\mathbf{r}}^{l-1}) = \mathcal{P}^{(1)}(\hat{\mathbf{z}}^{l-1}). \tag{33}$$

Now if we denote $\boldsymbol{W}^l \in \mathbb{R}^{C_l \times K_l^2 C_{l-1}}$ the reshaped matricial form of $\boldsymbol{\omega}^l$, we obtain that $\forall \alpha$: $(\hat{\mathbf{x}}^l_{\alpha,1}, \ldots, \hat{\mathbf{x}}^l_{\alpha,C_l})^{\mathrm{T}} = \boldsymbol{W}^l(\hat{\mathbf{r}}^{l-1}_{\alpha,1}, \ldots, \hat{\mathbf{r}}^{l-1}_{\alpha,K_l^2 C_{l-1}})^{\mathrm{T}}$, implying $\forall c$:

$$\mu_c(\hat{\mathbf{x}}^l) = \mathbb{E}_{\mathbf{x},\alpha}\left[\hat{\mathbf{x}}^l_{\alpha,c}\right] = \mathbb{E}_{\mathbf{x},\alpha}\left[\sum_{c'} \boldsymbol{W}^l_{cc'}\hat{\mathbf{r}}^{l-1}_{\alpha,c'}\right], \tag{34}$$

$$\mathcal{P}_c(\hat{\mathbf{x}}^l) = \mathbb{E}_{\mathbf{x},\alpha}\left[(\hat{\mathbf{x}}^l_{\alpha,c})^2\right] = \mathbb{E}_{\mathbf{x},\alpha}\left[\left(\sum_{c'} \boldsymbol{W}^l_{cc'}\hat{\mathbf{r}}^{l-1}_{\alpha,c'}\right)^2\right], \tag{35}$$

$$\mathcal{P}^{(1)}_c(\hat{\mathbf{x}}^l) = \mu_c(\hat{\mathbf{x}}^l)^2 = \mathbb{E}_{\mathbf{x},\alpha}\left[\sum_{c'} \boldsymbol{W}^l_{cc'}\hat{\mathbf{r}}^{l-1}_{\alpha,c'}\right]^2. \tag{36}$$

Further expanding Eq. (34), (35), (36), we get $\forall c$:

$$\mu_c(\hat{\mathbf{x}}^l) = \sum_{c'} \boldsymbol{W}^l_{cc'}\mathbb{E}_{\mathbf{x},\alpha}\left[\hat{\mathbf{r}}^{l-1}_{\alpha,c'}\right],$$

$$\mathcal{P}_c(\hat{\mathbf{x}}^l) = \sum_{c',c''} \boldsymbol{W}^l_{cc'}\boldsymbol{W}^l_{cc''}\mathbb{E}_{\mathbf{x},\alpha}\left[\hat{\mathbf{r}}^{l-1}_{\alpha,c'}\hat{\mathbf{r}}^{l-1}_{\alpha,c''}\right],$$

$$\mathcal{P}^{(1)}_c(\hat{\mathbf{x}}^l) = \sum_{c',c''} \boldsymbol{W}^l_{cc'}\boldsymbol{W}^l_{cc''}\mathbb{E}_{\mathbf{x},\alpha}\left[\hat{\mathbf{r}}^{l-1}_{\alpha,c'}\right]\mathbb{E}_{\mathbf{x},\alpha}\left[\hat{\mathbf{r}}^{l-1}_{\alpha,c''}\right],$$

$$\mathcal{P}_c(\hat{\mathbf{x}}^l) - \mathcal{P}^{(1)}_c(\hat{\mathbf{x}}^l) = \sum_{c',c''} \boldsymbol{W}^l_{cc'}\boldsymbol{W}^l_{cc''}\left(\mathbb{E}_{\mathbf{x},\alpha}\left[\hat{\mathbf{r}}^{l-1}_{\alpha,c'}\hat{\mathbf{r}}^{l-1}_{\alpha,c''}\right] - \mathbb{E}_{\mathbf{x},\alpha}\left[\hat{\mathbf{r}}^{l-1}_{\alpha,c'}\right]\mathbb{E}_{\mathbf{x},\alpha}\left[\hat{\mathbf{r}}^{l-1}_{\alpha,c''}\right]\right).$$

Since the components of $\sqrt{K_l^2 C_{l-1}}\boldsymbol{W}^l$ are sampled i.i.d. from the fixed distribution $\nu_{\boldsymbol{\omega}}$ which is assumed symmetric around zero, we get $\forall c$:

$$\mathbb{E}_{\theta^l}\left[\mu_c(\hat{\mathbf{x}}^l)\right] = 0, \tag{37}$$

$$\mathbb{E}_{\theta^l}\left[\mathcal{P}_c(\hat{\mathbf{x}}^l)\right] = \sum_{c'} \mathbb{E}_{\theta^l}\left[(\boldsymbol{W}^l_{cc'})^2\right]\mathbb{E}_{\mathbf{x},\alpha}\left[(\hat{\mathbf{r}}^{l-1}_{\alpha,c'})^2\right]$$

$$= \sum_{c'} \frac{\omega^2}{K_l^2 C_{l-1}}\mathcal{P}_{c'}(\hat{\mathbf{r}}^{l-1})$$

$$= \omega^2 \mathcal{P}(\hat{\mathbf{z}}^{l-1}), \tag{38}$$

$$\mathbb{E}_{\theta^l}\left[\mathcal{P}_c(\hat{\mathbf{x}}^l) - \mathcal{P}^{(1)}_c(\hat{\mathbf{x}}^l)\right] = \sum_{c'} \mathbb{E}_{\theta^l}\left[(\boldsymbol{W}^l_{cc'})^2\right]\left(\mathbb{E}_{\mathbf{x},\alpha}\left[(\hat{\mathbf{r}}^{l-1}_{\alpha,c'})^2\right] - \mathbb{E}_{\mathbf{x},\alpha}\left[\hat{\mathbf{r}}^{l-1}_{\alpha,c'}\right]^2\right)$$

$$= \sum_{c'} \frac{\omega^2}{K_l^2 C_{l-1}}\left(\mathcal{P}_{c'}(\hat{\mathbf{r}}^{l-1}) - \mathcal{P}^{(1)}_{c'}(\hat{\mathbf{r}}^{l-1})\right)$$

$$= \omega^2\left(\mathcal{P}(\hat{\mathbf{z}}^{l-1}) - \mathcal{P}^{(1)}(\hat{\mathbf{z}}^{l-1})\right), \tag{39}$$

where we recall that $\omega \equiv \mathbb{E}_{\theta^l}\left[\left(\sqrt{K_l^2 C_{l-1}} \boldsymbol{W}_{cc'}^l\right)^2\right]^{1/2} > 0$ is the $L^2$ norm (i.e. the root mean square) of $\sqrt{K_l^2 C_{l-1}} \boldsymbol{W}_{cc'}^l \sim \nu_{\boldsymbol{\omega}}$, and where we used Eq. (33) in Eq. (38) and Eq. (39).

Let us now bound $\mathbb{E}_{\theta^l}\left[\mu_c(\hat{\mathbf{x}}^l)^2\right]$, $\mathbb{E}_{\theta^l}\left[\mathcal{P}_c(\hat{\mathbf{x}}^l)^2\right]$, with the aim of bounding $\mathrm{Var}_{\theta^l}\left[\mu_c(\hat{\mathbf{y}}^l)\right]$, $\mathrm{Var}_{\theta^l}\left[\mathcal{P}_c(\hat{\mathbf{y}}^l)\right]$ later on. We start by expanding $\mu_c(\hat{\mathbf{x}}^l)^2$ and $\mathcal{P}_c(\hat{\mathbf{x}}^l)^2$ as

$$\mu_c(\hat{\mathbf{x}}^l)^2 = \mathcal{P}_c(\hat{\mathbf{x}}^l) - \left(\mathcal{P}_c(\hat{\mathbf{x}}^l) - \mathcal{P}_c^{(1)}(\hat{\mathbf{x}}^l)\right),$$

$$\mathcal{P}_c(\hat{\mathbf{x}}^l)^2 = \left(\sum_{c'}(\boldsymbol{W}_{cc'}^l)^2 \mathbb{E}_{\mathbf{x},\alpha}\left[(\hat{\mathbf{r}}_{\alpha,c'}^{l-1})^2\right] + \sum_{c'}\sum_{c''\neq c'} \boldsymbol{W}_{cc'}^l \boldsymbol{W}_{cc''}^l \mathbb{E}_{\mathbf{x},\alpha}\left[\hat{\mathbf{r}}_{\alpha,c'}^{l-1}\hat{\mathbf{r}}_{\alpha,c''}^{l-1}\right]\right)^2$$

$$= \left(\sum_{c'}(\boldsymbol{W}_{cc'}^l)^2 \mathbb{E}_{\mathbf{x},\alpha}\left[(\hat{\mathbf{r}}_{\alpha,c'}^{l-1})^2\right] + 2\sum_{c'}\sum_{c''<c'} \boldsymbol{W}_{cc'}^l \boldsymbol{W}_{cc''}^l \mathbb{E}_{\mathbf{x},\alpha}\left[\hat{\mathbf{r}}_{\alpha,c'}^{l-1}\hat{\mathbf{r}}_{\alpha,c''}^{l-1}\right]\right)^2.$$

For $\mathbb{E}_{\theta^l}\left[\mu_c(\hat{\mathbf{x}}^l)^2\right]$, we get from Eq. (38) and Eq. (39) that

$$\begin{aligned}
\mathbb{E}_{\theta^l}\left[\mu_c(\hat{\mathbf{x}}^l)^2\right] &= \omega^2 \mathcal{P}(\hat{\mathbf{z}}^{l-1}) - \omega^2\left(\mathcal{P}(\hat{\mathbf{z}}^{l-1}) - \mathcal{P}^{(1)}(\hat{\mathbf{z}}^{l-1})\right) \\
&= \omega^2 \mathcal{P}^{(1)}(\hat{\mathbf{z}}^{l-1}) \\
&\leq \omega^2 \mathcal{P}(\hat{\mathbf{z}}^{l-1}).
\end{aligned} \tag{40}$$

As for $\mathbb{E}_{\theta^l}\left[\mathcal{P}_c(\hat{\mathbf{x}}^l)^2\right]$, given that only terms in $(\boldsymbol{W}_{cc'}^l)^4$ and $(\boldsymbol{W}_{cc'}^l)^2(\boldsymbol{W}_{cc''}^l)^2$ remain when taking the expectation over $\theta^l$, we get

$$\mathbb{E}_{\theta^l}\left[\mathcal{P}_c(\hat{\mathbf{x}}^l)^2\right]$$

$$= \mathbb{E}_{\theta^l}\left[\left(\sum_{c'}(\boldsymbol{W}_{cc'}^l)^2 \mathbb{E}_{\mathbf{x},\alpha}\left[(\hat{\mathbf{r}}_{\alpha,c'}^{l-1})^2\right]\right)^2 + 4\sum_{c'}\sum_{c''<c'}(\boldsymbol{W}_{cc'}^l)^2(\boldsymbol{W}_{cc''}^l)^2 \mathbb{E}_{\mathbf{x},\alpha}\left[\hat{\mathbf{r}}_{\alpha,c'}^{l-1}\hat{\mathbf{r}}_{\alpha,c''}^{l-1}\right]^2\right]$$

$$\leq \mathbb{E}_{\theta^l}\left[\left(\sum_{c'}(\boldsymbol{W}_{cc'}^l)^2 \mathcal{P}_{c'}(\hat{\mathbf{r}}^{l-1})\right)^2 + 4\sum_{c'}\sum_{c''<c'}(\boldsymbol{W}_{cc'}^l)^2(\boldsymbol{W}_{cc''}^l)^2 \mathcal{P}_{c'}(\hat{\mathbf{r}}^{l-1})\mathcal{P}_{c''}(\hat{\mathbf{r}}^{l-1})\right] \tag{41}$$

$$\leq \mathbb{E}_{\theta^l}\left[\sum_{c'}(\boldsymbol{W}_{cc'}^l)^4 \mathcal{P}_{c'}(\hat{\mathbf{r}}^{l-1})^2 + 6\sum_{c'}\sum_{c''<c'}(\boldsymbol{W}_{cc'}^l)^2(\boldsymbol{W}_{cc''}^l)^2 \mathcal{P}_{c'}(\hat{\mathbf{r}}^{l-1})\mathcal{P}_{c''}(\hat{\mathbf{r}}^{l-1})\right]$$

$$\leq \sum_{c'}\mathbb{E}_{\theta^l}\left[(\boldsymbol{W}_{cc'}^l)^4\right]\mathcal{P}_{c'}(\hat{\mathbf{r}}^{l-1})^2 + 3\sum_{c'}\sum_{c''\neq c'}\mathbb{E}_{\theta^l}\left[(\boldsymbol{W}_{cc'}^l)^2(\boldsymbol{W}_{cc''}^l)^2\right]\mathcal{P}_{c'}(\hat{\mathbf{r}}^{l-1})\mathcal{P}_{c''}(\hat{\mathbf{r}}^{l-1})$$

$$\leq \sum_{c'}\mathbb{E}_{\theta^l}\left[(\boldsymbol{W}_{cc'}^l)^4\right]\mathcal{P}_{c'}(\hat{\mathbf{r}}^{l-1})^2 + 3\sum_{c'}\sum_{c''\neq c'}\mathbb{E}_{\theta^l}\left[(\boldsymbol{W}_{cc'}^l)^4\right]\mathcal{P}_{c'}(\hat{\mathbf{r}}^{l-1})\mathcal{P}_{c''}(\hat{\mathbf{r}}^{l-1}), \tag{42}$$

where Eq. (41) and Eq. (42) are obtained using Cauchy-Schwarz inequality combined with $\mathbb{E}_{\mathbf{x},\alpha}\left[(\hat{\mathbf{r}}_{\alpha,c'}^{l-1})^2\right] = \mathcal{P}_{c'}(\hat{\mathbf{r}}^{l-1})$ and $\mathbb{E}_{\mathbf{x},\alpha}\left[(\hat{\mathbf{r}}_{\alpha,c''}^{l-1})^2\right] = \mathcal{P}_{c''}(\hat{\mathbf{r}}^{l-1})$.

We may further process Eq. (42) to get

$$\begin{aligned}
\mathbb{E}_{\theta^l}\left[\mathcal{P}_c(\hat{\mathbf{x}}^l)^2\right] &\leq 3\sum_{c'}\mathbb{E}_{\theta^l}\left[(\boldsymbol{W}_{cc'}^l)^4\right]\mathcal{P}_{c'}(\hat{\mathbf{r}}^{l-1})^2 + 3\sum_{c'}\sum_{c''\neq c'}\mathbb{E}_{\theta^l}\left[(\boldsymbol{W}_{cc'}^l)^4\right]\mathcal{P}_{c'}(\hat{\mathbf{r}}^{l-1})\mathcal{P}_{c''}(\hat{\mathbf{r}}^{l-1}) \\
&\leq 3\sum_{c',c''}\mathbb{E}_{\theta^l}\left[(\boldsymbol{W}_{cc'}^l)^4\right]\mathcal{P}_{c'}(\hat{\mathbf{r}}^{l-1})\mathcal{P}_{c''}(\hat{\mathbf{r}}^{l-1}) \\
&\leq 3\mathbb{E}_{\theta^l}\left[(\boldsymbol{W}_{c,1}^l)^4\right]\left(K_l^2 C_{l-1}\right)^2 \mathcal{P}(\hat{\mathbf{r}}^{l-1})^2 \\
&\leq 3\mathbb{E}_{\theta^l}\left[\left(\sqrt{K_l^2 C_{l-1}} \boldsymbol{W}_{c,1}^l\right)^4\right]\mathcal{P}(\hat{\mathbf{z}}^{l-1})^2 \\
&\leq 3\tilde{\omega}^4 \mathcal{P}(\hat{\mathbf{z}}^{l-1})^2,
\end{aligned} \tag{43}$$

where $\tilde{\omega} \equiv \mathbb{E}_{\theta^l}\left[\left(\sqrt{K_l^2 C_{l-1}} \boldsymbol{W}_{c,1}^l\right)^4\right]^{1/4} \geq \omega > 0$ is the $L^4$ norm of $\sqrt{K_l^2 C_{l-1}} \boldsymbol{W}_{c,1}^l \sim \nu_{\boldsymbol{\omega}}$.

Now we turn to $\hat{\mathbf{y}}^l$ and $\breve{\mathbf{y}}^l$ defined in Eq. (30) and Eq. (31).

Due to $\mu_c(\hat{\mathbf{y}}^l) = \frac{1}{w\sqrt{\mathcal{P}(\mathbf{z}^{l-1})}}\mu_c(\hat{\mathbf{x}}^l)$, $\mathcal{P}_c(\hat{\mathbf{y}}^l) = \frac{1}{w^2 \mathcal{P}(\mathbf{z}^{l-1})}\mathcal{P}_c(\hat{\mathbf{x}}^l)$ and $\mathcal{P}_c^{(1)}(\hat{\mathbf{y}}^l) = \frac{1}{w^2 \mathcal{P}(\mathbf{z}^{l-1})}\mathcal{P}_c^{(1)}(\hat{\mathbf{x}}^l)$

and due to $\mathcal{P}(\hat{\mathbf{z}}^{l-1}) = \mathcal{P}(\mathbf{z}^{l-1})\mathbb{E}_{\mathbf{x}}\left[\frac{1}{\mathcal{P}_{\mathbf{x}}(\mathbf{z}^{l-1})}\mathcal{P}_{\mathbf{x}}(\mathbf{z}^{l-1})\right] = \mathcal{P}(\mathbf{z}^{l-1})$, Eq. (37), (38), (39) imply

$$\mathbb{E}_{\theta^l}\left[\mu_c(\hat{\mathbf{y}}^l)\right] = 0,$$

$$\mathbb{E}_{\theta^l}\left[\mathcal{P}_c(\hat{\mathbf{y}}^l)\right] = 1, \tag{44}$$

$$\mathbb{E}_{\theta^l}\left[\mathcal{P}_c(\hat{\mathbf{y}}^l) - \mathcal{P}_c^{(1)}(\hat{\mathbf{y}}^l)\right] = \frac{\mathcal{P}(\hat{\mathbf{z}}^{l-1}) - \mathcal{P}^{(1)}(\hat{\mathbf{z}}^{l-1})}{\mathcal{P}(\hat{\mathbf{z}}^{l-1})} = \varrho(\hat{\mathbf{z}}^{l-1}). \tag{45}$$

Using Eq. (40) and Eq. (43), we further get

$$\mathrm{Var}_{\theta^l}\left[\mu_c(\hat{\mathbf{y}}^l)\right] = \mathbb{E}_{\theta^l}\left[\mu_c(\hat{\mathbf{y}}^l)^2\right] \leq 1,$$

$$\mathrm{Var}_{\theta^l}\left[\mathcal{P}_c(\hat{\mathbf{y}}^l)\right] \leq \mathbb{E}_{\theta^l}\left[\mathcal{P}_c(\hat{\mathbf{y}}^l)^2\right] \leq 3\tilde{\omega}^4\omega^{-4}, \tag{46}$$

$$\mathrm{Var}_{\theta^l}\left[\mathcal{P}_c(\hat{\mathbf{y}}^l) - \mathcal{P}_c^{(1)}(\hat{\mathbf{y}}^l)\right] \leq \mathbb{E}_{\theta^l}\left[\left(\mathcal{P}_c(\hat{\mathbf{y}}^l) - \mathcal{P}_c^{(1)}(\hat{\mathbf{y}}^l)\right)^2\right] \leq \mathbb{E}_{\theta^l}\left[\mathcal{P}_c(\hat{\mathbf{y}}^l)^2\right] \leq 3\tilde{\omega}^4\omega^{-4}.$$

The terms $\mu_c(\hat{\mathbf{y}}^l)$, $\mathcal{P}_c(\hat{\mathbf{y}}^l)$ and $\mathcal{P}_c(\hat{\mathbf{y}}^l) - \mathcal{P}_c^{(1)}(\hat{\mathbf{y}}^l)$ being i.i.d. with respect to $\theta^l$ in the different channels $c$, we get

$$\mathbb{E}_{\theta^l}\left[\mu(\hat{\mathbf{y}}^l)\right] = 0, \qquad\qquad \mathrm{Var}_{\theta^l}\left[\mu(\hat{\mathbf{y}}^l)\right] \leq \frac{1}{C_l}, \tag{47}$$

$$\mathbb{E}_{\theta^l}\left[\mathcal{P}(\hat{\mathbf{y}}^l)\right] = 1, \qquad\qquad \mathrm{Var}_{\theta^l}\left[\mathcal{P}(\hat{\mathbf{y}}^l)\right] \leq \frac{3\tilde{\omega}^4\omega^{-4}}{C_l}, \tag{48}$$

$$\mathbb{E}_{\theta^l}\left[\mathcal{P}(\hat{\mathbf{y}}^l) - \mathcal{P}^{(1)}(\hat{\mathbf{y}}^l)\right] = \varrho(\hat{\mathbf{z}}^{l-1}), \qquad \mathrm{Var}_{\theta^l}\left[\mathcal{P}(\hat{\mathbf{y}}^l) - \mathcal{P}^{(1)}(\hat{\mathbf{y}}^l)\right] \leq \frac{3\tilde{\omega}^4\omega^{-4}}{C_l}. \tag{49}$$

Combining Eq. (48) and Eq. (49) with Chebyshev's inequality, we get for any $1 > \eta > 0$ that

$$\mathbb{P}_{\theta^l}\left[|\mathcal{P}(\hat{\mathbf{y}}^l) - 1| > \frac{\eta}{1+\eta}\right] \leq \left(\frac{1+\eta}{\eta}\right)^2 \frac{3\tilde{\omega}^4\omega^{-4}}{C_l},$$

$$\mathbb{P}_{\theta^l}\left[|\mathcal{P}(\hat{\mathbf{y}}^l) - 1| > \frac{\eta}{1-\eta}\right] \leq \left(\frac{1-\eta}{\eta}\right)^2 \frac{3\tilde{\omega}^4\omega^{-4}}{C_l},$$

$$\mathbb{P}_{\theta^l}\left[\left|\mathcal{P}(\hat{\mathbf{y}}^l) - \mathcal{P}^{(1)}(\hat{\mathbf{y}}^l) - \varrho(\hat{\mathbf{z}}^{l-1})\right| > \eta\right] \leq \left(\frac{1}{\eta}\right)^2 \frac{3\tilde{\omega}^4\omega^{-4}}{C_l}.$$

Thus, for any $1 > \eta > 0$ and any $\delta > 0$, there exists $N_1(\eta, \delta) \in \mathbb{N}^*$ independent of $\Theta^{l-1}$, $l$, such that if $C_l \geq N_1(\eta, \delta)$, it holds that

$$\mathbb{P}_{\theta^l}\left[\mathcal{P}(\hat{\mathbf{y}}^l) \geq \frac{1}{1+\eta}\right] \geq \mathbb{P}_{\theta^l}\left[|\mathcal{P}(\hat{\mathbf{y}}^l) - 1| \leq \frac{\eta}{1+\eta}\right] \geq 1 - \delta,$$

$$\mathbb{P}_{\theta^l}\left[\mathcal{P}(\hat{\mathbf{y}}^l) \leq \frac{1}{1-\eta}\right] \geq \mathbb{P}_{\theta^l}\left[|\mathcal{P}(\hat{\mathbf{y}}^l) - 1| \leq \frac{\eta}{1-\eta}\right] \geq 1 - \delta,$$

$$\mathbb{P}_{\theta^l}\left[\left|\mathcal{P}(\hat{\mathbf{y}}^l) - \mathcal{P}^{(1)}(\hat{\mathbf{y}}^l) - \varrho(\hat{\mathbf{z}}^{l-1})\right| \leq \eta\right] \geq 1 - \delta.$$

Thus, if $C_l \geq N_1(\eta, \delta)$ with $1 > \eta > 0$, it holds with probability greater than $1 - 3\delta$ with respect to $\theta^l$ that (i) $(1 + \eta)\mathcal{P}(\hat{\mathbf{y}}^l) \geq 1$, (ii) $(1 - \eta)\mathcal{P}(\hat{\mathbf{y}}^l) \leq 1$, (iii) $|\mathcal{P}(\hat{\mathbf{y}}^l) - \mathcal{P}^{(1)}(\hat{\mathbf{y}}^l) - \varrho(\hat{\mathbf{z}}^{l-1})| \leq \eta$, implying

$$
\begin{aligned}
\mathcal{P}(\hat{\mathbf{y}}^l) - \mathcal{P}^{(1)}(\hat{\mathbf{y}}^l) &\leq \varrho(\hat{\mathbf{z}}^{l-1}) + \eta \\
&\leq (\varrho(\hat{\mathbf{z}}^{l-1}) + \eta)(1 + \eta)\mathcal{P}(\hat{\mathbf{y}}^l) \\
&\leq (\varrho(\hat{\mathbf{z}}^{l-1}) + \varrho(\hat{\mathbf{z}}^{l-1})\eta + \eta + \eta^2)\mathcal{P}(\hat{\mathbf{y}}^l) \\
&\leq (\varrho(\hat{\mathbf{z}}^{l-1}) + 3\eta)\mathcal{P}(\hat{\mathbf{y}}^l), \\
\mathcal{P}(\hat{\mathbf{y}}^l) - \mathcal{P}^{(1)}(\hat{\mathbf{y}}^l) &\geq \max\left(0, \varrho(\hat{\mathbf{z}}^{l-1}) - \eta\right) \\
&\geq \max\left(0, \varrho(\hat{\mathbf{z}}^{l-1}) - \eta\right)(1 - \eta)\mathcal{P}(\hat{\mathbf{y}}^l) \\
&\geq \max\left(0, (\varrho(\hat{\mathbf{z}}^{l-1}) - \eta)(1 - \eta)\mathcal{P}(\hat{\mathbf{y}}^l)\right) \\
&\geq (\varrho(\hat{\mathbf{z}}^{l-1}) - \eta)(1 - \eta)\mathcal{P}(\hat{\mathbf{y}}^l) \\
&\geq (\varrho(\hat{\mathbf{z}}^{l-1}) - \varrho(\hat{\mathbf{z}}^{l-1})\eta - \eta + \eta^2)\mathcal{P}(\hat{\mathbf{y}}^l) \\
&\geq (\varrho(\hat{\mathbf{z}}^{l-1}) - 2\eta)\mathcal{P}(\hat{\mathbf{y}}^l),
\end{aligned}
$$

where we used $\varrho(\hat{\mathbf{z}}^{l-1}) \leq 1$ and $\eta^2 \leq \eta$ due to $\eta < 1$, as well as $\mathcal{P}(\hat{\mathbf{y}}^l) - \mathcal{P}^{(1)}(\hat{\mathbf{y}}^l) \geq 0$.

Given that $(1 + \eta)\mathcal{P}(\hat{\mathbf{y}}^l) \geq 1 \implies \mathcal{P}(\hat{\mathbf{y}}^l) > 0$, it follows that if $C_l \geq N_1(\eta, \delta)$ with $1 > \eta > 0$, it holds with probability greater than $1 - 3\delta$ with respect to $\theta^l$ that

$$
\begin{aligned}
\varrho(\hat{\mathbf{y}}^l) &\leq \varrho(\hat{\mathbf{z}}^{l-1}) + 3\eta, \\
\varrho(\hat{\mathbf{y}}^l) &\geq \varrho(\hat{\mathbf{z}}^{l-1}) - 2\eta, \\
|\varrho(\hat{\mathbf{y}}^l) - \varrho(\hat{\mathbf{z}}^{l-1})| &\leq 3\eta.
\end{aligned}
$$

Now, let $N_2$ be defined independently of $\Theta^{l-1}$, $l$ as $N_2(\eta, \delta) = N_1\left(\min\left(\frac{\eta}{3}, \frac{1}{2}\right), \frac{\delta}{3}\right)$, $\forall \eta > 0$, $\forall \delta > 0$. Then for any $\eta > 0$ and any $\delta > 0$, if $C_l \geq N_2(\eta, \delta)$, it holds that

$$
\begin{aligned}
\mathbb{P}_{\theta_l}\left[|\varrho(\hat{\mathbf{y}}^l) - \varrho(\hat{\mathbf{z}}^{l-1})| \leq \eta\right] &\geq \mathbb{P}_{\theta_l}\left[|\varrho(\hat{\mathbf{y}}^l) - \varrho(\hat{\mathbf{z}}^{l-1})| \leq 3\min\left(\frac{\eta}{3}, \frac{1}{2}\right)\right] \\
&\geq 1 - 3\frac{\delta}{3} \\
&\geq 1 - \delta.
\end{aligned} \tag{50}
$$

Let us apply a similar approach with respect to $\check{\mathbf{z}}^l$, first noting that $\forall \alpha, c$:

$$
\begin{aligned}
(\check{\mathbf{y}}_{\alpha,c}^l)^2 &= \left(\gamma_c^l \hat{\mathbf{y}}_{\alpha,c}^l + \beta_c^l\right)^2 = (\gamma_c^l)^2 (\hat{\mathbf{y}}_{\alpha,c}^l)^2 + (\beta_c^l)^2 + 2\gamma_c^l \beta_c^l \hat{\mathbf{y}}_{\alpha,c}^l, \\
\mathcal{P}_c(\check{\mathbf{y}}^l) &= (\gamma_c^l)^2 \mathcal{P}_c(\hat{\mathbf{y}}^l) + (\beta_c^l)^2 + 2\gamma_c^l \beta_c^l \mu_c(\hat{\mathbf{y}}^l), \\
\mathbb{E}_{\theta^l}\left[\mathcal{P}_c(\check{\mathbf{y}}^l)\right] &= \mathbb{E}_{\theta^l}\left[(\gamma_c^l)^2 \mathcal{P}_c(\hat{\mathbf{y}}^l) + (\beta_c^l)^2\right] = \gamma^2 \mathbb{E}_{\theta^l}\left[\mathcal{P}_c(\hat{\mathbf{y}}^l)\right] + \beta^2,
\end{aligned} \tag{51}
$$

where $\gamma$, $\beta$ are the $L^2$ norms (i.e. the root mean squares) of $\gamma_c^l \sim \nu_\gamma$ and $\beta_c^l \sim \nu_\beta$, and where we used the fact that $\gamma_c^l$ is independent from $\mathcal{P}_c(\hat{\mathbf{y}}^l)$ with respect to $\theta^l$, while $\beta_c^l$ is independent from $\gamma_c^l \mu_c(\hat{\mathbf{y}}^l)$ with respect to $\theta^l$, with the distribution of $\beta_c^l$ symmetric around zero.

At the same time, $\forall \alpha, c$:

$$
\begin{aligned}
\left(\check{\mathbf{y}}_{\alpha,c}^l - \mu_c(\check{\mathbf{y}}^l)\right)^2 &= \left(\gamma_c^l \hat{\mathbf{y}}_{\alpha,c}^l + \beta_c^l - \left(\gamma_c^l \mu_c(\hat{\mathbf{y}}^l) + \beta_c^l\right)\right)^2 = (\gamma_c^l)^2 \left(\hat{\mathbf{y}}_{\alpha,c}^l - \mu_c(\hat{\mathbf{y}}^l)\right)^2, \\
\mathcal{P}_c(\check{\mathbf{y}}^l) - \mathcal{P}_c^{(1)}(\check{\mathbf{y}}^l) &= (\gamma_c^l)^2 \left(\mathcal{P}_c(\hat{\mathbf{y}}^l) - \mathcal{P}_c^{(1)}(\hat{\mathbf{y}}^l)\right), \\
\mathbb{E}_{\theta^l}\left[\mathcal{P}_c(\check{\mathbf{y}}^l) - \mathcal{P}_c^{(1)}(\check{\mathbf{y}}^l)\right] &= \gamma^2 \mathbb{E}_{\theta^l}\left[\mathcal{P}_c(\hat{\mathbf{y}}^l) - \mathcal{P}_c^{(1)}(\hat{\mathbf{y}}^l)\right],
\end{aligned} \tag{52}
$$

where we used the fact that $\boldsymbol{\gamma}_c^l$ is independent from $\mathcal{P}_c(\hat{\mathbf{y}}^l) - \mathcal{P}_c^{(1)}(\hat{\mathbf{y}}^l)$ with respect to $\theta^l$.

Using Eq. (44) and Eq. (45), Eq. (51) and Eq. (52) imply that $\forall c$:

$$\mathbb{E}_{\theta^l}\left[\mathcal{P}_c(\check{\mathbf{y}}^l)\right] = \gamma^2 + \beta^2, \tag{53}$$

$$\mathbb{E}_{\theta^l}\left[\mathcal{P}_c(\check{\mathbf{y}}^l) - \mathcal{P}_c^{(1)}(\check{\mathbf{y}}^l)\right] = \gamma^2 \varrho(\hat{\mathbf{z}}^{l-1}). \tag{54}$$

Now, we may bound $\mathbb{E}_{\theta^l}\left[\mathcal{P}_c(\check{\mathbf{y}}^l)^2\right]$ using $\forall \alpha, c$:

$$(\check{\mathbf{y}}_{\alpha,c}^l)^2 = \left(\boldsymbol{\gamma}_c^l \hat{\mathbf{y}}_{\alpha,c}^l + \boldsymbol{\beta}_c^l\right)^2 \leq 2\left((\boldsymbol{\gamma}_c^l)^2(\hat{\mathbf{y}}_{\alpha,c}^l)^2 + (\boldsymbol{\beta}_c^l)^2\right),$$

$$\mathcal{P}_c(\check{\mathbf{y}}^l) \leq 2\left((\boldsymbol{\gamma}_c^l)^2\mathcal{P}_c(\hat{\mathbf{y}}^l) + (\boldsymbol{\beta}_c^l)^2\right),$$

$$\mathcal{P}_c(\check{\mathbf{y}}^l)^2 \leq 8\left((\boldsymbol{\gamma}_c^l)^4\mathcal{P}_c(\hat{\mathbf{y}}^l)^2 + (\boldsymbol{\beta}_c^l)^4\right),$$

$$\mathbb{E}_{\theta^l}\left[\mathcal{P}_c(\check{\mathbf{y}}^l)^2\right] \leq 8\left(\tilde{\gamma}^4 \mathbb{E}_{\theta^l}\left[\mathcal{P}_c(\hat{\mathbf{y}}^l)^2\right] + \tilde{\beta}^4\right), \tag{55}$$

where we defined $\tilde{\gamma} \equiv \mathbb{E}_{\theta^l}\left[(\boldsymbol{\gamma}_c^l)^4\right]^{1/4} > \gamma > 0$ and $\tilde{\beta} \equiv \mathbb{E}_{\theta^l}\left[(\boldsymbol{\beta}_c^l)^4\right]^{1/4} > \beta > 0$ the $L^4$ norms of $\boldsymbol{\gamma}_c^l \sim \nu_{\boldsymbol{\gamma}}$ and $\boldsymbol{\beta}_c^l \sim \nu_{\boldsymbol{\beta}}$, and where we used twice $(a+b)^2 \leq 2(a^2+b^2)$, $\forall a, b$.

Using Eq. (46), we then get $\forall c$:

$$\mathbb{E}_{\theta^l}\left[\mathcal{P}_c(\check{\mathbf{y}}^l)^2\right] \leq 24\tilde{\gamma}^4\tilde{\omega}^4\omega^{-4} + 8\tilde{\beta}^4. \tag{56}$$

Next, we consider $\check{\mathbf{z}}^l$. We adopt the notations $\check{\mathbf{y}}^{l,+}$, $\check{\mathbf{y}}^{l,-}$ for the positive and negative parts of $\check{\mathbf{y}}^l$ such that $\forall \alpha, c$:

$$\check{\mathbf{y}}_{\alpha,c}^{l,+} = \max(\check{\mathbf{y}}_{\alpha,c}^l, 0), \qquad\qquad \check{\mathbf{y}}_{\alpha,c}^{l,-} = \max(-\check{\mathbf{y}}_{\alpha,c}^l, 0).$$

The positive homogeneity of $\phi$ implies that $\forall \alpha, c$:

$$\check{\mathbf{z}}_{\alpha,c}^l = \phi(\check{\mathbf{y}}_{\alpha,c}^l) = \phi(1) \cdot \check{\mathbf{y}}_{\alpha,c}^{l,+} + \phi(-1) \cdot \check{\mathbf{y}}_{\alpha,c}^{l,-}, \qquad (\check{\mathbf{z}}_{\alpha,c}^l)^2 = \phi(1)^2 \cdot (\check{\mathbf{y}}_{\alpha,c}^{l,+})^2 + \phi(-1)^2 \cdot (\check{\mathbf{y}}_{\alpha,c}^{l,-})^2.$$

For any $c$, this implies for $\mu_c(\check{\mathbf{z}}^l)^2$ and $\mathcal{P}_c(\check{\mathbf{z}}^l)$ that

$$\mu_c(\check{\mathbf{z}}^l)^2 = \left(\phi(1)\mathbb{E}_{\mathbf{x},\alpha}\left[\check{\mathbf{y}}_{\alpha,c}^{l,+}\right] + \phi(-1)\mathbb{E}_{\mathbf{x},\alpha}\left[\check{\mathbf{y}}_{\alpha,c}^{l,-}\right]\right)^2$$

$$= \left(\phi(1)\mu_c(\check{\mathbf{y}}^{l,+}) + \phi(-1)\mu_c(\check{\mathbf{y}}^{l,-})\right)^2$$

$$= \phi(1)^2\mu_c(\check{\mathbf{y}}^{l,+})^2 + \phi(-1)^2\mu_c(\check{\mathbf{y}}^{l,-})^2 + 2\phi(1)\phi(-1)\mu_c(\check{\mathbf{y}}^{l,+})\mu_c(\check{\mathbf{y}}^{l,-}), \tag{57}$$

$$\mathcal{P}_c(\check{\mathbf{z}}^l) = \mathbb{E}_{\mathbf{x},\alpha}\left[\phi(1)^2(\check{\mathbf{y}}_{\alpha,c}^{l,+})^2 + \phi(-1)^2(\check{\mathbf{y}}_{\alpha,c}^{l,-})^2\right]$$

$$= \phi(1)^2\mathcal{P}_c(\check{\mathbf{y}}^{l,+}) + \phi(-1)^2\mathcal{P}_c(\check{\mathbf{y}}^{l,-}). \tag{58}$$

Now turning to $\check{\mathbf{y}}^l$, we have $\forall \alpha, c$:

$$\check{\mathbf{y}}_{\alpha,c}^l = \check{\mathbf{y}}_{\alpha,c}^{l,+} - \check{\mathbf{y}}_{\alpha,c}^{l,-}, \qquad\qquad (\check{\mathbf{y}}_{\alpha,c}^l)^2 = (\check{\mathbf{y}}_{\alpha,c}^{l,+})^2 + (\check{\mathbf{y}}_{\alpha,c}^{l,-})^2.$$

For any $c$, this implies for $\mu_c(\check{\mathbf{y}}^l)^2$ and $\mathcal{P}_c(\check{\mathbf{y}}^l)$ that

$$\mu_c(\check{\mathbf{y}}^l)^2 = \left(\mathbb{E}_{\mathbf{x},\alpha}\left[\check{\mathbf{y}}_{\alpha,c}^{l,+}\right] - \mathbb{E}_{\mathbf{x},\alpha}\left[\check{\mathbf{y}}_{\alpha,c}^{l,-}\right]\right)^2$$

$$= \left(\mu_c(\check{\mathbf{y}}^{l,+}) - \mu_c(\check{\mathbf{y}}^{l,-})\right)^2$$

$$= \mu_c(\check{\mathbf{y}}^{l,+})^2 + \mu_c(\check{\mathbf{y}}^{l,-})^2 - 2\mu_c(\check{\mathbf{y}}^{l,+})\mu_c(\check{\mathbf{y}}^{l,-}), \tag{59}$$

$$\mathcal{P}_c(\check{\mathbf{y}}^l) = \mathbb{E}_{\mathbf{x},\alpha}\left[(\check{\mathbf{y}}_{\alpha,c}^{l,+})^2 + (\check{\mathbf{y}}_{\alpha,c}^{l,-})^2\right]$$

$$= \mathcal{P}_c(\check{\mathbf{y}}^{l,+}) + \mathcal{P}_c(\check{\mathbf{y}}^{l,-}). \tag{60}$$

At this point, we note that $\hat{\mathbf{y}}^l$ and $-\hat{\mathbf{y}}^l$ have the same distribution with respect to $\theta^l$ by symmetry around zero of $\nu_{\boldsymbol{\omega}}$. From this and the symmetry around zero of $\nu_{\boldsymbol{\beta}}$, we deduce that $\check{\mathbf{y}}^l$ and $-\check{\mathbf{y}}^l$ have the same distribution with respect to $\theta^l$. In turn, this implies that $\check{\mathbf{y}}^{l,+}$ and $\check{\mathbf{y}}^{l,-}$ have the same distribution with respect to $\theta^l$.

Combined with Eq. (57) and Eq. (58), we deduce that $\forall c$:

$$\mathbb{E}_{\theta^l}\left[\mu_c(\check{\mathbf{z}}^l)^2\right] = \left(\phi(1)^2 + \phi(-1)^2\right)\mathbb{E}_{\theta^l}\left[\mu_c(\check{\mathbf{y}}^{l,+})^2\right] + 2\phi(1)\phi(-1)\mathbb{E}_{\theta^l}\left[\mu_c(\check{\mathbf{y}}^{l,+})\mu_c(\check{\mathbf{y}}^{l,-})\right]$$

$$= F_\phi \mathbb{E}_{\theta^l}\left[\mu_c(\check{\mathbf{y}}^{l,+})^2 + \mu_c(\check{\mathbf{y}}^{l,-})^2\right] + 2\phi(1)\phi(-1)\mathbb{E}_{\theta^l}\left[\mu_c(\check{\mathbf{y}}^{l,+})\mu_c(\check{\mathbf{y}}^{l,-})\right],$$

$$\mathbb{E}_{\theta^l}\left[\mathcal{P}_c(\check{\mathbf{z}}^l)\right] = \left(\phi(1)^2 + \phi(-1)^2\right)\mathbb{E}_{\theta^l}\left[\mathcal{P}_c(\check{\mathbf{y}}^{l,+})\right]$$

$$= F_\phi \mathbb{E}_{\theta^l}\left[\mathcal{P}_c(\check{\mathbf{y}}^{l,+}) + \mathcal{P}_c(\check{\mathbf{y}}^{l,-})\right],$$

where we defined $F_\phi \equiv \frac{\phi(1)^2 + \phi(-1)^2}{2} > 0$, with the strict positivity of $F_\phi$ following from the assumption that $\phi$ is nonzero.

Given $|\phi(1)\phi(-1)| \leq \frac{\phi(1)^2 + \phi(-1)^2}{2} \implies \phi(1)\phi(-1) \geq -F_\phi$, and given $\mu_c(\check{\mathbf{y}}^{l,+})\mu_c(\check{\mathbf{y}}^{l,-}) \geq 0$, we deduce that $\forall c$:

$$\mathbb{E}_{\theta^l}\left[\mu_c(\check{\mathbf{z}}^l)^2\right] \geq F_\phi \mathbb{E}_{\theta^l}\left[\mu_c(\check{\mathbf{y}}^{l,+})^2 + \mu_c(\check{\mathbf{y}}^{l,-})^2 - 2\mu_c(\check{\mathbf{y}}^{l,+})\mu_c(\check{\mathbf{y}}^{l,-})\right]$$

$$\geq F_\phi \mathbb{E}_{\theta^l}\left[\mu_c(\check{\mathbf{y}}^l)^2\right]$$

$$\geq F_\phi \mathbb{E}_{\theta^l}\left[\mathcal{P}_c^{(1)}(\check{\mathbf{y}}^l)\right], \tag{61}$$

$$\mathbb{E}_{\theta^l}\left[\mathcal{P}_c(\check{\mathbf{z}}^l)\right] = F_\phi \mathbb{E}_{\theta^l}\left[\mathcal{P}_c(\check{\mathbf{y}}^l)\right]$$

$$= F_\phi\left(\gamma^2 + \beta^2\right), \tag{62}$$

$$\mathbb{E}_{\theta^l}\left[\mathcal{P}_c(\check{\mathbf{z}}^l) - \mathcal{P}_c^{(1)}(\check{\mathbf{z}}^l)\right] = \mathbb{E}_{\theta^l}\left[\mathcal{P}_c(\check{\mathbf{z}}^l) - \mu_c(\check{\mathbf{z}}^l)^2\right]$$

$$\leq F_\phi \mathbb{E}_{\theta^l}\left[\mathcal{P}_c(\check{\mathbf{y}}^l) - \mathcal{P}_c^{(1)}(\check{\mathbf{y}}^l)\right]$$

$$\leq F_\phi \gamma^2 \varrho(\hat{\mathbf{z}}^{l-1}). \tag{63}$$

where we used Eq. (59), (60) and Eq. (53), (54).

Let us now define $\chi(\hat{\mathbf{z}}^{l-1}) \in \mathbb{R}^+$ independently of $c$ as

$$\chi(\hat{\mathbf{z}}^{l-1}) \equiv \begin{cases} \frac{\mathbb{E}_{\theta^l}\left[\mathcal{P}_c(\check{\mathbf{z}}^l) - \mathcal{P}_c^{(1)}(\check{\mathbf{z}}^l)\right]}{F_\phi \gamma^2 \varrho(\hat{\mathbf{z}}^{l-1})} & \text{if } \varrho(\hat{\mathbf{z}}^{l-1}) > 0, \\ 1 & \text{otherwise.} \end{cases}$$

We note that $\chi(\hat{\mathbf{z}}^{l-1})$ is independent of $\theta^l$ and that $\chi(\hat{\mathbf{z}}^{l-1}) \leq 1$ in general, and $\chi(\hat{\mathbf{z}}^{l-1}) = 1$ if $\phi = \text{identity}$ since the inequalities of Eq. (61) and Eq. (63) become equalities when $\phi = \text{identity}$.

Given this definition of $\chi(\hat{\mathbf{z}}^{l-1})$, we may rewrite $\mathbb{E}_{\theta^l}\left[\mathcal{P}_c(\check{\mathbf{z}}^l) - \mathcal{P}_c^{(1)}(\check{\mathbf{z}}^l)\right]$ for any $c$ as

$$\mathbb{E}_{\theta^l}\left[\mathcal{P}_c(\check{\mathbf{z}}^l) - \mathcal{P}_c^{(1)}(\check{\mathbf{z}}^l)\right] = \chi(\hat{\mathbf{z}}^{l-1})F_\phi \gamma^2 \varrho(\hat{\mathbf{z}}^{l-1}). \tag{64}$$

Now let us bound $\mathbb{E}_{\theta^l}\left[\mathcal{P}_c(\check{\mathbf{z}}^l)^2\right]$ with the goal of bounding $\text{Var}_{\theta^l}\left[\mathcal{P}_c(\check{\mathbf{z}}^l)\right]$, $\text{Var}_{\theta^l}\left[\mathcal{P}_c(\check{\mathbf{z}}^l) - \mathcal{P}_c^{(1)}(\check{\mathbf{z}}^l)\right]$. We get from Eq. (58) that $\forall c$:

$$\mathcal{P}_c(\check{\mathbf{z}}^l) \leq \left(\phi(1)^2 + \phi(-1)^2\right)\left(\mathcal{P}_c(\check{\mathbf{y}}^{l,+}) + \mathcal{P}_c(\check{\mathbf{y}}^{l,-})\right)$$

$$\leq 2F_\phi \mathcal{P}_c(\check{\mathbf{y}}^l),$$

$$\mathbb{E}_{\theta^l}\left[\mathcal{P}_c(\check{\mathbf{z}}^l)^2\right] \leq 4F_\phi^2 \mathbb{E}_{\theta^l}\left[\mathcal{P}_c(\check{\mathbf{y}}^l)^2\right]$$

$$\leq 4F_\phi^2\left(24\tilde{\gamma}^4\tilde{\omega}^4\omega^{-4} + 8\tilde{\beta}^4\right),$$

where we used Eq. (56).

This gives for $\text{Var}_{\theta^l}\Big[\mathcal{P}_c(\check{\mathbf{z}}^l)\Big]$, $\text{Var}_{\theta^l}\Big[\mathcal{P}_c(\check{\mathbf{z}}^l) - \mathcal{P}_c^{(1)}(\check{\mathbf{z}}^l)\Big]$ that

$$\text{Var}_{\theta^l}\Big[\mathcal{P}_c(\check{\mathbf{z}}^l)\Big] \leq \mathbb{E}_{\theta^l}\Big[\mathcal{P}_c(\check{\mathbf{z}}^l)^2\Big]$$
$$\leq 4F_\phi^2\Big(24\tilde{\gamma}^4\tilde{\omega}^4\omega^{-4} + 8\tilde{\beta}^4\Big), \qquad (65)$$
$$\text{Var}_{\theta^l}\Big[\mathcal{P}_c(\check{\mathbf{z}}^l) - \mathcal{P}_c^{(1)}(\check{\mathbf{z}}^l)\Big] \leq \mathbb{E}_{\theta^l}\Big[\big(\mathcal{P}_c(\check{\mathbf{z}}^l) - \mathcal{P}_c^{(1)}(\check{\mathbf{z}}^l)\big)^2\Big]$$
$$\leq \mathbb{E}_{\theta^l}\Big[\mathcal{P}_c(\check{\mathbf{z}}^l)^2\Big]$$
$$\leq 4F_\phi^2\Big(24\tilde{\gamma}^4\tilde{\omega}^4\omega^{-4} + 8\tilde{\beta}^4\Big). \qquad (66)$$

Using Eq. (62), (64),(65), (66) and the fact that the terms $\mathcal{P}_c(\check{\mathbf{z}}^l)$ and $\mathcal{P}_c(\check{\mathbf{z}}^l) - \mathcal{P}_c^{(1)}(\check{\mathbf{z}}^l)$ are i.i.d. in the different channels $c$, we get

$$\mathbb{E}_{\theta^l}\Big[\mathcal{P}(\check{\mathbf{z}}^l)\Big] = F_\phi\big(\gamma^2 + \beta^2\big), \qquad\qquad \text{Var}_{\theta^l}\Big[\mathcal{P}(\check{\mathbf{z}}^l)\Big] \leq \frac{4F_\phi^2\Big(24\tilde{\gamma}^4\tilde{\omega}^4\omega^{-4} + 8\tilde{\beta}^4\Big)}{C_l},$$
$$(67)$$

$$\mathbb{E}_{\theta^l}\Big[\mathcal{P}(\check{\mathbf{z}}^l) - \mathcal{P}^{(1)}(\check{\mathbf{z}}^l)\Big] = \chi(\hat{\mathbf{z}}^{l-1})F_\phi\gamma^2\varrho(\hat{\mathbf{z}}^{l-1}), \quad \text{Var}_{\theta^l}\Big[\mathcal{P}(\check{\mathbf{z}}^l) - \mathcal{P}^{(1)}(\check{\mathbf{z}}^l)\Big] \leq \frac{4F_\phi^2\Big(24\tilde{\gamma}^4\tilde{\omega}^4\omega^{-4} + 8\tilde{\beta}^4\Big)}{C_l}.$$

Now if we define $\check{\mathbf{z}}^l$ such that $\forall \alpha, c$: $\check{\mathbf{z}}^l_{\alpha,c} = \frac{\check{\mathbf{z}}^l_{\alpha,c}}{\sqrt{F_\phi(\gamma^2+\beta^2)}}$, we get

$$\mathbb{E}_{\theta^l}\Big[\mathcal{P}(\check{\mathbf{z}}^l)\Big] = 1, \qquad\qquad \text{Var}_{\theta^l}\Big[\mathcal{P}(\check{\mathbf{z}}^l)\Big] \leq \frac{1}{C_l}\frac{4\Big(24\tilde{\gamma}^4\tilde{\omega}^4\omega^{-4} + 8\tilde{\beta}^4\Big)}{\big(\gamma^2 + \beta^2\big)^2},$$

$$\mathbb{E}_{\theta^l}\Big[\mathcal{P}(\check{\mathbf{z}}^l) - \mathcal{P}^{(1)}(\check{\mathbf{z}}^l)\Big] = \rho\chi(\hat{\mathbf{z}}^{l-1})\varrho(\hat{\mathbf{z}}^{l-1}), \quad \text{Var}_{\theta^l}\Big[\mathcal{P}(\check{\mathbf{z}}^l) - \mathcal{P}^{(1)}(\check{\mathbf{z}}^l)\Big] \leq \frac{1}{C_l}\frac{4\Big(24\tilde{\gamma}^4\tilde{\omega}^4\omega^{-4} + 8\tilde{\beta}^4\Big)}{\big(\gamma^2 + \beta^2\big)^2},$$

where $\rho = \frac{\gamma^2}{\gamma^2+\beta^2}$.

The reasoning that yielded Eq. (50) from Eq. (48) and Eq. (49) can be immediately transposed by replacing $\hat{\mathbf{y}}^l$ by $\check{\mathbf{z}}^l$, $\varrho(\hat{\mathbf{z}}^{l-1})$ by $\rho\chi(\hat{\mathbf{z}}^{l-1})\varrho(\hat{\mathbf{z}}^{l-1})$ and $3\tilde{\omega}^4\omega^{-4}$ by $\frac{4\big(24\tilde{\gamma}^4\tilde{\omega}^4\omega^{-4}+8\tilde{\beta}^4\big)}{\big(\gamma^2+\beta^2\big)^2}$.

Consequently, for any $\eta > 0$ and any $\delta > 0$, there exists $N_3(\eta, \delta) \in \mathbb{N}^*$ independent of $\Theta^{l-1}$, $l$, such that if $C_l \geq N_3(\eta, \delta)$, it holds that

$$\mathbb{P}_{\theta_l}\left[|\varrho(\check{\mathbf{z}}^l) - \rho\chi(\hat{\mathbf{z}}^{l-1})\varrho(\hat{\mathbf{z}}^{l-1})| \leq \eta\right] \geq 1 - \delta,$$

$$\mathbb{P}_{\theta_l}\left[|\varrho(\check{\mathbf{z}}^l) - \rho\chi(\hat{\mathbf{z}}^{l-1})\varrho(\hat{\mathbf{z}}^{l-1})| \leq \eta\right] \geq 1 - \delta,$$

where we used the fact that $\varrho(\check{\mathbf{z}}^l) = \varrho(\check{\mathbf{z}}^l)$.

Let us finally define $N'$ independently of $\Theta^l$, $l$ as $N'(\eta, \delta) = \max\big(N_2(\eta, \delta), N_3(\eta, \delta)\big)$, $\forall \eta > 0$, $\forall \delta > 0$. Then for any $\eta > 0$ and any $\delta > 0$, if $C_l \geq N'(\eta, \delta)$, it holds that

$$\mathbb{P}_{\theta_l}\Big[|\varrho(\hat{\mathbf{y}}^l) - \varrho(\hat{\mathbf{z}}^{l-1})| \leq \eta\Big] \geq 1 - \delta,$$

$$\mathbb{P}_{\theta_l}\Big[|\varrho(\check{\mathbf{z}}^l) - \rho\chi(\hat{\mathbf{z}}^{l-1})\varrho(\hat{\mathbf{z}}^{l-1})| \leq \eta\Big] \geq 1 - \delta,$$

where we recall that $\chi(\hat{\mathbf{z}}^{l-1}) \leq 1$ in general, and that $\chi(\hat{\mathbf{z}}^{l-1}) = 1$ if $\phi = \text{identity}$.    $\square$

**Lemma 2.** *Fix a layer $l \geq 1$, $\nu_{\boldsymbol{\omega}}$, $\nu_{\boldsymbol{\beta}}$, $\nu_{\boldsymbol{\gamma}}$, $\mathcal{D}$ in Definition 1 and model parameters $\Theta^{l-1}$ up to layer $l-1$ such that $\mathcal{P}_{\mathbf{x}}(\mathbf{z}^{l-1}) > 0$, $\forall \mathbf{x}$. Further suppose $\mathrm{Norm} = \mathrm{LN}$ and suppose and that the convolution of Eq. (2) uses periodic boundary conditions.*

*Then for any $\eta > 0$ and any $\delta > 0$, there exists $N''(\eta, \delta) \in \mathbb{N}^*$ independent of $\Theta^{l-1}$, $l$ such that if $C_l \geq N''(\eta, \delta)$, it holds for random nets of Definition 1 with probability greater than $1 - \delta$ with respect to $\theta^l$ that*

$$|\mathcal{P}^{(1)}(\hat{\mathbf{y}}^l) - \mathcal{P}^{(1)}(\mathbf{y}^l)| \leq \eta, \qquad\qquad |\mathcal{P}^{(1)}(\check{\mathbf{z}}^l) - \mathcal{P}^{(1)}(\mathbf{z}^l)| \leq \eta,$$
$$|\mathcal{P}_{\mathbf{x}}(\hat{\mathbf{y}}^l) - \mathcal{P}_{\mathbf{x}}(\mathbf{y}^l)| \leq \eta, \quad \forall \mathbf{x} \in \mathcal{D}, \qquad |\mathcal{P}_{\mathbf{x}}(\check{\mathbf{z}}^l) - \mathcal{P}_{\mathbf{x}}(\mathbf{z}^l)| \leq \eta, \quad \forall \mathbf{x} \in \mathcal{D},$$
$$|\mathcal{P}_{\mathbf{x}}(\hat{\mathbf{y}}^l) - 1| \leq \eta, \quad \forall \mathbf{x} \in \mathcal{D}, \qquad \left|\mathcal{P}_{\mathbf{x}}(\check{\mathbf{z}}^l) - F_\phi(\gamma^2 + \beta^2)\right| \leq \eta, \quad \forall \mathbf{x} \in \mathcal{D},$$

*where $F_\phi \equiv \frac{\phi(1)^2 + \phi(-1)^2}{2} > 0$ and $\hat{\mathbf{y}}^l$, $\check{\mathbf{z}}^l$ are defined in Eq. (30) and Eq. (32).*

**Proof.** Let us start by noting that $\forall \alpha, c$:

$$\hat{\mathbf{x}}_{\alpha,c}^l = \sqrt{\frac{\mathcal{P}(\mathbf{z}^{l-1})}{\mathcal{P}_{\mathbf{x}}(\mathbf{z}^{l-1})}} \mathbf{x}_{\alpha,c}^l, \qquad \hat{\mathbf{y}}_{\alpha,c}^l = \frac{1}{\omega\sqrt{\mathcal{P}(\mathbf{z}^{l-1})}} \hat{\mathbf{x}}_{\alpha,c}^l = \frac{1}{\omega\sqrt{\mathcal{P}_{\mathbf{x}}(\mathbf{z}^{l-1})}} \mathbf{x}_{\alpha,c}^l.$$

This implies that $\hat{\mathbf{y}}^l$ only depends on $\mathbf{x}$ and not on other inputs in the dataset.

Thus, Eq. (47), (48), (67) still hold when considering the moments conditioned on $\mathbf{x}$, such that $\forall \mathbf{x}$:

$$\mathbb{E}_{\theta^l}\left[\mu_{\mathbf{x}}(\hat{\mathbf{y}}^l)\right] = 0, \qquad\qquad \mathrm{Var}_{\theta^l}\left[\mu_{\mathbf{x}}(\hat{\mathbf{y}}^l)\right] \leq \frac{1}{C_l}, \tag{68}$$

$$\mathbb{E}_{\theta^l}\left[\mathcal{P}_{\mathbf{x}}(\hat{\mathbf{y}}^l)\right] = 1, \qquad\qquad \mathrm{Var}_{\theta^l}\left[\mathcal{P}_{\mathbf{x}}(\hat{\mathbf{y}}^l)\right] \leq \frac{3\tilde{\omega}^4 \omega^{-4}}{C_l}, \tag{69}$$

$$\mathbb{E}_{\theta^l}\left[\mathcal{P}_{\mathbf{x}}(\check{\mathbf{z}}^l)\right] = F_\phi(\gamma^2 + \beta^2), \qquad \mathrm{Var}_{\theta^l}\left[\mathcal{P}_{\mathbf{x}}(\check{\mathbf{z}}^l)\right] \leq \frac{4F_\phi^2\left(24\tilde{\gamma}^4\tilde{\omega}^4\omega^{-4} + 8\tilde{\beta}^4\right)}{C_l}. \tag{70}$$

Combining Eq. (68), (69), (70) with Chebyshev's inequality, we get for any $\eta > 0$ and any $\delta > 0$ that there exists $N_4(\eta, \delta) \in \mathbb{N}^*$ independent of $\Theta^{l-1}$, $l$ such that, if $C_l \geq N_4(\eta, \delta)$, it holds for any $\mathbf{x}$ that

$$\mathbb{P}_{\theta^l}\left[|\mu_{\mathbf{x}}(\hat{\mathbf{y}}^l)| \leq \eta\right] \geq 1 - \delta, \tag{71}$$

$$\mathbb{P}_{\theta^l}\left[|\mathcal{P}_{\mathbf{x}}(\hat{\mathbf{y}}^l) - 1| \leq \eta\right] \geq 1 - \delta, \tag{72}$$

$$\mathbb{P}_{\theta^l}\left[\left|\mathcal{P}_{\mathbf{x}}(\check{\mathbf{z}}^l) - F_\phi(\gamma^2 + \beta^2)\right| \leq \eta\right] \geq 1 - \delta. \tag{73}$$

Next we turn to $\mathcal{P}_{\mathbf{x}}(\hat{\mathbf{y}}^l - \mathbf{y}^l)$, $\mathcal{P}_{\mathbf{x}}(\check{\mathbf{z}}^l - \mathbf{z}^l)$. Given that $\mathbf{y}_{\alpha,c}^l = \frac{\mathbf{x}_{\alpha,c}^l - \mu_{\mathbf{x}}(\mathbf{x}^l)}{\sigma_{\mathbf{x}}(\mathbf{x}^l)} = \frac{\hat{\mathbf{y}}_{\alpha,c}^l - \mu_{\mathbf{x}}(\hat{\mathbf{y}}^l)}{\sigma_{\mathbf{x}}(\hat{\mathbf{y}}^l)}$, $\forall \mathbf{x}, \alpha, c$, we deduce $\forall \mathbf{x}, \alpha, c$:

$$\hat{\mathbf{y}}_{\alpha,c}^l = \sigma_{\mathbf{x}}(\hat{\mathbf{y}}^l)\mathbf{y}_{\alpha,c}^l + \mu_{\mathbf{x}}(\hat{\mathbf{y}}^l)$$
$$= \mathbf{y}_{\alpha,c}^l + (\sigma_{\mathbf{x}}(\hat{\mathbf{y}}^l) - 1)\mathbf{y}_{\alpha,c}^l + \mu_{\mathbf{x}}(\hat{\mathbf{y}}^l),$$
$$\check{\mathbf{y}}_{\alpha,c}^l = \gamma_c^l \mathbf{y}_{\alpha,c}^l + \beta_c^l + \gamma_c^l(\sigma_{\mathbf{x}}(\hat{\mathbf{y}}^l) - 1)\mathbf{y}_{\alpha,c}^l + \gamma_c^l\mu_{\mathbf{x}}(\hat{\mathbf{y}}^l)$$
$$= \tilde{\mathbf{y}}_{\alpha,c}^l + \gamma_c^l(\sigma_{\mathbf{x}}(\hat{\mathbf{y}}^l) - 1)\mathbf{y}_{\alpha,c}^l + \gamma_c^l\mu_{\mathbf{x}}(\hat{\mathbf{y}}^l).$$

Now let us fix $\mathbf{x}$ and bound $\mathcal{P}_{\mathbf{x}}(\hat{\mathbf{y}}^l - \mathbf{y}^l)$ and $\mathcal{P}_{\mathbf{x}}(\check{\mathbf{z}}^l - \mathbf{z}^l)$. We start by noting that

$$(\hat{\mathbf{y}}_{\alpha,c}^l - \mathbf{y}_{\alpha,c}^l)^2 \leq 2(\sigma_{\mathbf{x}}(\hat{\mathbf{y}}^l) - 1)^2(\mathbf{y}_{\alpha,c}^l)^2 + 2\mu_{\mathbf{x}}(\hat{\mathbf{y}}^l)^2,$$
$$(\check{\mathbf{y}}_{\alpha,c}^l - \tilde{\mathbf{y}}_{\alpha,c}^l)^2 \leq 2(\sigma_{\mathbf{x}}(\hat{\mathbf{y}}^l) - 1)^2(\gamma_c^l\mathbf{y}_{\alpha,c}^l)^2 + 2\mu_{\mathbf{x}}(\hat{\mathbf{y}}^l)^2(\gamma_c^l)^2,$$
$$(\check{\mathbf{z}}_{\alpha,c}^l - \mathbf{z}_{\alpha,c}^l)^2 = \left(\phi(\check{\mathbf{y}}_{\alpha,c}^l) - \phi(\tilde{\mathbf{y}}_{\alpha,c}^l)\right)^2$$
$$\leq 2F_\phi(\check{\mathbf{y}}_{\alpha,c}^l - \tilde{\mathbf{y}}_{\alpha,c}^l)^2$$
$$\leq 4F_\phi(\sigma_{\mathbf{x}}(\hat{\mathbf{y}}^l) - 1)^2(\gamma_c^l\mathbf{y}_{\alpha,c}^l)^2 + 4F_\phi\mu_{\mathbf{x}}(\hat{\mathbf{y}}^l)^2(\gamma_c^l)^2,$$

where we used $(a+b)^2 \leq 2a^2 + 2b^2$, $\forall a, b$ and $\big(\phi(a) - \phi(b)\big)^2 \leq (F'_\phi)^2(a-b)^2 \leq 2F_\phi(a-b)^2$, $\forall a, b$, by $F'_\phi$-Lipschitzness of $\phi$ with $F'_\phi = \max\big(|\phi(1)|, |\phi(-1)|\big) \leq \sqrt{2F_\phi}$.

We deduce for $\mathcal{P}_{\mathbf{x}}(\hat{\mathbf{y}}^l - \mathbf{y}^l)$ and $\mathcal{P}_{\mathbf{x}}(\check{\mathbf{z}}^l - \mathbf{z}^l)$ that

$$\mathcal{P}_{\mathbf{x}}(\hat{\mathbf{y}}^l - \mathbf{y}^l) \leq 2(\sigma_{\mathbf{x}}(\hat{\mathbf{y}}^l) - 1)^2 \mathcal{P}_{\mathbf{x}}(\mathbf{y}^l) + 2\mu_{\mathbf{x}}(\hat{\mathbf{y}}^l)^2$$
$$\leq 2\Big((\sigma_{\mathbf{x}}(\hat{\mathbf{y}}^l) - 1)^2 + \mu_{\mathbf{x}}(\hat{\mathbf{y}}^l)^2\Big), \tag{74}$$

$$\mathcal{P}_{\mathbf{x}}(\check{\mathbf{z}}^l - \mathbf{z}^l) \leq 4F_\phi(\sigma_{\mathbf{x}}(\hat{\mathbf{y}}^l) - 1)^2 \mathbb{E}_c\Big[(\gamma_c^l)^2 \mathcal{P}_{\mathbf{x},c}(\mathbf{y}^l)\Big] + 4F_\phi \mu_{\mathbf{x}}(\hat{\mathbf{y}}^l)^2 \mathbb{E}_c\Big[(\gamma_c^l)^2\Big]$$
$$\leq 4F_\phi\Big((\sigma_{\mathbf{x}}(\hat{\mathbf{y}}^l) - 1)^2 + \mu_{\mathbf{x}}(\hat{\mathbf{y}}^l)^2\Big) \mathbb{E}_c\Big[(\gamma_c^l)^2\big(\mathcal{P}_{\mathbf{x},c}(\mathbf{y}^l) + 1\big)\Big], \tag{75}$$

where we used $\mathcal{P}_{\mathbf{x}}(\mathbf{y}^l) \leq 1$.

Next, let us bound the expectation over $\theta^l$ of $\mathbb{E}_c\Big[(\gamma_c^l)^2\big(\mathcal{P}_{\mathbf{x},c}(\mathbf{y}^l) + 1\big)\Big]$:

$$\mathbb{E}_{\theta^l}\Big[\mathbb{E}_c\Big[(\gamma_c^l)^2\big(\mathcal{P}_{\mathbf{x},c}(\mathbf{y}^l) + 1\big)\Big]\Big] = \mathbb{E}_c\Big[\mathbb{E}_{\theta^l}\Big[(\gamma_c^l)^2\big(\mathcal{P}_{\mathbf{x},c}(\mathbf{y}^l) + 1\big)\Big]\Big]$$
$$= \mathbb{E}_c\Big[\mathbb{E}_{\theta^l}\Big[(\gamma_c^l)^2\Big]\mathbb{E}_{\theta^l}\Big[\mathcal{P}_{\mathbf{x},c}(\mathbf{y}^l) + 1\Big]\Big]$$
$$= \gamma^2 \mathbb{E}_{\theta^l}\Big[\mathbb{E}_c\Big[\mathcal{P}_{\mathbf{x},c}(\mathbf{y}^l) + 1\Big]\Big]$$
$$= \gamma^2 \mathbb{E}_{\theta^l}\Big[\mathcal{P}_{\mathbf{x}}(\mathbf{y}^l) + 1\Big]$$
$$\leq 2\gamma^2,$$

where we used the independence of $\gamma_c^l$ and $\mathcal{P}_{\mathbf{x},c}(\mathbf{y}^l)$ with respect to $\theta^l$ for any $c$, and again $\mathcal{P}_{\mathbf{x}}(\mathbf{y}^l) \leq 1$.

Markov's inequality then gives for any $\delta > 0$ that

$$\mathbb{P}_{\theta^l}\left[\mathbb{E}_c\Big[(\gamma_c^l)^2\big(\mathcal{P}_{\mathbf{x},c}(\mathbf{y}^l) + 1\big)\Big] \geq \frac{2\gamma^2}{\delta}\right] \leq 2\gamma^2 \frac{\delta}{2\gamma^2} = \delta.$$

Thus, for any $1 \geq \eta > 0$ and any $\delta > 0$, if $C_l \geq N_4(\eta, \delta)$, it holds for any $\mathbf{x}$ with probability greater than $1 - 4\delta$ with respect to $\theta^l$ that

$$|\mu_{\mathbf{x}}(\hat{\mathbf{y}}^l)| \leq \eta, \tag{76}$$
$$|\mathcal{P}_{\mathbf{x}}(\hat{\mathbf{y}}^l) - 1| \leq \eta, \tag{77}$$
$$\big|\mathcal{P}_{\mathbf{x}}(\check{\mathbf{z}}^l) - F_\phi(\gamma^2 + \beta^2)\big| \leq \eta, \tag{78}$$
$$\mathbb{E}_c\Big[(\gamma_c^l)^2\big(\mathcal{P}_{\mathbf{x},c}(\mathbf{y}^l) + 1\big)\Big] \leq \frac{2\gamma^2}{\delta}. \tag{79}$$

If both inequalities of Eq. (76) and Eq. (77) hold with $1 \geq \eta > 0$, then

$$|\sigma_{\mathbf{x}}(\hat{\mathbf{y}}^l) - 1| \leq |\sigma_{\mathbf{x}}(\hat{\mathbf{y}}^l) - 1||\sigma_{\mathbf{x}}(\hat{\mathbf{y}}^l) + 1| = |\sigma_{\mathbf{x}}(\hat{\mathbf{y}}^l)^2 - 1|$$
$$\leq |\mathcal{P}_{\mathbf{x}}(\hat{\mathbf{y}}^l) - 1| + \mu_{\mathbf{x}}(\hat{\mathbf{y}}^l)^2$$
$$\leq \eta + \eta^2 \leq 2\eta,$$
$$(\sigma_{\mathbf{x}}(\hat{\mathbf{y}}^l) - 1)^2 \leq 4\eta^2 \leq 4\eta,$$
$$\mu_{\mathbf{x}}(\hat{\mathbf{y}}^l)^2 \leq \eta^2 \leq \eta,$$

where we used $\eta^2 \leq \eta$ for $1 \geq \eta > 0$.

Injecting this into Eq. (74) and Eq. (75), we get that, if $C_l \geq N_4(\eta, \delta)$ with $1 \geq \eta > 0$, it holds for any $\mathbf{x}$ with probability greater than $1 - 4\delta$ with respect to $\theta^l$ that

$$|\mathcal{P}_{\mathbf{x}}(\hat{\mathbf{y}}^l) - 1| \leq \eta,$$
$$\mathcal{P}_{\mathbf{x}}(\hat{\mathbf{y}}^l - \mathbf{y}^l) \leq 10\eta,$$
$$\big|\mathcal{P}_{\mathbf{x}}(\check{\mathbf{z}}^l) - F_\phi(\gamma^2 + \beta^2)\big| \leq \eta,$$
$$\mathcal{P}_{\mathbf{x}}(\check{\mathbf{z}}^l - \mathbf{z}^l) \leq 20F_\phi \eta \frac{2\gamma^2}{\delta}.$$

Let us define $N_5$ independently of $\Theta^{l-1}$, $l$ as $N_5(\eta, \delta) = N_4\Big(\min(\frac{\eta}{10}, \frac{\eta}{20F_\phi}\frac{\delta}{2\gamma^2}, 1), \frac{\delta}{4|\mathcal{D}|}\Big)$, $\forall \eta > 0$, $\forall \delta > 0$. Then $\forall \eta > 0$, $\forall \delta > 0$ if $C_l \geq N_5(\eta, \delta)$, it holds with probability greater than $1 - \delta$ with respect to $\theta^l$ that

$$|\mathcal{P}_\mathbf{x}(\hat{\mathbf{y}}^l) - 1| \leq \eta, \quad \forall \mathbf{x} \in \mathcal{D}, \qquad \big|\mathcal{P}_\mathbf{x}(\check{\mathbf{z}}^l) - F_\phi(\gamma^2 + \beta^2)\big| \leq \eta, \quad \forall \mathbf{x} \in \mathcal{D}, \qquad (80)$$

$$\mathcal{P}_\mathbf{x}(\hat{\mathbf{y}}^l - \mathbf{y}^l) \leq \eta, \quad \forall \mathbf{x} \in \mathcal{D}, \qquad\qquad \mathcal{P}_\mathbf{x}(\check{\mathbf{z}}^l - \mathbf{z}^l) \leq \eta, \quad \forall \mathbf{x} \in \mathcal{D}. \qquad (81)$$

If Eq. (80) and Eq. (81) hold, then

$$
\begin{aligned}
\big|\mathcal{P}^{(1)}(\hat{\mathbf{y}}^l) - \mathcal{P}^{(1)}(\mathbf{y}^l)\big| &= \big|\mathcal{P}^{(1)}(\hat{\mathbf{y}}^l) - \mathcal{P}^{(1)}(\hat{\mathbf{y}}^l + \mathbf{y}^l - \hat{\mathbf{y}}^l)\big| \\
&\leq \mathcal{P}^{(1)}(\mathbf{y}^l - \hat{\mathbf{y}}^l) + 2\Big|\mathbb{E}_c\Big[\mu_c(\hat{\mathbf{y}}^l)\mu_c(\mathbf{y}^l - \hat{\mathbf{y}}^l)\Big]\Big| \\
&\leq \mathcal{P}^{(1)}(\mathbf{y}^l - \hat{\mathbf{y}}^l) + 2\mathbb{E}_c\Big[|\mu_c(\hat{\mathbf{y}}^l)||\mu_c(\mathbf{y}^l - \hat{\mathbf{y}}^l)|\Big] \\
&\leq \mathcal{P}^{(1)}(\mathbf{y}^l - \hat{\mathbf{y}}^l) + 2\sqrt{\mathbb{E}_c\Big[\mu_c(\hat{\mathbf{y}}^l)^2\Big]\mathbb{E}_c\Big[\mu_c(\mathbf{y}^l - \hat{\mathbf{y}}^l)^2\Big]} \\
&\leq \mathcal{P}(\mathbf{y}^l - \hat{\mathbf{y}}^l) + 2\sqrt{\mathcal{P}(\hat{\mathbf{y}}^l)\mathcal{P}(\mathbf{y}^l - \hat{\mathbf{y}}^l)} \\
&\leq \mathbb{E}_\mathbf{x}\Big[\mathcal{P}_\mathbf{x}(\mathbf{y}^l - \hat{\mathbf{y}}^l)\Big] + 2\sqrt{\mathbb{E}_\mathbf{x}\Big[\mathcal{P}_\mathbf{x}(\hat{\mathbf{y}}^l)\Big]\mathbb{E}_\mathbf{x}\Big[\mathcal{P}_\mathbf{x}(\mathbf{y}^l - \hat{\mathbf{y}}^l)\Big]} \\
&\leq \eta + 2\sqrt{(1 + \eta)\eta},
\end{aligned}
$$

where we used $\mathcal{P}^{(1)}(\mathbf{y}^l - \hat{\mathbf{y}}^l) \leq \mathcal{P}(\mathbf{y}^l - \hat{\mathbf{y}}^l)$ and $\mathcal{P}^{(1)}(\hat{\mathbf{y}}^l) \leq \mathcal{P}(\hat{\mathbf{y}}^l)$, as well as Jensen's inequality and Cauchy-Schwartz inequality.

Similarly, if Eq. (80) and Eq. (81) hold, then $\forall \mathbf{x}$:

$$
\begin{aligned}
\big|\mathcal{P}_\mathbf{x}(\hat{\mathbf{y}}^l) - \mathcal{P}_\mathbf{x}(\mathbf{y}^l)\big| &= \big|\mathcal{P}_\mathbf{x}(\hat{\mathbf{y}}^l) - \mathcal{P}_\mathbf{x}(\hat{\mathbf{y}}^l + \mathbf{y}^l - \hat{\mathbf{y}}^l)\big| \\
&\leq \mathcal{P}_\mathbf{x}(\mathbf{y}^l - \hat{\mathbf{y}}^l) + 2\mu_\mathbf{x}(|\hat{\mathbf{y}}^l||\mathbf{y}^l - \hat{\mathbf{y}}^l|) \\
&\leq \mathcal{P}_\mathbf{x}(\mathbf{y}^l - \hat{\mathbf{y}}^l) + 2\sqrt{\mathcal{P}_\mathbf{x}(\hat{\mathbf{y}}^l)\mathcal{P}_\mathbf{x}(\mathbf{y}^l - \hat{\mathbf{y}}^l)} \\
&\leq \eta + 2\sqrt{(1 + \eta)\eta}.
\end{aligned}
$$

A similar calculation with $\check{\mathbf{z}}^l$, $\mathbf{z}^l$ shows that if Eq. (80) and Eq. (81) hold, then

$$\big|\mathcal{P}^{(1)}(\check{\mathbf{z}}^l) - \mathcal{P}^{(1)}(\mathbf{z}^l)\big| \leq \eta + 2\sqrt{\eta}\sqrt{F_\phi(\gamma^2 + \beta^2) + \eta},$$

$$\big|\mathcal{P}_\mathbf{x}(\check{\mathbf{z}}^l) - \mathcal{P}_\mathbf{x}(\mathbf{z}^l)\big| \leq \eta + 2\sqrt{\eta}\sqrt{F_\phi(\gamma^2 + \beta^2) + \eta}, \quad \forall \mathbf{x}.$$

Given that the three terms $\eta$, $\eta + 2\sqrt{(1 + \eta)\eta}$ and $\eta + 2\sqrt{\eta}\sqrt{F_\phi(\gamma^2 + \beta^2) + \eta}$ converge to 0 as $\eta \to 0$, it follows that there exists a mapping $h$ such that for any $\eta > 0$: $h(\eta) > 0$ and

$$h(\eta) \leq \eta,$$
$$h(\eta) + 2\sqrt{(1 + h(\eta))h(\eta)} \leq \eta,$$
$$h(\eta) + 2\sqrt{h(\eta)}\sqrt{F_\phi(\gamma^2 + \beta^2) + h(\eta)} \leq \eta.$$

Let us finally define $N''$ independently of $\Theta^l$, $l$ as $N''(\eta, \delta) = N_5(h(\eta), \delta)$, $\forall \eta > 0$, $\forall \delta > 0$. Then $\forall \eta > 0$, $\forall \delta > 0$, if $C_l \geq N''(\eta, \delta)$, it holds with probability greater than $1 - \delta$ with respect to $\theta^l$ that

$$|\mathcal{P}^{(1)}(\hat{\mathbf{y}}^l) - \mathcal{P}^{(1)}(\mathbf{y}^l)| \leq \eta, \qquad\qquad |\mathcal{P}^{(1)}(\check{\mathbf{z}}^l) - \mathcal{P}^{(1)}(\mathbf{z}^l)| \leq \eta,$$

$$|\mathcal{P}_\mathbf{x}(\hat{\mathbf{y}}^l) - \mathcal{P}_\mathbf{x}(\mathbf{y}^l)| \leq \eta, \quad \forall \mathbf{x} \in \mathcal{D}, \qquad |\mathcal{P}_\mathbf{x}(\check{\mathbf{z}}^l) - \mathcal{P}_\mathbf{x}(\mathbf{z}^l)| \leq \eta, \quad \forall \mathbf{x} \in \mathcal{D},$$

$$|\mathcal{P}_\mathbf{x}(\hat{\mathbf{y}}^l) - 1| \leq \eta, \quad \forall \mathbf{x} \in \mathcal{D}, \qquad \big|\mathcal{P}_\mathbf{x}(\check{\mathbf{z}}^l) - F_\phi(\gamma^2 + \beta^2)\big| \leq \eta, \quad \forall \mathbf{x} \in \mathcal{D}. \qquad \square$$

**Lemma 3.** *Fix a layer $l \geq 1$, $\nu_{\boldsymbol{\omega}}$, $\nu_{\boldsymbol{\beta}}$, $\nu_{\boldsymbol{\gamma}}$, $\mathcal{D}$ in Definition 1 and model parameters $\Theta^{l-1}$ up to layer $l-1$ such that $\mathcal{P}_{\mathbf{x}}(\mathbf{z}^{l-1}) > 0$, $\forall \mathbf{x}$. Further suppose* Norm = LN *and suppose that the convolution of Eq. (2) uses periodic boundary conditions.*

*Then for any $\eta > 0$ and any $\delta > 0$, there exists $N'''(\eta, \delta) \in \mathbb{N}^*$ independent of $\Theta^{l-1}$, $l$ such that if $C_l \geq N'''(\eta, \delta)$, it holds for random nets of Definition 1 with probability greater than $1 - \delta$ with respect to $\theta^l$ that*

$$|\varrho(\mathbf{y}^l) - \varrho(\hat{\mathbf{z}}^{l-1})| \leq \eta, \qquad\qquad |\varrho(\mathbf{z}^l) - \rho\chi(\hat{\mathbf{z}}^{l-1})\varrho(\hat{\mathbf{z}}^{l-1})| \leq \eta,$$

$$|\mathcal{P}_{\mathbf{x}}(\mathbf{y}^l) - 1| \leq \eta, \quad \forall \mathbf{x} \in \mathcal{D}, \qquad \left|\mathcal{P}_{\mathbf{x}}(\mathbf{z}^l) - F_\phi(\gamma^2 + \beta^2)\right| \leq \eta, \quad \forall \mathbf{x} \in \mathcal{D},$$

*where $F_\phi \equiv \frac{\phi(1)^2 + \phi(-1)^2}{2} > 0$, $\rho = \frac{\gamma^2}{\gamma^2 + \beta^2} < 1$ and $\chi(\hat{\mathbf{z}}^{l-1}) \in \mathbb{R}^+$ is dependent on $\Theta^{l-1}$ but independent of $\theta^l$ such that $\chi(\hat{\mathbf{z}}^{l-1}) \leq 1$ in general and $\chi(\hat{\mathbf{z}}^{l-1}) = 1$ if $\phi = $ identity.*

**Proof.** First let us note that

$$|\mathcal{P}_{\mathbf{x}}(\hat{\mathbf{y}}^l) - \mathcal{P}_{\mathbf{x}}(\mathbf{y}^l)| \leq \eta, \quad \forall \mathbf{x} \in \mathcal{D} \qquad \Longrightarrow \qquad |\mathcal{P}(\hat{\mathbf{y}}^l) - \mathcal{P}(\mathbf{y}^l)| \leq \eta,$$

$$|\mathcal{P}_{\mathbf{x}}(\hat{\mathbf{y}}^l) - 1| \leq \eta, \quad \forall \mathbf{x} \in \mathcal{D} \qquad \Longrightarrow \qquad |\mathcal{P}(\hat{\mathbf{y}}^l) - 1| \leq \eta,$$

$$|\mathcal{P}_{\mathbf{x}}(\check{\mathbf{z}}^l) - \mathcal{P}_{\mathbf{x}}(\mathbf{z}^l)| \leq \eta, \quad \forall \mathbf{x} \in \mathcal{D} \qquad \Longrightarrow \qquad |\mathcal{P}(\check{\mathbf{z}}^l) - \mathcal{P}(\mathbf{z}^l)| \leq \eta,$$

$$\left|\mathcal{P}_{\mathbf{x}}(\check{\mathbf{z}}^l) - F_\phi(\gamma^2 + \beta^2)\right| \leq \eta, \quad \forall \mathbf{x} \in \mathcal{D} \qquad \Longrightarrow \qquad \left|\mathcal{P}(\check{\mathbf{z}}^l) - F_\phi(\gamma^2 + \beta^2)\right| \leq \eta.$$

Combined with Lemma 2, we deduce for any $\eta > 0$ and any $\delta > 0$ that there exists $N''(\eta, \delta) \in \mathbb{N}^*$ independent of $\Theta^{l-1}$, $l$ such that if $C_l \geq N''(\eta, \delta)$, it holds with probability greater than $1 - \delta$ with respect to $\theta^l$ that

$$|\mathcal{P}^{(1)}(\hat{\mathbf{y}}^l) - \mathcal{P}^{(1)}(\mathbf{y}^l)| \leq \eta, \qquad\qquad |\mathcal{P}^{(1)}(\check{\mathbf{z}}^l) - \mathcal{P}^{(1)}(\mathbf{z}^l)| \leq \eta, \qquad (82)$$

$$|\mathcal{P}(\hat{\mathbf{y}}^l) - \mathcal{P}(\mathbf{y}^l)| \leq \eta, \qquad\qquad |\mathcal{P}(\check{\mathbf{z}}^l) - \mathcal{P}(\mathbf{z}^l)| \leq \eta, \qquad (83)$$

$$|\mathcal{P}(\hat{\mathbf{y}}^l) - 1| \leq \eta, \qquad\qquad \left|\mathcal{P}(\check{\mathbf{z}}^l) - F_\phi(\gamma^2 + \beta^2)\right| \leq \eta, \qquad (84)$$

where $F_\phi \equiv \frac{\phi(1)^2 + \phi(-1)^2}{2} > 0$.

If all inequalities of Eq. (82), (83), (84) hold with $\eta \leq \frac{1}{4}F_\phi(\gamma^2 + \beta^2)$, then $\varrho(\mathbf{z}^l) - \varrho(\check{\mathbf{z}}^l)$ may be upper bounded using

$$\mathcal{P}(\mathbf{z}^l) - \mathcal{P}^{(1)}(\mathbf{z}^l) \leq \mathcal{P}(\check{\mathbf{z}}^l) - \mathcal{P}^{(1)}(\check{\mathbf{z}}^l) + |\mathcal{P}(\mathbf{z}^l) - \mathcal{P}(\check{\mathbf{z}}^l)| + |\mathcal{P}^{(1)}(\check{\mathbf{z}}^l) - \mathcal{P}^{(1)}(\mathbf{z}^l)|$$

$$\leq \varrho(\check{\mathbf{z}}^l)\mathcal{P}(\check{\mathbf{z}}^l) + 2\eta$$

$$\leq \varrho(\check{\mathbf{z}}^l)\left(F_\phi(\gamma^2 + \beta^2) + \eta\right) + 2\eta$$

$$\leq \varrho(\check{\mathbf{z}}^l)F_\phi(\gamma^2 + \beta^2) + 3\eta,$$

$$\mathcal{P}(\mathbf{z}^l) \geq \mathcal{P}(\check{\mathbf{z}}^l) - \eta$$

$$\geq F_\phi(\gamma^2 + \beta^2) - 2\eta,$$

$$\varrho(\mathbf{z}^l) \leq \frac{\varrho(\check{\mathbf{z}}^l)F_\phi(\gamma^2 + \beta^2) + 3\eta}{F_\phi(\gamma^2 + \beta^2) - 2\eta}$$

$$\leq \frac{\varrho(\check{\mathbf{z}}^l) + \frac{3\eta}{F_\phi(\gamma^2 + \beta^2)}}{1 - \frac{2\eta}{F_\phi(\gamma^2 + \beta^2)}}$$

$$\leq \left(\varrho(\check{\mathbf{z}}^l) + \frac{3\eta}{F_\phi(\gamma^2 + \beta^2)}\right)\left(1 + \frac{8\eta}{F_\phi(\gamma^2 + \beta^2)}\right)$$

$$\leq \varrho(\check{\mathbf{z}}^l) + \frac{35\eta}{F_\phi(\gamma^2 + \beta^2)},$$

where we used $\varrho(\check{\mathbf{z}}^l) \leq 1$, as well as $\frac{1}{1-x} \leq 1 + 4x$ for $x \leq \frac{1}{2}$ and $\left(\frac{\eta}{F_\phi(\gamma^2 + \beta^2)}\right)^2 \leq \frac{\eta}{F_\phi(\gamma^2 + \beta^2)}$ for $\frac{\eta}{F_\phi(\gamma^2 + \beta^2)} \leq 1$.

Similarly, if all inequalities of Eq. (82), (83), (84) hold with $\eta \leq \frac{1}{4} F_\phi(\gamma^2 + \beta^2)$, then $\varrho(\mathbf{z}^l) - \varrho(\check{\mathbf{z}}^l)$ may be lower bounded using

$$
\begin{aligned}
\mathcal{P}(\mathbf{z}^l) - \mathcal{P}^{(1)}(\mathbf{z}^l) &\geq \mathcal{P}(\check{\mathbf{z}}^l) - \mathcal{P}^{(1)}(\check{\mathbf{z}}^l) - |\mathcal{P}(\mathbf{z}^l) - \mathcal{P}(\check{\mathbf{z}}^l)| - |\mathcal{P}^{(1)}(\check{\mathbf{z}}^l) - \mathcal{P}^{(1)}(\mathbf{z}^l)| \\
&\geq \varrho(\check{\mathbf{z}}^l)\mathcal{P}(\check{\mathbf{z}}^l) - 2\eta \\
&\geq \varrho(\check{\mathbf{z}}^l)\big(F_\phi(\gamma^2 + \beta^2) - \eta\big) - 2\eta \\
&\geq \varrho(\check{\mathbf{z}}^l)F_\phi(\gamma^2 + \beta^2) - 3\eta, \\
\mathcal{P}(\mathbf{z}^l) &\leq \mathcal{P}(\check{\mathbf{z}}^l) + \eta \leq F_\phi(\gamma^2 + \beta^2) + 2\eta, \\
\varrho(\mathbf{z}^l) &\geq \frac{\max\big(\varrho(\check{\mathbf{z}}^l)F_\phi(\gamma^2 + \beta^2) - 3\eta, 0\big)}{F_\phi(\gamma^2 + \beta^2) + 2\eta} \\
&\geq \frac{\max\big(\varrho(\check{\mathbf{z}}^l) - \frac{3\eta}{F_\phi(\gamma^2+\beta^2)}, 0\big)}{1 + \frac{2\eta}{F_\phi(\gamma^2+\beta^2)}} \\
&\geq \max\left(\Big(\varrho(\check{\mathbf{z}}^l) - \frac{3\eta}{F_\phi(\gamma^2 + \beta^2)}\Big)\Big(1 - \frac{2\eta}{F_\phi(\gamma^2 + \beta^2)}\Big), 0\right) \\
&\geq \varrho(\check{\mathbf{z}}^l) - \frac{5\eta}{F_\phi(\gamma^2 + \beta^2)},
\end{aligned}
$$

where we used $\varrho(\check{\mathbf{z}}^l) \leq 1$, as well as $\frac{1}{1+x} \geq 1 - x \geq 0$ for $0 \leq x \leq 1$.

We deduce that if all inequalities of Eq. (82), (83), (84) hold with $\eta \leq \frac{1}{4} F_\phi(\gamma^2 + \beta^2)$, then

$$
|\varrho(\mathbf{z}^l) - \varrho(\check{\mathbf{z}}^l)| \leq \frac{35\eta}{F_\phi(\gamma^2 + \beta^2)}. \tag{85}
$$

The reasoning that yielded Eq. (85) from Eq. (82), (83), (84) can be immediately transposed by replacing $\mathbf{z}^l$ by $\mathbf{y}^l$, $\check{\mathbf{z}}^l$ by $\hat{\mathbf{y}}^l$ and $F_\phi(\gamma^2 + \beta^2)$ by 1.

Consequently, if all inequalities of Eq. (82), (83), (84) hold with $\eta \leq \frac{1}{4}$, then

$$
|\varrho(\mathbf{y}^l) - \varrho(\hat{\mathbf{y}}^l)| \leq 35\eta.
$$

Lemma 1 also tells us that for any $\eta > 0$ and any $\delta > 0$, there exists $N'(\eta, \delta) \in \mathbb{N}^*$ independent of $\Theta^{l-1}$, $l$ such that if $C_l \geq N'(\eta, \delta)$, it holds with probability greater than $1 - 2\delta$ with respect to $\theta^l$ that

$$
|\varrho(\hat{\mathbf{y}}^l) - \varrho(\hat{\mathbf{z}}^{l-1})| \leq \eta, \qquad\qquad |\varrho(\check{\mathbf{z}}^l) - \rho\chi(\hat{\mathbf{z}}^{l-1})\varrho(\hat{\mathbf{z}}^{l-1})| \leq \eta,
$$

where $\rho = \frac{\gamma^2}{\gamma^2 + \beta^2} < 1$ and $\chi(\hat{\mathbf{z}}^{l-1}) \in \mathbb{R}^+$ is dependent on $\Theta^{l-1}$ but independent of $\theta^l$ such that $\chi(\hat{\mathbf{z}}^{l-1}) \leq 1$ in general and $\chi(\hat{\mathbf{z}}^{l-1}) = 1$ if $\phi = \text{identity}$.

Let us then define $N_6$ independently of $\Theta^{l-1}$, $l$ as

$$
N_6(\eta, \delta) = \max\left(N'(\eta, \delta), N''\Big(\min\big(\eta, \frac{1}{4}, \frac{1}{4}F_\phi(\gamma^2 + \beta^2)\big), \delta\Big)\right), \quad \forall \eta > 0, \quad \forall \delta > 0.
$$

Then $\forall \eta > 0$, $\forall \delta > 0$, if $C_l \geq N_6(\eta, \delta)$, it holds with probability greater than $1 - 3\delta$ with respect to $\theta^l$ that

$$
\begin{aligned}
|\varrho(\mathbf{y}^l) - \varrho(\hat{\mathbf{z}}^{l-1})| &\leq |\varrho(\mathbf{y}^l) - \varrho(\hat{\mathbf{y}}^l)| + |\varrho(\hat{\mathbf{y}}^l) - \varrho(\hat{\mathbf{z}}^{l-1})| \\
&\leq 35\eta + \eta \\
&\leq 36\eta, \\
|\varrho(\mathbf{z}^l) - \rho\chi(\hat{\mathbf{z}}^{l-1})\varrho(\hat{\mathbf{z}}^{l-1})| &\leq |\varrho(\mathbf{z}^l) - \varrho(\check{\mathbf{z}}^l)| + |\varrho(\check{\mathbf{z}}^l) - \rho\chi(\hat{\mathbf{z}}^{l-1})\varrho(\hat{\mathbf{z}}^{l-1})| \\
&\leq \frac{35\eta}{F_\phi(\gamma^2 + \beta^2)} + \eta \\
&\leq \left(\frac{35 + F_\phi(\gamma^2 + \beta^2)}{F_\phi(\gamma^2 + \beta^2)}\right)\eta.
\end{aligned}
$$

Now, let us define $N_7$ independently of $\Theta^{l-1}$, $l$ as

$$N_7(\eta, \delta) = N_6\left(\min\left(\frac{\eta}{36}, \frac{F_\phi(\gamma^2 + \beta^2)}{35 + F_\phi(\gamma^2 + \beta^2)}\eta\right), \frac{\delta}{3}\right), \quad \forall \eta > 0, \quad \forall \delta > 0.$$

Then $\forall \eta > 0$, $\forall \delta > 0$, if $C_l \geq N_7(\eta, \delta)$, it holds with probability greater than $1 - \delta$ with respect to $\theta^l$ that

$$|\varrho(\mathbf{y}^l) - \varrho(\hat{\mathbf{z}}^{l-1})| \leq \eta, \qquad\qquad |\varrho(\mathbf{z}^l) - \rho\chi(\hat{\mathbf{z}}^{l-1})\varrho(\hat{\mathbf{z}}^{l-1})| \leq \eta.$$

Lemma 2 can be used again to deduce for any $\eta > 0$ and any $\delta > 0$, that if $C_l \geq N''(\eta, \delta)$, it holds with probability greater than $1 - \delta$ with respect to $\theta^l$ that $\forall \mathbf{x} \in \mathcal{D}$:

$$|\mathcal{P}_\mathbf{x}(\mathbf{y}^l) - 1| \leq |\mathcal{P}_\mathbf{x}(\mathbf{y}^l) - \mathcal{P}_\mathbf{x}(\hat{\mathbf{y}}^l)| + |\mathcal{P}_\mathbf{x}(\hat{\mathbf{y}}^l) - 1|$$
$$\leq 2\eta,$$
$$|\mathcal{P}_\mathbf{x}(\mathbf{z}^l) - F_\phi(\gamma^2 + \beta^2)| \leq |\mathcal{P}_\mathbf{x}(\mathbf{z}^l) - \mathcal{P}_\mathbf{x}(\check{\mathbf{z}}^l)| + |\mathcal{P}_\mathbf{x}(\check{\mathbf{z}}^l) - F_\phi(\gamma^2 + \beta^2)|$$
$$\leq 2\eta.$$

Let us finally define $N'''$ independently of $\Theta^{l-1}$, $l$ as $N'''(\eta, \delta) = \max\left(N_7(\eta, \frac{\delta}{2}), N''(\frac{\eta}{2}, \frac{\delta}{2})\right)$, $\forall \eta > 0$, $\forall \delta > 0$. Then, for any $\eta > 0$ and any $\delta > 0$, if $C_l \geq N'''(\eta, \delta)$, it holds with probability greater than $1 - \delta$ with respect to $\theta^l$ that

$$|\varrho(\mathbf{y}^l) - \varrho(\hat{\mathbf{z}}^{l-1})| \leq \eta, \qquad\qquad |\varrho(\mathbf{z}^l) - \rho\chi(\hat{\mathbf{z}}^{l-1})\varrho(\hat{\mathbf{z}}^{l-1})| \leq \eta,$$
$$|\mathcal{P}_\mathbf{x}(\mathbf{y}^l) - 1| \leq \eta, \quad \forall \mathbf{x} \in \mathcal{D}, \qquad \left|\mathcal{P}_\mathbf{x}(\mathbf{z}^l) - F_\phi(\gamma^2 + \beta^2)\right| \leq \eta, \quad \forall \mathbf{x} \in \mathcal{D}. \qquad \square$$

## D.3 Proof of Theorem 1

**Theorem 1.** *Fix a layer $l \geq 1$ and $\nu_{\boldsymbol{\omega}}$, $\nu_{\boldsymbol{\beta}}$, $\nu_{\boldsymbol{\gamma}}$, $\mathcal{D}$ in Definition 1. Further suppose $\mathrm{Norm} = \mathrm{LN}$ and suppose that the convolution of Eq. (2) uses periodic boundary conditions.*

*Then for any $\eta > 0$ and any $\delta > 0$, there exists $N(\eta, \delta) \in \mathbb{N}^*$ such that if $\min_{1 \leq k \leq l} C_k \geq N(\eta, \delta)$, it holds for random nets of Definition 1 with probability greater than $1 - \delta$ with respect to $\Theta^l$ that*

$$\mathcal{P}(\mathbf{y}^l) - \mathcal{P}^{(1)}(\mathbf{y}^l) \leq \rho^{l-1} + \eta, \qquad\qquad \mathcal{P}(\mathbf{y}^l) = 1, \qquad (86)$$

*where $\rho \equiv \frac{\gamma^2}{\gamma^2 + \beta^2} < 1$.*

**Proof.** For fixed $\Theta^{k-1}$ such that $\mathcal{P}_\mathbf{x}(\mathbf{z}^{k-1}) > 0$, $\forall \mathbf{x} \in \mathcal{D}$, Lemma 3 tells us for any $\delta > 0$ that, if $C_k \geq N'''\left(\min\left(\frac{1}{2}, \frac{F_\phi(\gamma^2 + \beta^2)}{2}\right), \delta\right)$ with $F_\phi \equiv \frac{\phi(1)^2 + \phi(-1)^2}{2} > 0$, it holds with probability greater than $1 - \delta$ with respect to $\theta^k$ that $\forall \mathbf{x} \in \mathcal{D}$:

$$|\mathcal{P}_\mathbf{x}(\mathbf{y}^k) - 1| \leq \frac{1}{2}, \qquad\qquad \mathcal{P}_\mathbf{x}(\mathbf{y}^k) \geq 1 - \frac{1}{2} > 0,$$
$$\left|\mathcal{P}_\mathbf{x}(\mathbf{z}^k) - F_\phi(\gamma^2 + \beta^2)\right| \leq \frac{F_\phi(\gamma^2 + \beta^2)}{2}, \qquad \mathcal{P}_\mathbf{x}(\mathbf{z}^k) \geq F_\phi(\gamma^2 + \beta^2) - \frac{F_\phi(\gamma^2 + \beta^2)}{2} > 0.$$

Let us define the event $A^{k-1} \equiv \left\{\mathcal{P}_\mathbf{x}(\mathbf{y}^{k-1}) > 0, \forall \mathbf{x} \in \mathcal{D}\right\} \cap \left\{\mathcal{P}_\mathbf{x}(\mathbf{z}^{k-1}) > 0, \forall \mathbf{x} \in \mathcal{D}\right\}$ with $A^0 \equiv \left\{\mathcal{P}_\mathbf{x}(\mathbf{x}) > 0, \forall \mathbf{x} \in \mathcal{D}\right\}$.

Given that $N'''$ is independent of $\Theta^{k-1}$, $k$, we deduce for $C_k \geq N'''\left(\min\left(\frac{1}{2}, \frac{F_\phi(\gamma^2 + \beta^2)}{2}\right), \delta\right)$ that

$$\mathbb{P}_{\Theta^k | A^{k-1}}\left[A^k\right] = \mathbb{P}_{\Theta^k | A^{k-1}}\left[\left(\mathcal{P}_\mathbf{x}(\mathbf{y}^k) > 0, \forall \mathbf{x} \in \mathcal{D}\right) \wedge \left(\mathcal{P}_\mathbf{x}(\mathbf{z}^k) > 0, \forall \mathbf{x} \in \mathcal{D}\right)\right] \geq 1 - \delta.$$

Using the fact that $\mathcal{P}_{\mathbf{x}}(\mathbf{x}) > 0$, $\forall \mathbf{x} \in \mathcal{D}$ by Definition 1, this implies for $\min_{1 \le k \le l} C_k \ge N'''\left(\min\left(\frac{1}{2}, \frac{F_\phi(\gamma^2 + \beta^2)}{2}\right), \delta\right)$ that

$$\mathbb{P}_{\Theta^l}\left[A^l\right] = \mathbb{P}_{\Theta^l}\left[\left(\mathcal{P}_{\mathbf{x}}(\mathbf{y}^l) > 0, \forall \mathbf{x} \in \mathcal{D}\right) \wedge \left(\mathcal{P}_{\mathbf{x}}(\mathbf{z}^l) > 0, \forall \mathbf{x} \in \mathcal{D}\right)\right] \ge (1 - \delta)^l.$$

Thus, for any $\delta > 0$ there exists $N_8(\delta) \in \mathbb{N}^*$ such that, if $\min_{1 \le k \le l} C_k \ge N_8(\delta)$, it holds with probability greater than $1 - \delta$ with respect to $\Theta^l$ that

$$\left(\mathcal{P}_{\mathbf{x}}(\mathbf{y}^l) > 0, \forall \mathbf{x} \in \mathcal{D}\right) \wedge \left(\mathcal{P}_{\mathbf{x}}(\mathbf{z}^l) > 0, \forall \mathbf{x} \in \mathcal{D}\right).$$

Given that $\left(\mathcal{P}_{\mathbf{x}}(\mathbf{y}^l) > 0, \forall \mathbf{x} \in \mathcal{D}\right) \implies \mathcal{P}(\mathbf{y}^l) = 1$, we deduce for any $\delta > 0$ that, if $\min_{1 \le k \le l} C_k \ge N_8(\delta)$, it holds with probability greater than $1 - \delta$ with respect to $\Theta^l$ that

$$\mathcal{P}(\mathbf{y}^l) = 1.$$

Now, until further notice, let us fix some $k$ and some $\Theta^{k-1}$ such that $\mathcal{P}_{\mathbf{x}}(\mathbf{z}^{k-1}) > 0$, $\forall \mathbf{x} \in \mathcal{D}$.

Using again Lemma 3, we get for any $1 > \eta > 0$ and any $\delta > 0$ that, if $C_k \ge N'''\left(F_\phi(\gamma^2 + \beta^2)\eta, \delta\right)$, it holds with probability greater than $1 - \delta$ with respect to $\theta^k$ that

$$F_\phi(\gamma^2 + \beta^2)(1 - \eta) \le \mathcal{P}_{\mathbf{x}}(\mathbf{z}^k) \le F_\phi(\gamma^2 + \beta^2)(1 + \eta), \quad \forall \mathbf{x} \in \mathcal{D},$$
$$F_\phi(\gamma^2 + \beta^2)(1 - \eta) \le \mathcal{P}(\mathbf{z}^k) \le F_\phi(\gamma^2 + \beta^2)(1 + \eta), \tag{87}$$
$$\sqrt{\frac{1 - \eta}{1 + \eta}} \le \sqrt{\frac{\mathcal{P}(\mathbf{z}^k)}{\mathcal{P}_{\mathbf{x}}(\mathbf{z}^k)}} \le \sqrt{\frac{1 + \eta}{1 - \eta}}, \quad \forall \mathbf{x} \in \mathcal{D},$$
$$\left(\sqrt{\frac{\mathcal{P}(\mathbf{z}^k)}{\mathcal{P}_{\mathbf{x}}(\mathbf{z}^k)}} - 1\right)^2 \le g(\eta) \equiv \max\left(\left(\sqrt{\frac{1 - \eta}{1 + \eta}} - 1\right)^2, \left(\sqrt{\frac{1 + \eta}{1 - \eta}} - 1\right)^2\right), \quad \forall \mathbf{x} \in \mathcal{D}. \tag{88}$$

If Eq. (88) holds, then

$$\hat{\mathbf{z}}_{\alpha,c}^k - \mathbf{z}_{\alpha,c}^k = \left(\sqrt{\frac{\mathcal{P}(\mathbf{z}^k)}{\mathcal{P}_{\mathbf{x}}(\mathbf{z}^k)}} - 1\right)\mathbf{z}_{\alpha,c}^k,$$

$$\mathcal{P}_{\mathbf{x}}(\hat{\mathbf{z}}^k - \mathbf{z}^k) = \left(\sqrt{\frac{\mathcal{P}(\mathbf{z}^k)}{\mathcal{P}_{\mathbf{x}}(\mathbf{z}^k)}} - 1\right)^2 \mathcal{P}_{\mathbf{x}}(\mathbf{z}^k) \le g(\eta)\mathcal{P}_{\mathbf{x}}(\mathbf{z}^k),$$

$$\mathcal{P}(\hat{\mathbf{z}}^k - \mathbf{z}^k) \le g(\eta)\mathcal{P}(\mathbf{z}^k).$$

In turn, this implies that if both Eq. (87) and Eq. (88) hold, then
$$|\mathcal{P}(\hat{\mathbf{z}}^k) - \mathcal{P}(\mathbf{z}^k)| = |\mathcal{P}(\hat{\mathbf{z}}^k - \mathbf{z}^k + \mathbf{z}^k) - \mathcal{P}(\mathbf{z}^k)|$$
$$\le \mathcal{P}(\hat{\mathbf{z}}^k - \mathbf{z}^k) + 2\mu(|\hat{\mathbf{z}}^k - \mathbf{z}^k||\mathbf{z}^k|)$$
$$\le \mathcal{P}(\hat{\mathbf{z}}^k - \mathbf{z}^k) + 2\sqrt{\mathcal{P}(\hat{\mathbf{z}}^k - \mathbf{z}^k)\mathcal{P}(\mathbf{z}^k)}$$
$$\le \left(g(\eta) + 2\sqrt{g(\eta)}\right)\mathcal{P}(\mathbf{z}^k)$$
$$\le \left(g(\eta) + 2\sqrt{g(\eta)}\right)F_\phi(\gamma^2 + \beta^2)(1 + \eta),$$
$$|\mathcal{P}^{(1)}(\hat{\mathbf{z}}^k) - \mathcal{P}^{(1)}(\mathbf{z}^k)| = |\mathcal{P}^{(1)}(\hat{\mathbf{z}}^k - \mathbf{z}^k + \mathbf{z}^k) - \mathcal{P}^{(1)}(\mathbf{z}^k)|$$
$$\le \mathcal{P}^{(1)}(\hat{\mathbf{z}}^k - \mathbf{z}^k) + 2\mathbb{E}_c\left[|\mu_c(\hat{\mathbf{z}}^k - \mathbf{z}^k)||\mu_c(\mathbf{z}^k)|\right]$$
$$\le \mathcal{P}(\hat{\mathbf{z}}^k - \mathbf{z}^k) + 2\sqrt{\mathcal{P}(\hat{\mathbf{z}}^k - \mathbf{z}^k)\mathcal{P}(\mathbf{z}^k)}$$
$$\le \left(g(\eta) + 2\sqrt{g(\eta)}\right)\mathcal{P}(\mathbf{z}^k)$$
$$\le \left(g(\eta) + 2\sqrt{g(\eta)}\right)F_\phi(\gamma^2 + \beta^2)(1 + \eta).$$

Since $\big(g(\eta) + 2\sqrt{g(\eta)}\big)F_\phi(\gamma^2 + \beta^2)(1+\eta) \to 0$ as $\eta \to 0$, we deduce for any $\eta > 0$ and any $\delta > 0$ that there exists $N_9(\eta, \delta) \in \mathbb{N}^*$ independent of $\Theta^{k-1}$, $k$ such that if $C_k \geq N_9(\eta, \delta)$, it holds with probability greater than $1 - \delta$ with respect to $\theta^k$ that

$$|\mathcal{P}_\mathbf{x}(\mathbf{z}^k) - F_\phi(\gamma^2 + \beta^2)| \leq \eta \quad \forall \mathbf{x} \in \mathcal{D}, \qquad |\mathcal{P}(\mathbf{z}^k) - F_\phi(\gamma^2 + \beta^2)| \leq \eta,$$
$$|\mathcal{P}(\hat{\mathbf{z}}^k) - \mathcal{P}(\mathbf{z}^k)| \leq \eta,$$
$$|\mathcal{P}^{(1)}(\hat{\mathbf{z}}^k) - \mathcal{P}^{(1)}(\mathbf{z}^k)| \leq \eta.$$

Defining $N_{10}$ independently of $\Theta^{k-1}$, $k$ as $N_{10}(\eta, \delta) = \max\left(N'''(\eta, \frac{\delta}{2}), N_9(\eta, \frac{\delta}{2})\right)$, $\forall \eta > 0$, $\forall \delta > 0$, we deduce for any $\eta > 0$ and any $\delta > 0$ that, if $C_k \geq N_{10}(\eta, \delta)$, it holds with probability greater than $1 - \delta$ with respect to $\theta^k$ that

$$|\varrho(\mathbf{y}^k) - \varrho(\hat{\mathbf{z}}^{k-1})| \leq \eta,$$
$$|\varrho(\mathbf{z}^k) - \rho\chi(\hat{\mathbf{z}}^{k-1})\varrho(\hat{\mathbf{z}}^{k-1})| \leq \eta,$$
$$|\mathcal{P}(\mathbf{z}^k) - F_\phi(\gamma^2 + \beta^2)| \leq \eta,$$
$$|\mathcal{P}(\hat{\mathbf{z}}^k) - \mathcal{P}(\mathbf{z}^k)| \leq \eta,$$
$$|\mathcal{P}^{(1)}(\hat{\mathbf{z}}^k) - \mathcal{P}^{(1)}(\mathbf{z}^k)| \leq \eta.$$

The reasoning that yielded Eq. (85) from Eq. (82), (83), (84) can be immediately transposed by replacing $\check{\mathbf{z}}^l$ by $\mathbf{z}^k$ and $\mathbf{z}^l$ by $\hat{\mathbf{z}}^k$.

Thus, if $C_k \geq N_{10}(\eta, \delta)$, it holds with probability greater than $1 - \delta$ with respect to $\theta^k$ that

$$|\varrho(\mathbf{y}^k) - \varrho(\hat{\mathbf{z}}^{k-1})| \leq \eta,$$
$$|\varrho(\mathbf{z}^k) - \rho\chi(\hat{\mathbf{z}}^{k-1})\varrho(\hat{\mathbf{z}}^{k-1})| \leq \eta,$$
$$|\varrho(\hat{\mathbf{z}}^k) - \varrho(\mathbf{z}^k)| \leq \frac{35\eta}{F_\phi(\gamma^2 + \beta^2)},$$
$$|\varrho(\hat{\mathbf{z}}^k) - \rho\chi(\hat{\mathbf{z}}^{k-1})\varrho(\hat{\mathbf{z}}^{k-1})| \leq \eta + \frac{35\eta}{F_\phi(\gamma^2 + \beta^2)} = \frac{F_\phi(\gamma^2 + \beta^2) + 35}{F_\phi(\gamma^2 + \beta^2)}\eta.$$

Defining $N_{11}$ independently of $\Theta^{k-1}$, $k$ as $N_{11}(\eta, \delta) = N_{10}\left(\frac{F_\phi(\gamma^2 + \beta^2)}{F_\phi(\gamma^2 + \beta^2) + 35}\eta, \delta\right)$, $\forall \eta > 0$, $\forall \delta > 0$, we deduce for any $\eta > 0$ and any $\delta > 0$ that, if $C_k \geq N_{11}(\eta, \delta)$, it holds that

$$\mathbb{P}_{\theta^k}\left[|\varrho(\mathbf{y}^k) - \varrho(\hat{\mathbf{z}}^{k-1})| \leq \eta\right] \geq \mathbb{P}_{\theta^k}\left[|\varrho(\mathbf{y}^k) - \varrho(\hat{\mathbf{z}}^{k-1})| \leq \frac{F_\phi(\gamma^2 + \beta^2)}{F_\phi(\gamma^2 + \beta^2) + 35}\eta\right] \geq 1 - \delta,$$
$$\mathbb{P}_{\theta^k}\left[|\varrho(\hat{\mathbf{z}}^k) - \rho\chi(\hat{\mathbf{z}}^{k-1})\varrho(\hat{\mathbf{z}}^{k-1})| \leq \eta\right] \geq 1 - \delta.$$

Considering again $k$ and $\Theta^{k-1}$ as not fixed, we deduce for any $k$ that, if $C_k \geq N_{11}(\eta, \delta)$, it holds that

$$\mathbb{P}_{\Theta^k | A^{k-1}}\left[|\varrho(\mathbf{y}^k) - \varrho(\hat{\mathbf{z}}^{k-1})| \leq \eta\right] \geq 1 - \delta,$$
$$\mathbb{P}_{\Theta^k | A^{k-1}}\left[|\varrho(\hat{\mathbf{z}}^k) - \rho\chi(\hat{\mathbf{z}}^{k-1})\varrho(\hat{\mathbf{z}}^{k-1})| \leq \eta\right] \geq 1 - \delta.$$

Defining $N_{12}$ independently of $\Theta^k$, $k$ as $N_{12}(\eta, \delta) = \max\left(N_{11}(\eta, \frac{\delta}{2}), N_8(\frac{\delta}{2})\right)$, $\forall \eta > 0$, $\forall \delta > 0$, we deduce for any $k$, any $\eta > 0$ and any $\delta > 0$ that, if $C_k \geq N_{12}(\eta, \delta)$, it holds that

$$\mathbb{P}_{\Theta^k}\left[|\varrho(\mathbf{y}^k) - \varrho(\hat{\mathbf{z}}^{k-1})| \leq \eta\right] \geq \mathbb{P}_{\Theta^k | A^{k-1}}\left[|\varrho(\mathbf{y}^k) - \varrho(\hat{\mathbf{z}}^{k-1})| \leq \eta\right]\mathbb{P}_{\Theta^{k-1}}\left[A^{k-1}\right]$$
$$\geq \mathbb{P}_{\Theta^k | A^{k-1}}\left[|\varrho(\mathbf{y}^k) - \varrho(\hat{\mathbf{z}}^{k-1})| \leq \eta\right]\mathbb{P}_{\Theta^l}\left[A^l\right]$$
$$\geq \left(1 - \frac{\delta}{2}\right)\left(1 - \frac{\delta}{2}\right) \geq 1 - \delta,$$
$$\mathbb{P}_{\Theta^k}\left[|\varrho(\hat{\mathbf{z}}^k) - \rho\chi(\hat{\mathbf{z}}^{k-1})\varrho(\hat{\mathbf{z}}^{k-1})| \leq \eta\right] \geq \mathbb{P}_{\Theta^k | A^{k-1}}\left[|\varrho(\hat{\mathbf{z}}^k) - \rho\chi(\hat{\mathbf{z}}^{k-1})\varrho(\hat{\mathbf{z}}^{k-1})| \leq \eta\right]\mathbb{P}_{\Theta^l}\left[A^l\right]$$
$$\geq \left(1 - \frac{\delta}{2}\right)\left(1 - \frac{\delta}{2}\right) \geq 1 - \delta.$$

Thus, for any $\eta > 0$ and any $\delta > 0$, if $\min_{1 \leq k \leq l} C_k \geq N_{12}(\eta, \delta)$, it holds with probability greater than $1 - l\delta$ with respect to $\Theta^l$ that

$$|\varrho(\hat{\mathbf{z}}^k) - \rho\chi(\hat{\mathbf{z}}^{k-1})\varrho(\hat{\mathbf{z}}^{k-1})| \leq \eta, \quad \forall k \leq l - 1, \tag{89}$$

$$|\varrho(\mathbf{y}^l) - \varrho(\hat{\mathbf{z}}^{l-1})| \leq \eta. \tag{90}$$

Given $\chi(\hat{\mathbf{z}}^{k-1}) \leq 1, \forall k$ and given $\varrho(\mathbf{z}^0) = \varrho(\mathbf{x}) \leq 1$, we note that if Eq. (89) and Eq. (90) hold, then

$$\varrho(\hat{\mathbf{z}}^1) \leq \rho\chi(\mathbf{z}^0)\varrho(\mathbf{z}^0) + \eta \leq \rho + \eta,$$
$$\varrho(\hat{\mathbf{z}}^2) \leq \rho\chi(\hat{\mathbf{z}}^1)\varrho(\hat{\mathbf{z}}^1) + \eta \leq \rho^2 + \rho\eta + \eta,$$
$$\vdots$$

$$\varrho(\hat{\mathbf{z}}^{l-1}) \leq \rho\chi(\hat{\mathbf{z}}^{l-2})\varrho(\hat{\mathbf{z}}^{l-2}) + \eta \leq \rho^{l-1} + \Big(\sum_{k=0}^{l-2} \rho^k\Big)\eta \leq \rho^{l-1} + \frac{1}{1-\rho}\eta,$$

$$\varrho(\mathbf{y}^l) \leq \varrho(\hat{\mathbf{z}}^{l-1}) + \eta \leq \rho^{l-1} + \frac{1}{1-\rho}\eta + \eta \leq \rho^{l-1} + \Big(\frac{2-\rho}{1-\rho}\Big)\eta. \tag{91}$$

Defining $N_{13}$ such that $N_{13}(\eta, \delta) = N_{12}\Big(\frac{1-\rho}{2-\rho}\eta, \frac{1}{l}\delta\Big)$, $\forall \eta > 0$, $\forall \delta > 0$, we deduce for any $\eta > 0$ and any $\delta > 0$ that, if $\min_{1 \leq k \leq l} C_k \geq N_{13}(\eta, \delta)$, it holds with probability greater than $1 - \delta$ with respect to $\Theta^l$ that

$$\varrho(\mathbf{y}^l) \leq \rho^{l-1} + \eta.$$

Finally, let us define $N$ such that $N(\eta, \delta) = \max\Big(N_8(\frac{\delta}{2}), N_{13}(\eta, \frac{\delta}{2})\Big)$, $\forall \eta > 0$, $\forall \delta > 0$. Then for any $\eta > 0$ and any $\delta > 0$, if $\min_{1 \leq k \leq l} C_k \geq N(\eta, \delta)$, it holds with probability greater than $1 - \delta$ with respect to $\Theta^l$ that

$$\varrho(\mathbf{y}^l) = \mathcal{P}(\mathbf{y}^l) - \mathcal{P}^{(1)}(\mathbf{y}^l) \leq \rho^{l-1} + \eta, \qquad \mathcal{P}(\mathbf{y}^l) = 1. \qquad \square$$

### D.4  Case $\phi = \text{identity}$

**Proposition 6.** *Fix a layer $l \geq 1$ and $\nu_{\boldsymbol{\omega}}$, $\nu_{\boldsymbol{\beta}}$, $\nu_{\boldsymbol{\gamma}}$, $\mathcal{D}$ in Definition 1, with $\mathcal{D}$ "centered" such that $\mathcal{P}^{(1)}(\mathbf{z}^0) = \mathcal{P}^{(1)}(\mathbf{x}) = 0$. Further suppose $\text{Norm} = \text{LN}$, $\phi = \text{identity}$, and suppose that the convolution of Eq. (2) uses periodic boundary conditions.*

*Then for any $\eta > 0$ and any $\delta > 0$, there exists $N(\eta, \delta) \in \mathbb{N}^*$ such that if $\min_{1 \leq k \leq l} C_k \geq N(\eta, \delta)$, it holds for random nets of Definition 1 with probability greater than $1 - \delta$ with respect to $\Theta^l$ that*

$$|\mathcal{P}(\mathbf{y}^l) - \mathcal{P}^{(1)}(\mathbf{y}^l) - \rho^{l-1}| \leq \eta, \qquad \mathcal{P}(\mathbf{y}^l) = 1,$$

*where $\rho \equiv \frac{\gamma^2}{\gamma^2 + \beta^2} < 1$.*

**Proof.** Since $\phi = \text{identity}$ is a particular case of positive homogeneous activation function, the whole proof of Theorem 1 still applies. Let us then define $N_8$, $N_{12}$ as in the proof of Theorem 1.

Then for any $\delta > 0$, if $\min_{1 \leq k \leq l} C_k \geq N_8(\delta)$, it holds with probability greater than $1 - \delta$ with respect to $\Theta^l$ that

$$\mathcal{P}(\mathbf{y}^l) = 1.$$

In addition, for any $\eta > 0$ and any $\delta > 0$, if $\min_{1 \leq k \leq l} C_k \geq N_{12}(\eta, \delta)$, it holds with probability greater than $1 - l\delta$ with respect to $\Theta^l$ that

$$|\varrho(\hat{\mathbf{z}}^k) - \rho\chi(\hat{\mathbf{z}}^{k-1})\varrho(\hat{\mathbf{z}}^{k-1})| = |\varrho(\hat{\mathbf{z}}^k) - \rho\varrho(\hat{\mathbf{z}}^{k-1})| \leq \eta, \quad \forall k \leq l - 1,$$

$$|\varrho(\mathbf{y}^l) - \varrho(\hat{\mathbf{z}}^{l-1})| \leq \eta,$$

where we used the fact that $\chi(\hat{\mathbf{z}}^{k-1}) = 1, \forall k$ when $\phi = $ identity.

Next we note that: (i) the assumptions that $\mathcal{P}^{(1)}(\mathbf{z}^0) = 0$ and $\mathcal{P}_\mathbf{x}(\mathbf{z}^0) > 0, \forall \mathbf{x}$ (cf Definition 1) together imply $\varrho(\mathbf{z}^0) = \varrho(\mathbf{x}) = 1$; (ii) the reasoning yielding Eq. (91) from Eq. (89), (90) still applies.

We deduce that, if $\min_{1 \leq k \leq l} C_k \geq N_{12}(\eta, \delta)$, it holds with probability greater than $1 - l\delta$ with respect to $\Theta^l$ that

$$\varrho(\hat{\mathbf{z}}^1) \geq \rho\varrho(\mathbf{x}) - \eta = \rho - \eta,$$
$$\varrho(\hat{\mathbf{z}}^2) \geq \rho\varrho(\hat{\mathbf{z}}^1) - \eta \geq \rho^2 - \rho\eta - \eta,$$
$$\vdots$$
$$\varrho(\hat{\mathbf{z}}^{l-1}) \geq \rho\varrho(\hat{\mathbf{z}}^{l-2}) - \eta \geq \rho^{l-1} - \Big(\sum_{k=0}^{l-2}\rho^k\Big)\eta \geq \rho^{l-1} - \frac{1}{1-\rho}\eta,$$
$$\varrho(\mathbf{y}^l) \geq \varrho(\hat{\mathbf{z}}^{l-1}) - \eta \geq \rho^{l-1} - \frac{1}{1-\rho}\eta - \eta \geq \rho^{l-1} - \Big(\frac{2-\rho}{1-\rho}\Big)\eta,$$
$$\varrho(\mathbf{y}^l) \leq \rho^{l-1} + \Big(\frac{2-\rho}{1-\rho}\Big)\eta, \tag{92}$$
$$|\varrho(\mathbf{y}^l) - \rho^{l-1}| \leq \Big(\frac{2-\rho}{1-\rho}\Big)\eta,$$

where Eq. (92) follows from Eq. (91).

As in the proof of Theorem 1, defining $N_{13}$ such that $N_{13}(\eta, \delta) = N_{12}\Big(\frac{1-\rho}{2-\rho}\eta, \frac{1}{l}\delta\Big)$, we deduce for any $\eta > 0$ and any $\delta > 0$ that, if $\min_{1 \leq k \leq l} C_k \geq N_{13}(\eta, \delta)$, it holds with probability greater than $1 - \delta$ with respect to $\Theta^l$ that

$$|\varrho(\mathbf{y}^l) - \rho^{l-1}| \leq \eta.$$

As in the proof of Theorem 1, defining $N$ such that $N(\eta, \delta) = \max\Big(N_8(\frac{\delta}{2}), N_{13}(\eta, \frac{\delta}{2})\Big), \forall \eta > 0$, $\forall \delta > 0$, we deduce for any $\eta > 0$ and any $\delta > 0$ that, if $\min_{1 \leq k \leq l} C_k \geq N(\eta, \delta)$, it holds with probability greater than $1 - \delta$ with respect to $\Theta^l$ that

$$|\varrho(\mathbf{y}^l) - \rho^{l-1}| = |\mathcal{P}(\mathbf{y}^l) - \mathcal{P}^{(1)}(\mathbf{y}^l) - \rho^{l-1}| \leq \eta, \qquad \mathcal{P}(\mathbf{y}^l) = 1. \qquad \square$$

# E   Proof of Theorem 2

**Theorem 2 .** *Fix a layer $l \in \{1, \ldots, L\}$ and lift any assumptions on $\phi$. Further suppose* Norm = IN, *with Eq. (3) having nonzero denominator at layer $l$ for all inputs and channels.*

*Then it holds that*

- $\mathbf{y}^l$ *is normalized in each channel $c$ with*
$$\mathcal{P}_c^{(1)}(\mathbf{y}^l) = 0, \qquad\qquad \mathcal{P}_c(\mathbf{y}^l) = 1;$$
- $\mathbf{y}^l$ *lacks variability in instance statistics in each channel $c$ with*
$$\mathcal{P}_c^{(2)}(\mathbf{y}^l) = 0, \qquad \mathcal{P}_c^{(3)}(\mathbf{y}^l) = 1, \qquad \mathcal{P}_c^{(4)}(\mathbf{y}^l) = 0.$$

**Proof.** With Norm = IN, if $\forall \mathbf{x}, c: \sigma_{\mathbf{x},c}(\mathbf{x}^l) > 0$, then instance statistics are given by

$$\mu_{\mathbf{x},c}(\mathbf{y}^l) = \mathbb{E}_\alpha\big[\mathbf{y}_{\alpha,c}^l\big] = \frac{\mathbb{E}_\alpha\big[\mathbf{x}_{\alpha,c}^l\big] - \mu_{\mathbf{x},c}(\mathbf{x}^l)}{\sigma_{\mathbf{x},c}(\mathbf{x}^l)} = \frac{\mu_{\mathbf{x},c}(\mathbf{x}^l) - \mu_{\mathbf{x},c}(\mathbf{x}^l)}{\sigma_{\mathbf{x},c}(\mathbf{x}^l)} = 0,$$

$$\mathcal{P}_{\mathbf{x},c}(\mathbf{y}^l) = \mathbb{E}_\alpha\Big[(\mathbf{y}_{\alpha,c}^l)^2\Big] = \frac{\mathbb{E}_\alpha\Big[\big(\mathbf{x}_{\alpha,c}^l - \mu_{\mathbf{x},c}(\mathbf{x}^l)\big)^2\Big]}{\sigma_{\mathbf{x},c}(\mathbf{x}^l)^2} = \frac{\sigma_{\mathbf{x},c}(\mathbf{x}^l)^2}{\sigma_{\mathbf{x},c}(\mathbf{x}^l)^2} = 1,$$

$$\sigma_{\mathbf{x},c}(\mathbf{y}^l) = \sqrt{\mathcal{P}_{\mathbf{x},c}(\mathbf{y}^l) - \mu_{\mathbf{x},c}(\mathbf{y}^l)^2} = 1.$$

In turn, this implies for the different power terms:

$$\mathcal{P}_c(\mathbf{y}^l) = \mathbb{E}_{\mathbf{x}}\Big[\mathcal{P}_{\mathbf{x},c}(\mathbf{y}^l)\Big] = \mathbb{E}_{\mathbf{x}}\big[1\big] = 1,$$

$$\mathcal{P}_c^{(1)}(\mathbf{y}^l) = \mathbb{E}_{\mathbf{x}}\Big[\mu_{\mathbf{x},c}(\mathbf{y}^l)\Big]^2 = \mathbb{E}_{\mathbf{x}}\big[0\big]^2 = 0, \quad \mathcal{P}_c^{(2)}(\mathbf{y}^l) = \text{Var}_{\mathbf{x}}\Big[\mu_{\mathbf{x},c}(\mathbf{y}^l)\Big] = \text{Var}_{\mathbf{x}}\big[0\big] = 0,$$

$$\mathcal{P}_c^{(3)}(\mathbf{y}^l) = \mathbb{E}_{\mathbf{x}}\Big[\sigma_{\mathbf{x},c}(\mathbf{y}^l)\Big]^2 = \mathbb{E}_{\mathbf{x}}\big[1\big]^2 = 1, \quad \mathcal{P}_c^{(4)}(\mathbf{y}^l) = \text{Var}_{\mathbf{x}}\Big[\sigma_{\mathbf{x},c}(\mathbf{y}^l)\Big] = \text{Var}_{\mathbf{x}}\big[1\big] = 0. \quad \square$$

# F  Proof of Theorem 3

**Theorem 3 .** *Fix a layer $l \in \{1, \dots, L\}$ and lift any assumptions on $\phi$ and $\mathbf{x}$'s distribution. Further suppose that the neural network implements Eq. (2), (3), (7) at every layer up to depth $L$, with $\epsilon = 0$ and Eq. (3), (7) having nonzero denominators for all layers, inputs and channels.*

*Finally suppose that*

- *The proxy-normalized post-activations $\tilde{\mathbf{z}}^{l-1}$ at layer $l-1$ are normalized in each channel $c$:*
$$\mathcal{P}_c^{(1)}(\tilde{\mathbf{z}}^{l-1}) = 0, \qquad\qquad \mathcal{P}_c(\tilde{\mathbf{z}}^{l-1}) = 1;$$

- *The convolution and normalization steps at layer $l$ do not cause any aggravation of channel-wise collapse and channel imbalance, i.e. $\forall c, c'$:*
$$\mathcal{P}_c^{(1)}(\mathbf{y}^l) = 0, \qquad\qquad \mathcal{P}_c(\mathbf{y}^l) = \mathcal{P}_{c'}(\mathbf{y}^l);$$

- *The pre-activations $\mathbf{y}^l$ at layer $l$ are Gaussian in each channel $c$ and PN's additional parameters $\tilde{\boldsymbol{\beta}}^l, \tilde{\boldsymbol{\gamma}}^l$ are zero.*

*Then both the pre-activations $\mathbf{y}^l$ and the proxy-normalized post-activations $\tilde{\mathbf{z}}^l$ at layer $l$ are normalized in each channel $c$:*

$$\mathcal{P}_c^{(1)}(\mathbf{y}^l) = 0, \qquad\qquad\qquad \mathcal{P}_c(\mathbf{y}^l) = 1, \qquad\qquad (93)$$
$$\mathcal{P}_c^{(1)}(\tilde{\mathbf{z}}^l) = 0, \qquad\qquad\qquad \mathcal{P}_c(\tilde{\mathbf{z}}^l) = 1. \qquad\qquad (94)$$

**Proof of Eq. (93).** If the denominator of Eq. (3) is nonzero for all layers, inputs and channels, then it follows from Proposition 1 that $\mathcal{P}(\mathbf{y}^l) = \mathbb{E}_c\big[\mathcal{P}_c(\mathbf{y}^l)\big] = 1$.

Combined with $\mathcal{P}_c^{(1)}(\mathbf{y}^l) = 0, \forall c$ and $\mathcal{P}_c(\mathbf{y}^l) = \mathcal{P}_{c'}(\mathbf{y}^l), \forall c, c'$, we deduce that in each channel $c$:
$$\mathcal{P}_c^{(1)}(\mathbf{y}^l) = 0, \qquad\qquad\qquad \mathcal{P}_c(\mathbf{y}^l) = 1. \qquad\qquad \square$$

**Proof of Eq. (94).** Given Eq. (93) and given that $\mathbf{y}^l$ is Gaussian in each channel $c$, it holds $\forall c$:
$$\mathbf{y}_{\alpha,c}^l \underset{\mathbf{x},\alpha}{\sim} \mathcal{N}\big(0,1\big).$$

We deduce that the distribution of $\mathbf{z}_{\alpha,c}^l = \phi\big(\gamma_c^l \mathbf{y}_{\alpha,c}^l + \beta_c^l\big)$ with respect to $(\mathbf{x}, \alpha)$ and the distribution of $\phi\big(\gamma_c^l Y_c^l + \beta_c^l\big)$ with respect to $Y_c^l \sim \mathcal{N}\big(\tilde{\beta}_c^l, (1 + \tilde{\gamma}_c^l)^2\big) = \mathcal{N}\big(0,1\big)$ are equal.

We then get from Eq. (7) at layer $l$ that
$$\tilde{\mathbf{z}}_{\alpha,c}^l = \frac{\mathbf{z}_{\alpha,c}^l - \mathbb{E}_{Y_c^l}\big[\phi\big(\gamma_c^l Y_c^l + \beta_c^l\big)\big]}{\sqrt{\text{Var}_{Y_c^l}\big[\phi\big(\gamma_c^l Y_c^l + \beta_c^l\big)\big] + \epsilon}} = \frac{\mathbf{z}_{\alpha,c}^l - \mathbb{E}_{\mathbf{x},\alpha}\big[\mathbf{z}_{\alpha,c}^l\big]}{\sqrt{\text{Var}_{\mathbf{x},\alpha}\big[\mathbf{z}_{\alpha,c}^l\big]}} = \frac{\mathbf{z}_{\alpha,c}^l - \mu_c(\mathbf{z}^l)}{\sigma_c(\mathbf{z}^l)},$$

where we used $\epsilon = 0$.

We deduce that $\mu_c(\tilde{\mathbf{z}}^l) = 0$ and $\mathcal{P}_c(\tilde{\mathbf{z}}^l) = 1$ in each channel $c$, implying that in each channel $c$:
$$\mathcal{P}_c^{(1)}(\tilde{\mathbf{z}}^l) = \mu_c(\tilde{\mathbf{z}}^l)^2 = 0, \qquad\qquad\qquad \mathcal{P}_c(\tilde{\mathbf{z}}^l) = 1. \qquad\qquad \square$$