# OpenReview forum: "Proxy-Normalizing Activations to Match Batch Normalization while Removing Batch Dependence"
_NeurIPS.cc/2021/Conference — NeurIPS 2021 Poster_

### Official Review · Reviewer_89jK · 2021-07-15

**Rating:** 6
**Confidence:** 3

**Summary:**

This paper analyzes the properties of normalization methods. It shows how layer normalization induces a collapse towards channel-wise constant functions with assumptions, and how the constraint imposed by normalization affects the performance of instance normalization. It also proposes Proxy Normalization that emulates batch normalization’s behavior and performance, when combined with batch-independent normalization (e.g, layer normalization or group normalization) .

**Limitations And Societal Impact:**

yes

**Main Review:**

**Pros:**

+This paper summarizes many properties of normalization methods, even though these properties of normalization have been separately shown in previous works.
+The analysis shown in section 3 is novel, even though I have concerns on the claims.
+The experiments of this paper are comprehensive.


**Cons:**

-It is not rigorous to state that BN preserve the expressivity. Particularly, there are work [1*] showing that the batch size affects the layer representation/expressivity of BN, and further affects the model’s representation/expressivity. For example, using a batch size of 2 for a MLP, it is clear that activation is constraint to be 1 and -1 during training. Therefore, BN could change the expressivity, noting that the optimization of BN is over the mini-batch statistics, not the population statistics during training.

-I do not recognize the practicality of Proxy Normalization:
 (1) This paper should conduct experiments on small batch size scenario (e.g., batch size of 2 on ImageNet or object detection tasks on CoCo) to support its claims in Line-306 and 309. Based on the current experimental results, I do not find why one person should use PN+GN/LN, rather than BN, considering that GN/LN introduces extra computational cost during inference.
(2) There are two extra hyper-parameters $\tilde{\beta}_c^l$ and  $\tilde{\gamma}_c^l$, How to determine them? Do the results sensitive to these hyper-parameters? This paper should conduct experiments to discuss the sensitivity of these two hyper-parameters over different architecture/dataset.


-Some description and notation is not rigorous:
(1) Line 140, why  $\tilde{y}^l$ is unlikely to have its channel-wise distributions collapsed?, if the $\beta$ is biased to a very large negative value, all the normalized output will less than 0, leading to all 0 output.
(2) Line 117~118, what is $E_\mathbf{x}$, a, does this mean the expectation over the mini-batch dimension? If yes, why denote $\mathbf{x} \in R^{H \times W \times C_l}$?


**Ref:**

[1*] Group Whitening: Balancing Learning Efficiency and Representational Capacity, arxiv: 2009.13333



**Update**: I have read the authors responses and other reviewer’s comments, The authors mostly addressed my concerns relating to the clarity. I still has the concerns on the practicality, considering that LN/GN is required (For their scale invariant to stabilize training) in the solution, while LN/GN introduces extra computational cost during inference.
I do recognize the analysis shown in section 3, which is novel to me. I thus keep positive to this paper and keep my score as 6.


**Time Spent Reviewing:**

6 hours

---

> ### Author Response · Authors · 2021-08-10
> **Response to reviewer 89jK**
>
> We thank the reviewer very much for his/her time and for the very useful comments and feedback. We provide below a response to each comment.
>
> **Comment: “It is not rigorous to state that BN preserve the expressivity. Particularly, there are work [1] showing that the batch size affects the layer representation/expressivity of BN, and further affects the model’s representation/expressivity.”**
>
> We agree with the reviewer that BN does not exactly preserve expressivity in the mini-batch setting. In fact, we do state that in the caption of Table 1.
>
> To make this point clearer, we will also state in the text of Section 2 that the preservation of expressivity is only strict in the full-batch setting. In Table 1, we will state that check marks denote the "presence/approximate presence" of a property and that BN is considered in the mini-batch setting, with this mini-batch setting close enough to the large-batch setting for the expressivity to be approximately preserved.
>
> We thank the reviewer for pointing out the reference Huang et al. 2021, that we had missed. We will refer to it in the caption of Table 1 and in the discussion of Section 4 when relating constraints on instance statistics to alteration of the expressivity.
>
> **Comment: “This paper should conduct experiments on small batch size scenario (e.g., batch size of 2 on ImageNet or object detection tasks on CoCo) to support its claims in Line-306 and 309.”**
>
> In our experiments, we intentionally:
> - Use the same batch size with BN and with batch-independent norms
> - Set the value of this batch size in a regime where BN works well
>
> Since BN is matched/outperformed with GN+PN in such a regime of batch size, we expect that it would a fortiori be matched/outperformed at small batch size. Indeed:
> - With batch-independent norms, we expect at least the same performance if we use a smaller batch size with a linear scaling rule (Goyal et al., 2017). In any case, we can always plainly aggregate gradients over multiple iterations to fall back to the results obtained at larger batch size.
> - With BN, on the other hand, we expect degraded performance if we use a small batch size, as documented in the literature (cf e.g. Ioffe, 2017; Wu & He 2018).
>
> For clarity, we will insert the above discussion at the beginning of Section 6 to justify our choice of batch size.
>
> **Comment: “There are two extra hyper-parameters $\tilde{\boldsymbol{\beta}}^l$ and $\tilde{\boldsymbol{\gamma}}^l$, How to determine them? Do the results sensitive to these hyper-parameters?”**
>
> We want to clarify that $\tilde{\boldsymbol{\beta}}^l$, $\tilde{\boldsymbol{\gamma}}^l$ are not hyper-parameters but "additional parameters" (as we state on line 251). These parameters are learnable. They are initialized at 0 and learned during training, so they do not require to be specified a priori.
>
> We thank the reviewer for raising the point regarding the sensitivity of our results to the presence of these additional parameters. Following the reviewer’s comment, we have rerun the experiments of Tables 3 and 4 for the case of PN without $\tilde{\boldsymbol{\beta}}^l$, $\tilde{\boldsymbol{\gamma}}^l$. The results are as follows (single run per case, with formatting X / Y where X, Y are the results without and with extra regularization, respectively):
>
> || RN50        | RN101       | RNX50       | RNX101      | depthwise convs EN-B0       | depthwise convs EN-B2       | group convs EN-B0       | group convs EN-B2       |
> |:-----------------------------|:------------|:------------|:------------|:------------|:------------|:------------|:------------|:------------|
> | GN+PN w/ $\tilde{\boldsymbol{\beta}}^l$,$\tilde{\boldsymbol{\gamma}}^l$  | 76.3 / **76.7** | 77.6 / **78.6** | 76.7 / **77.8** | 77.7 / **79.0** | 76.7 / **77.0** | 79.4 / **80.0** | 76.6 / **76.8** | 79.3 / **80.1** |
> | GN+PN w/o $\tilde{\boldsymbol{\beta}}^l$,$\tilde{\boldsymbol{\gamma}}^l$ | 76.3 / **76.8** | 77.5 / **78.5** | 76.4 / **77.6** | 77.7 / **78.9** | 76.7 / **76.7** | 79.3 / **79.8** | 76.5 / **76.6** | 79.2 / **79.9** |
>
> We find that the drop of performance that would result from dropping $\tilde{\boldsymbol{\beta}}^l$, $\tilde{\boldsymbol{\gamma}}^l$ in PN is in fact low.
>
> We will provide these additional results in appendix (with multiple runs per case). We will also state in Section 5 that $\tilde{\boldsymbol{\beta}}^l$, $\tilde{\boldsymbol{\gamma}}^l$ are optional.
>
> **Comment: “Some description and notation is not rigorous:**
>
> **(1) Line 140, why $\tilde{\mathbf{y}}^l$ is unlikely to have its channel-wise distributions collapsed?, if the $\boldsymbol{\beta}$ is biased to a very large negative value, all the normalized output will less than 0, leading to all 0 output.“**
>
> **(2) Line 117~118, what is $E_\mathbf{x}$, a, does this mean the expectation over the mini-batch dimension? If yes, why denote $\mathbf{x}\in R^{H \times W \times C_l}$?“**
>
> To reassure the reviewer regarding the notational rigor:
>
> (1) We state that it is "unlikely" that $\tilde{\mathbf{y}}^l$ has its channel-wise distributions collapsed — and that it falls on only one side of piece-wise linearity — but not that it is impossible. Under reasonable values of the r.m.s. of $\boldsymbol{\beta}^l$, $\boldsymbol{\gamma}^l$ in the context of random networks of Definition 1, there is only a small probability that the distribution of $\tilde{\mathbf{y}}^l$ is channel-wise collapsed (note that $\boldsymbol{\beta}^l$ is assumed centered around 0 in Definition 1). We will insert this clarification in the text.
>
> (2) We state at the beginning of Section 2 that the intermediate activations have a dependence on the input $\mathbf{x}$ that is kept implicit (in the same way as the dependence on model parameters). The expectation with respect to $\mathbf{x}$ is thus an expectation with respect to the sampling of the input $\mathbf{x}\sim\mathcal{D}$ that the intermediate activations depend on.
>
> We will remind this implicit dependence on $\mathbf{x}$ just after the equations introducing the moments in Section 3.
>
> **References**
>
> Goyal et al., “Accurate, Large Minibatch SGD: Training ImageNet in 1 Hour”, 2017
>
> Ioffe, “Batch Renormalization: Towards Reducing Minibatch Dependence in Batch-Normalized Models”, 2017
>
> Wu & He, “Group Normalization”, 2018

---

### Official Review · Reviewer_AgAg · 2021-07-19

**Rating:** 7
**Confidence:** 3

**Summary:**

This work proposes some theoretical analyses on different normalization schemes in deep neural networks. It found that LN suffers from the collapsing effect over the number of layers, which makes the computation more and more linear, whereas IN suffers from a lack of expressivity. Towards fixing these issues, the paper proposed ProxyNorm that adds another normalization layer after the activation function. Empirical results suggest that the combination of ProxyNorm and LayerNorm or GroupNorm can lead to matching or sometimes superior performance compared to BatchNorm.

**Main Review:**

------------------------------------------
### Strengths
- Insightful analysis on different failure modes of LN and IN.
- Proposed a working method that shows empirical improvement over standard LN and GN, with a matching performance to BN. This is a very exciting result.
- Overall, the experiment numbers seem solid.
- The experimental details and implementation details are included in the appendix.

------------------------------------------
### Weaknesses
- **More empirical studies on the collapsing phenomenon in LN and GN.** Currently the plot in Figure 3 is not very clear about the collapsing effect. $\mathcal{P}^{(1)}$ does grow, but it does not converge to 1, and the rest of the terms do not converge to 0, unlike suggested by the theorem. So I am wondering if the paper could improve the empirical studies to verify the collapsing effect. First, you don’t have to use a ResNet-50, but it could be a regular convolutional network with many more layers. Second, I would also like to see other metrics to measure the linearity of the layers empirically over the number of layers. E.g. the proportion of negative entries in the pre-activation.
- **Analysis depends on the use of ReLU activations:** Line 111 states that the activation needs to be positive homogeneous and non-zero, and based on this it is clear that we only have scaled or flipped ReLU type activations. There have also been other developments of activation functions such as SeLU that have self-normalizing effects. I wonder if this can be baked into the analysis and lead to other insights? Instead of adding another normalization layer after the activation function.
- **Reliance to be preceded by another normalization function:** First of all, I don’t quite understand the requirement stated in Line 257-258: “The only required constraint for PN to work optimally is that each activation step should be immediately preceded by a normalization step.” I expect to see more concrete justification for this requirement and the transition of going into LN+PN and GN+PN is too fast. Secondly, doing a double normalization makes the computation less elegant. The work would have more impact by just introducing one single normalization step.
- **Guarantees on LN+PN/GN+PN:** I might have entirely missed the point but I would appreciate it if there is a theorem or remark showing that LN+PN / GN+PN is able to avoid the collapsing effect.

------------------------------------------
### Clarification
- **Computing statistics analytically:** If we assume that we have a ReLU activation, can we analytically compute the mean and variance of the proxy distribution? It can simplify the modification to standard networks a lot, by simply introducing pre-computed affine transformations instead of collecting statistics from samples.
- **The reason for a proxy distribution:** It also was not clear to me what is the purpose for adding a proxy distribution. Why not just normalize the post-activations using the statistics from themselves?

------------------------------------------
### Conclusion
In conclusion, I found this is a nice paper for advancing our understanding of neural network normalizations. However, I found the proposed solution somehow less elegant, and there is some disconnect between the theoretical and the experimental sections. Some practical design choices are not well explained (e.g. the need for applying together with an existing normalization layer), and a lack of theoretical guarantees on the proposed LN+PN and GN+PN. Overall, I think the paper is above the bar for acceptance.

Update: The review addresses most of my concerns and I have raised the score from 6 to 7.

**Time Spent Reviewing:**

8

---

> ### Author Response · Authors · 2021-08-10
> **Response to reviewer AgAg (Part 1/2)**
>
> We thank the reviewer very much for his/her time and for the useful and insightful feedback. We provide below a response to each comment.
>
> **Comment: “Guarantees on LN+PN/GN+PN: I might have entirely missed the point but I would appreciate it if there is a theorem or remark showing that LN+PN / GN+PN is able to avoid the collapsing effect.”**
>
> Originally, we did *not* provide a formal theoretical result on the guarantees with LN+PN/GN+PN. Instead, we specified these guarantees in a high-level discussion at the beginning of Section 5.
>
> To address the reviewer’s comment, we will turn this high-level discussion into a formal theorem.
>
> To be precise, we will add a theorem (“iterative guarantee of channel-wise normalization in proxy-normalized networks”) stating that if:
> - $\mathbf{x}$ is now sampled from a distribution $\nu_\mathbf{x}$ instead of a finite dataset $\mathcal{D}$,
> - $\tilde{\mathbf{z}}^{l-1}$ after PN-Act is channel-wise normalised,
> - The convolution and the norm do not play any role in the channel-wise collapse and the channel imbalance, i.e. $\mathcal{P}^{(1)}(\mathbf{y}^l)$ remains equal to 0 and $\mathcal{P}_c(\mathbf{y}^l)$ remain constant for all $c$,
> - $\mathbf{y}^l$ is Gaussian,
> - $\tilde{\boldsymbol{\beta}}^l$, $\tilde{\boldsymbol{\gamma}}^l$, $\epsilon$ are set to zero in PN,
>
> then $\mathbf{y}^l$ is channel-wise normalised, and $\tilde{\mathbf{z}}^l$ after PN-Act is channel-wise normalised.
>
> **Comment: “More empirical studies on the collapsing phenomenon in LN and GN. Currently the plot in Figure 3 is not very clear about the collapsing effect. $\mathcal{P}^{(1)}$ does grow, but it does not converge to 1, and the rest of the terms do not converge to 0, unlike suggested by the theorem. So I am wondering if the paper could improve the empirical studies to verify the collapsing effect."**
>
> As pointed out by the reviewer, Figure 3 is obtained with:
> - Residual networks of "effective" depth lower than the corresponding non-residual networks (see e.g. our discussion in Appendix A.2).
> - Zero-padding boundary conditions.
> - Model parameters considered on the training trajectory.
>
> On the other hand, Theorem 1 applies to:
> - Non-residual networks (i.e. networks without skip connections).
> - Periodic boundary conditions.
> - Model parameters considered as random (as per Definition 1) to quantify the "soft inductive bias".
>
> To better verify Theorem 1 and address the reviewer’s concern, we have conducted additional experiments in non-residual networks with periodic boundary conditions and random model parameters. The networks have $L=200$ layers, $C_l = 1024$ channels, kernel sizes $K_l=3$, activation function $\phi=\text{ReLU}$ and $\boldsymbol{\beta}^l\sim \mathcal{N}(0,0.2^2)$ and $\boldsymbol{\gamma}^l\sim \mathcal{N}(1,0.2^2)$.
>
> The results show that in layer-normalized networks:
> - There *is* an exponential collapse with a rate upper bounded (as expected) by the bound in Theorem 1.
> - The pseudo-linearity becomes exponentially stronger with depth, when measuring this pseudo-linearity with the additional measure $\mathcal{P}(\phi(\tilde{\mathbf{y}}^l)- \widetilde{\hspace{1pt}\mathbf{z}\hspace{1pt}}^l) / \mathcal{P}(\phi(\tilde{\mathbf{y}}^l))$ (where $\widetilde{\hspace{1pt}\mathbf{z}\hspace{1pt}}^l$ is the channel-wise linear best-fit of $\phi(\tilde{\mathbf{y}}^l)$ defined in Eq. (7) of the appendix).
>
> We provide the results in figures available at the following external anonymized link (viewing is at the discretion of the reviewer):
>
> https://figshare.com/s/fcec87d4fa36de094a37
>
> In the main text, we will insert the power plots from these additional experiments and discuss how they confirm the theoretical results of Sections 4 and 5. In appendix, we will add:
> - The plots of $\mathcal{P}(\mathbf{y}^l)-\mathcal{P}^{(1)}(\mathbf{y}^l)$ in log scale to precisely verify the upper bound of Theorem 1.
> - The plots of the additional measure of pseudo-linearity.
>
> **Comment: “Analysis depends on the use of ReLU activations”. “There have also been other developments of activation functions such as SeLU that have self-normalizing effects. I wonder if this can be baked into the analysis and lead to other insights? Instead of adding another normalization layer after the activation function.“**
>
> *Part of our theoretical analysis relating to the lack of variability in instance statistics (notably Theorem 2).* In fact, this part of our analysis does *not* rely on any assumption on the activation function. We will add this precision in the text.
>
> *Part of our theoretical analysis relating to channel-wise collapse (notably Theorem 1).* As pointed out by the reviewer, this part of our analysis does rely on the assumption of positive homogeneity of the activation function.
>
> Experimentally, however, all the non-saturating nonlinearities that we tried (positive homogeneous or not) *did* undergo exponential channel-wise collapse in non-residual networks with periodic boundary conditions and random model parameters (as per Definition 1).
>
> We found that this exponential collapse was still present when choosing SeLU as the activation function and either keeping or removing the norm. The reason, we believe, is that the assumptions of Klambauer et al. (2017) do not match with ours. Indeed, Klambauer et al. (2017) assumes that the biases of the neural network are zero, meaning that self-normalization is a property at initialization rather than a “soft inductive bias” in the full hypothesis space.
>
> In practice, we think that relying on SeLU would be detrimental performance-wise as it would limit our choice of activation function. In contrast, our approach based on LN+PN or GN+PN works by mimicking the behaviour/performance of the combination of BN with any choice of activation function (whatever choice is most favourable). To implement this approach, we simply replace BN layers with LN/GN layers and activation layers with proxy-normalized activation layers.
>
> **Reference**
>
> Klambauer et al., "Self-Normalizing Neural Networks", 2017

---

> > ### Author Response · Authors · 2021-08-10
> > **Response to reviewer AgAg (Part 2/2)**
> >
> > **Comment 1: “Reliance to be preceded by another normalization function: First of all, I don’t quite understand the requirement.” “I expect to see more concrete justification for this requirement and the transition of going into LN+PN and GN+PN is too fast. Secondly, doing a double normalization makes the computation less elegant.”**
> >
> > **Comment 2: ”The reason for a proxy distribution: It also was not clear to me what is the purpose for adding a proxy distribution. Why not just normalize the post-activations using the statistics from themselves?”**
> >
> > We understand that the rationale of our approach based on the combination of LN/GN has been detailed too quickly and not clearly enough.
> >
> > Our goal in this work is to retain all the beneficial properties of BN (detailed in Section 2) with a batch-independent normalization. Notably, we aim at avoiding the common failure modes of batch-independent normalization: channel-wise collapse and alteration of expressivity. We achieve this by combining LN/GN with PN, i.e. by replacing all BN layers with LN/GN layers and all activation layers with proxy-normalized activation layers.
> >
> > The reason for using PN is that it enables to alleviate the failure of channel-wise collapse for "free", i.e. without causing as a flip side an aggravation of the alteration of expressivity. If we were to use *only* LN/GN/IN layers, on the other hand, any alleviation of one failure mode would lead to an aggravation of the other failure. This means that even if we inserted multiple of these norms LN/GN/IN at each depth $l$ in the network, we would never avoid both failure modes simultaneously.
> >
> > The reason for still keeping LN/GN in our approach is twofold:
> > - We need to be sure that "proxy" activations in PN correctly track "real" activations. For this, we need $\mathbf{y}^l$ to be at the same scale as the proxy $Y^l$ in PN (whose distribution is close to $\mathcal{N}(0,1)$).
> > - Even if avoiding the channel-wise collapse and alteration of expressivity, we still need to retain the properties of "scale invariance" (w.r.t. weights) and "control of activation scale" (cf Section 2). If we used only PN, these two latter properties would not be retained.
> >
> > For clarification, we will add the above discussion in Section 5, after stating the theoretical guarantee with PN.
> >
> > Another precision: while we understand that using two norms might seem less elegant, part of this perception could be a matter of conceptualization/terminology. Indeed, if we see PN as a plug-in attached to the activation function rather than as an additional norm, our proposal consists in replacing BN+Act with LN+PN-Act or GN+PN-Act.
> >
> > **Comment: “Computing statistics analytically: If we assume that we have a ReLU activation, can we analytically compute the mean and variance of the proxy distribution?”**
> >
> > Thank you very much for raising this point. With ReLU, there are indeed closed-form expressions of the “proxy” mean and variance:
> > - $ \mathbb{E}[Z^l] = \mu f(\hat{\mu}) + \sigma F(\hat{\mu})$,
> > - $\mathrm{Var}[Z^l] = \sigma^2 (\hat{\mu} f(\hat{\mu}) + (\hat{\mu}^2+1) F(\hat{\mu}) ) - \mathbb{E}[Z^l] ^2$,
> >
> > where:
> > - $\mu$, $\sigma$ are the mean and standard deviation of the “proxy“ activations before the activation ($\mu = \boldsymbol{\gamma}^l, \tilde{\boldsymbol{\beta}}^l + \boldsymbol{\beta} ^l$ and $\sigma = \boldsymbol{\gamma}^l (1+\tilde{\boldsymbol{\gamma}}^l) $),
> > - $\hat{\mu} = \mu / \sigma$,
> > - $f$ and $F$ are the p.d.f and c.d.f. of a standard Gaussian variable.
> >
> > Unfortunately: (1) these closed-form expressions still involve “expensive” operations, and (2) for other activation functions (e.g. Swish/SiLU) closed-form solutions are not always available.
> >
> > Until now, for the sake of generality, we favoured the implementation provided in Appendix B.
> > In this implementation, the proxy distribution is derived from a finite sample approximation of $\mathcal{N}(0,1)$ that is generated *once and for all before training* and *reused at each iteration*. This implementation enables us to plug PN onto any activation function without having to work out closed-formed expressions or “craft” analytic approximations each time we change the activation function.
> >
> > We plan to keep working on improving the practical efficiency and ease of implementation of PN to facilitate its adoption by the community.

---

### Official Review · Reviewer_Wjpi · 2021-07-26

**Rating:** 6
**Confidence:** 5

**Summary:**

This paper studies normalization strategies for deep residual networks, both theoretically and empirically, and finds the strategies that don't rely on batch-statistics to have particular failure modes, the two most notable ones being channel-wise collapse, where instances collapse to a constant in each channel, and reduction in instance-variability, where the first and second order statistics per channel are identical for each instance, thereby reducing expressivity and the ability to recognize higher order concepts. The paper then proceeds to propose proxy-normalization, a normalization method suitable for small-batch learning (i.e., not relying on batch-statistics) that, when combined with other normalization strategies, such as Layer Norm or Group Norm, alleviates these failure modes. On ImageNet and CIFAR experiments with deep residual networks (RN50 and larger), the work finds that Proxy-Normalization (with sufficient additional regularization such as label-smoothing, dropout, and stochastic depth), can be used to match the performance of BN without relying on batch statistics. The proposed method standardizes the activations of the network according to the mean and variance of a channel-wise gaussian random variable, with learnable mean and variance parameters.

**Limitations And Societal Impact:**

No clear negative societal impacts.

**Main Review:**

### Originality & Significance
- Improving residual normalization strategies to have no dependence on batch statistics is an important problem with great practical applicability. A broad body of work has been dedicated to this problem.
- The proposed Proxy-Normalization strategy looks novel to the best of my knowledge, but is somewhat related to existing normalization strategies based on proxy-statistics of the activations [1,2,3].
- The power decomposition is quite intriguing, and provides a nice grounding for subsequent arguments about normalization strategies and failure modes. Since such power quantities can be measured, it also makes it easy to directly verify these arguments with empirical observations, as the authors do in Figure 3. Their logic is as follows: Power concentration in P1 (at the dataset scale) means all instances have collapsed in that channel (equal to a constant). Collapsing channels result in a pseudo-linearity failure, whereby the network does not effectively utilize its depth. Lack of power concentration in P2 and P4 imply that all instance activations have the same exact mean and variance (no inter-instance variability). Lack of inter-instance variability (low power in P2 and P4) reduces expressivity of the network, making it more difficult to recognize higher order concepts.

### Clarity
The paper is well written and relatively clear, however some details are unclear.
- For example, from my understanding C_k in theorem 1 is the number of channels at k. Can you please clarify this point?
- The arguments relating inter-instance variability to expressivity is also not entirely clear and is a little hand-wavy.

### Quality
There are sufficient technical contributions in the paper, both empirically and theoretically. However, there are limitations
- The proposed Proxy Normalization method while conceptually simple, introduces some non-trivial complexity into training, in having to sample from a Gaussian random variable with learnable parameters, and then feed that into the activation before standardizing.
- The proposed proxy-normalization is demonstrated to address the channel collapse issue (large P1), but is not demonstrated to address the inter-instance variability issue. The concentration of power in the dataset scale (channel collapse) is more much more stark with LN and GN, but the improvement in inter-instance variability (expressivity), to be determined by P2+P4 moving away from 0, is not as significant in Figure 3, but I suppose that according to the provided arguments LN and GN should not really have this issue.
- The proposed method requires additional regularization factors to match BN performance, and from my parsing of the appendix, these factors seem to vary from one architecture to another. Furthermore, it is unclear which regularization strategies are necessary or why. Such an ablation and detailed understanding would significantly improve the paper.


[1] Arpit et al., Normalization propagation: A parametric technique for removing internal covariate shift in deep networks, 2016.

[2] Laurent et al., Recurrent normalization propagation, 2017.

[3] Shekhovtsov et al., Normalization of neural networks using analytic variance propagation, 2018.

**Time Spent Reviewing:**

5

---

> ### Author Response · Authors · 2021-08-10
> **Response to reviewer Wjpi**
>
> We thank the reviewer very much for his/her time and for the insightful comments. We provide below a response to each comment.
>
> **Comment: "Proxy-Normalization strategy looks novel to the best of my knowledge, but is somewhat related to existing normalization strategies based on proxy-statistics of the activations."**
>
> The main elements of novelty of our approach are (we will state explicitly these elements in Section 5):
> - *The ability to successfully scale to large models*. To the best of our knowledge, such a successful scaling has never been shown with previous approaches based on proxy statistics. In fact, Shang et al. (2017) shows that, even after non-trivial adaptations, Normalization Propagation still incurs a performance degradation compared to BN in residual networks.
> - *The tractability and ease of implementation of PN*. All previous approaches required the tracking of statistics throughout the network, whereas our approach only requires a local tracking. This eases the implementation, notably when models are sharded in a distributed setting.
> - *The inclusion of the correction of the affine transform when training with PN*. Such a correction is not included in Normalization Propagation. The impact of this grows as the depth increases.
>
> **Comment: "from my understanding $C_k$ in theorem 1 is the number of channels at $k$. Can you please clarify this point?"**
>
> Yes, that is correct. This notation is introduced in Section 2. We will emphasize the introduction of this notation in Section 2 and provide a reminder of it in Theorem 1.
>
> **Comment: "The arguments relating inter-instance variability to expressivity is also not entirely clear and is a little hand-wavy."**
>
> We will clarify these arguments.
>
> Firstly, we will add the reference Huang et al. (2021) pointed out by another reviewer that discusses how the constrains imposed by IN ($2 C_l$ constraints per sample) lead to an alteration of expressivity.
>
> Secondly, we will add an argument based on the universal approximation theorem:
> - On the one hand, the universal approximation theorem implies that (provided the width is large enough) we can find unnormalized networks as close as we want to the identity, i.e. with $\sup ||\mathbf{x} -\Phi^l(\mathbf{x})||$ arbitrarily close to 0, where $\tilde{\mathbf{y}}^l=\Phi^l(\mathbf{x})$ for $\mathbf{x} $ in some ball $B(0,r)$ with $r>0$.
> - On the other hand, instance-normalised networks can never arbitrarily approach the identity since they always have $\sup ||\mathbf{x} -\Phi^l(\mathbf{x})||\geq C_r$ for some $ C_r >0$ dependent on the choice of $r$. This can be seen by considering some $\mathbf{x}_0$ with non-zero instance means, and by noting that $\Phi^l(0)$ and $\Phi^l(\mathbf{x}_0)$ have the same instance means. If $\sup ||\mathbf{x}-\Phi^l(\mathbf{x})||$ could be made arbitrarily close to 0, then instance means of $\Phi^l(0)$ would be made arbitrarily close to 0 and, simultaneously, instance means of $\Phi^l(\mathbf{x}_0)$ would be made arbitrarily close to the non-zero instance means of $\mathbf{x}_0$. This would lead to a contradiction.
>
> **Comment: "The proposed Proxy Normalization method while conceptually simple, introduces some non-trivial complexity into training, in having to sample from a Gaussian random variable with learnable parameters, and then feed that into the activation before standardizing."**
>
> *PN's implementation.* While PN may appear to add complexity at first look, we think that its practical implementation is actually simple.
>
> In our practical implementation provided in Appendix B, the proxy distribution is derived from a finite sample approximation of $\mathcal{N}(0,1)$ that is generated *once and for all before training* and *reused at each iteration*.
>
> Regarding the learnable parameters, the parameters $\boldsymbol{\beta}^l$, $\boldsymbol{\gamma}^l$ are already present without PN. As for $\tilde{\boldsymbol{\beta}}^l$, $\tilde{\boldsymbol{\gamma}}^l$, a new set of experiments that we conducted to answer another reviewer’s comments shows that the drop of performance that would result from dropping them is in fact low. We provide below these new results (single run per case, with formatting X / Y where X, Y are the results without and with extra regularization, respectively):
>
> || RN50        | RN101       | RNX50       | RNX101      | depthwise convs EN-B0       | depthwise convs EN-B2       | group convs EN-B0       | group convs EN-B2       |
> |:-----------------------------|:------------|:------------|:------------|:------------|:------------|:------------|:------------|:------------|
> | GN+PN w/ $\tilde{\boldsymbol{\beta}}^l$,$\tilde{\boldsymbol{\gamma}}^l$  | 76.3 / **76.7** | 77.6 / **78.6** | 76.7 / **77.8** | 77.7 / **79.0** | 76.7 / **77.0** | 79.4 / **80.0** | 76.6 / **76.8** | 79.3 / **80.1** |
> | GN+PN w/o $\tilde{\boldsymbol{\beta}}^l$,$\tilde{\boldsymbol{\gamma}}^l$ | 76.3 / **76.8** | 77.5 / **78.5** | 76.4 / **77.6** | 77.7 / **78.9** | 76.7 / **76.7** | 79.3 / **79.8** | 76.5 / **76.6** | 79.2 / **79.9** |
>
> We will provide these additional results in appendix (with multiple runs per case). We will also state in Section 5 that $\tilde{\boldsymbol{\beta}}^l$, $\tilde{\boldsymbol{\gamma}}^l$ are in fact optional.
>
> *PN's FLOPs.* PN's operations on "proxy" activations typically involve many fewer FLOPs than the operations on the “real” activations that they mirror. That is because "proxy" activations have dimensionality $\sim 200 \times C_l$, whereas “real” activations have dimensionality $B \times H \times W \times C_l$, with $B$, $H$, $W$, $C_l$ the batch size, height, width and number of channels.
>
> **Comment: "The proposed proxy-normalization is demonstrated to address the channel collapse issue (large $\mathcal{P}^{(1)}$), but is not demonstrated to address the inter-instance variability issue."**
>
> As the reviewer points out, PN neither improves nor worsens the issue relating to inter-instance variability.
>
> That is the reason behind our choice of combining PN with either LN or GN with a small number of groups $G$. Just like LN or GN with small $G$, this combination is only weakly affected by the issue relating to inter-instance variability. And thanks to PN, this combination is only weakly affected by the issue of channel-wise collapse.
>
> We will clarify the rationale behind the combination of LN/GN with PN in Section 5.
>
> **Comment: "The proposed method requires additional regularization factors to match BN performance, and from my parsing of the appendix, these factors seem to vary from one architecture to another."**
>
> Our choices of additional regularization mainly follow Brock et al. (2021). These choices were not cherry-picked. The reason for our different choice of additional regularization in EfficientNets is that these networks natively incorporate the regularization forms that we add in ResNets/ResNeXts (label smoothing, dropout, stochastic depth).
>
> To further address the reviewer's comment, we have conducted new experiments in ResNets/ResNeXts with GN/GN+PN and with the same choice of additional regularization as in EfficientNets. The results are as follows (single run per case, with formatting X / Y where X, Y are the results without and with extra regularization, respectively):
>
> |        | RN50            | RN101           | RNX50           | RNX101          |
> |:----------------------------------------------|:----------------|:----------------|:----------------|:----------------|
> | GN    | 75.4 / **75.8** | 77.0 / **77.6** | 76.2 / **76.6** | 77.4 / **78.2** |
> | GN+PN | 76.3 / **77.0** | 77.6 / **78.9** | 76.7 / **77.6** | 77.7 / **79.7** |
>
> The qualitative conclusions on the comparison between GN and GN+PN remain unchanged. We will add these results in appendix (with multiple runs per case and with the inclusion of BN) together with a discussion on them.
>
> **References**
>
> Shang et al., "Exploring Normalization in Deep Residual Networks with Concatenated Rectified Linear Units", 2017
>
> Huang et al., “Group Whitening: Balancing Learning Efficiency and Representational Capacity”, 2021
>
> Brock et al., "Characterizing signal propagation to close the performance gap in unnormalized ResNets", 2021

---

> > ### Comment · Reviewer_Wjpi · 2021-09-02
> > **Response to Authors**
> >
> > I have read the author responses and other reviews
> >
> > The authors have addressed my concerns about the implementation issues, and I also think the theoretical motivations and exposition for the proposed proxy-normalized method are fascinating.
> >
> > However, as pointed out by other reviewers, the method still needs to be combined with other normalization strategies (LayerNorm/GroupNorm) to work, and the empirical improvements in model performance with vs without proxy-normalization + LN/GN is somewhat mild.
> >
> > In light of these points, I would definitely lean more on the positive side, and will maintain my current recommendation.

---

### Author Response · Authors · 2021-08-10
**General response to reviewers**

We thank the reviewers for their time and effort in reviewing our paper. We are grateful that reviewers appreciated the novelty (R1, R3) and significance (R1) of our work, thought that our paper was both exciting and insightful (R2), and considered our results to be solid (R2) and comprehensive (R3).

We understand that some reviewers expressed concerns regarding the lack of elegance (R2), practicality (R3), simplicity (R1) and scope (R1) of our solution. Two reviewers also commented on the relation to existing strategies (R2) and a lack of clarity around the reason for a proxy distribution (R2).

We have responded to each reviewer individually to address these concerns and clarify specific points.

The reviewer's comments have greatly helped us to identify areas where we can reinforce the claims and contributions of the paper. They have notably led us to perform new experiments that strengthen our claims, as discussed in the individual responses to reviewers.

Based on the current reviewer suggestions and feedback, we will revise the manuscript by implementing the changes that we detail in the individual responses. We will be happy to clarify any other concerns raised by the reviewers.

---

> ### Author Response · Authors · 2021-09-02
> **Thank you for the post-rebuttal feedback**
>
> We wish to thank all the reviewers for their post-rebuttal feedback.

---

### Decision · Program_Chairs · 2021-09-27

**Decision:**

Accept (Poster)

**Comment:**

The paper proposes a theoretical analysis on different normalization schemes. It demonstrates that normalizing schemes which don't rely on batch-statistics suffer from channel collapse or lack or expressivity.  It then proposes a novel normalization scheme to address those issues.

Although the practicality of the algorithm could be improved, proxy-norm is able to match BN while not using the batch statistics. Additionally, the analysis presented in the paper is novel and reviewers agreed that it would be of value to the community.